# The brain hierarchically represents the past and future during multistep anticipation

Hannah Tarder-Stoll [1,2] ✉, Christopher Baldassano [1,4] & Mariam Aly [1,3,4]

Memory for temporal structure enables both planning of future events and retrospection of past events. We investigated how the brain flexibly represents extended temporal sequences into the past and future during anticipation. Participants learned sequences of environments in immersive virtual reality. Pairs of sequences had the same environments in a different order, enabling context-specific learning. During fMRI, participants anticipated upcoming environments multiple steps into the future in a given sequence. Temporal structure was represented in the hippocampus and across higher-order visual regions (1) bidirectionally, with graded representations into the past and future and (2) hierarchically, with further events into the past and future represented in successively more anterior brain regions. In hippocampus, these bidirectional representations were context-specific, and suppression of far-away environments predicted response time costs in anticipation. Together, this work sheds light on how we flexibly represent sequential structure to enable planning over multiple timescales.

Memory allows us to use past experience to generate expectations about the future. Integration of past information to predict future events enables efficient planning and flexible behavior in complex environments[1–4] and has been proposed to be a primary function of memory systems[5] and of the brain itself[6,7]. For predictions to usefully impact behavior, they should be represented on multiple timescales, allowing us to anticipate not just immediately upcoming events but also events further in the future. Furthermore, predictions that are relevant for the current context should be flexibly prioritized over those that are less relevant. For example, when riding the subway, it would be useful to anticipate multiple stations ahead on the relevant line, but we need not anticipate upcoming stops on other lines passing through the same stations. Such context-specific prediction may be supported by leveraging memories of past stops, which contextualize where we are in the present. Here, we aimed to test three central hypotheses. First, that the brain will flexibly anticipate events at multiple timescales in the future; second, that the future and the past will be represented simultaneously in the same brain regions; and third, that anticipatory representations will be prioritized for events that are contextually relevant.

To test these hypotheses, we drew on prior research showing anticipatory signals across the brain, particularly in memory and sensory systems[5,8–12]. For example, predictions about upcoming items or locations in a sequence are represented in the visual cortex[13–16] and hippocampus[17–21], suggesting coordination between these regions in memory-based prediction of visual stimuli[16]. Although earlier research on prediction typically focused on one or a few brain regions[13,15–17,20] and predictions about immediately upcoming events[13,16,20], more recent work has shown that the brain represents anticipatory signals at multiple timescales simultaneously, with shorter timescales of prediction in more posterior regions and successively longer anticipatory timescales in progressively more anterior regions[12,22]. For example, during repeated viewing of a movie clip, posterior regions like the visual cortex primarily represent the current moment, while progressively more anterior regions represent upcoming events successively further into the future[12]. This past work has also highlighted the insula as a brain region that shows particularly far-reaching predictions[12] perhaps due to its role in generating social predictions during naturalistic events[23]. These findings of multistep anticipatory signals are generally consistent with computational theories that the brain builds

[1]Department of Psychology, Columbia University, New York, USA. [2]Rotman Research Institute, Baycrest Health Sciences, Toronto, Canada. [3]Department of Psychology, University of California, Berkeley, Berkeley, CA, USA. [4]These authors contributed equally: Christopher Baldassano, Mariam Aly. ✉e-mail: htarder-stoll@research.baycrest.org

models of the world that cache temporal information about successive events, with different predictive timescales in different brain regions (i.e., multi-scale successor representations[22,24]).

This research on multiscale anticipation in the brain complements earlier work showing hierarchical representations of past states[25]. Mirroring the predictive hierarchy for future states[12,22], information from the past lingers in the brain during ongoing experience, with shorter timescales of past information represented in posterior regions and longer timescales in anterior regions[25–29]. In the hippocampus specifically, temporal coding in the form of sequence reactivation can extend into the past or the future[30–33]. Furthermore, the brain's representations of the past and future can be flexibly modulated based on task demands[34], with past and future states represented distinctly from current states in the hippocampus[35]. Although this work suggests that the brain may represent both anticipated and past events, these prior studies did not test whether forward and backward representations of temporally extended structure existed simultaneously in the same brain regions. Such bidirectional representations would accord with temporal context models that propose that events experienced nearby one another come to be associated with similar representations, such that the retrieval of a given item can be a strong cue for both preceding and subsequent items[36,37]. We therefore examined whether the brain contains bidirectional representations of the past and future, with the scale of these representations varying systematically across the brain.

For the brain's representations of temporal structure to be adaptive for behavior, they should flexibly change depending on context. Recent work in humans has shown context-specific patterns of activity in the hippocampus during goal-directed planning of future trajectories, suggesting that anticipation of temporally structured experience is specific to the upcoming items in a given context[38]. However, it remains unknown whether contextual modulation of temporal structure representations is specific to planning trajectories in the forward direction or if associations with preceding items are also activated in a context-specific way. If bidirectional representations help disambiguate overlapping contexts – which may overlap in the future, the past, or both – then context-specific representations of both past and future states would be useful in planning trajectories within a context.

In the present study, we investigated how context-specific temporal structure is represented in the brain during a novel multistep anticipation task. Participants learned, in immersive virtual reality, four temporally extended sequences of eight environments each (Fig. 1). Critically, pairs of sequences (Green Path vs. Blue Path) contained the same environments in a different order, requiring individuals to flexibly anticipate environments based on the current sequence context.

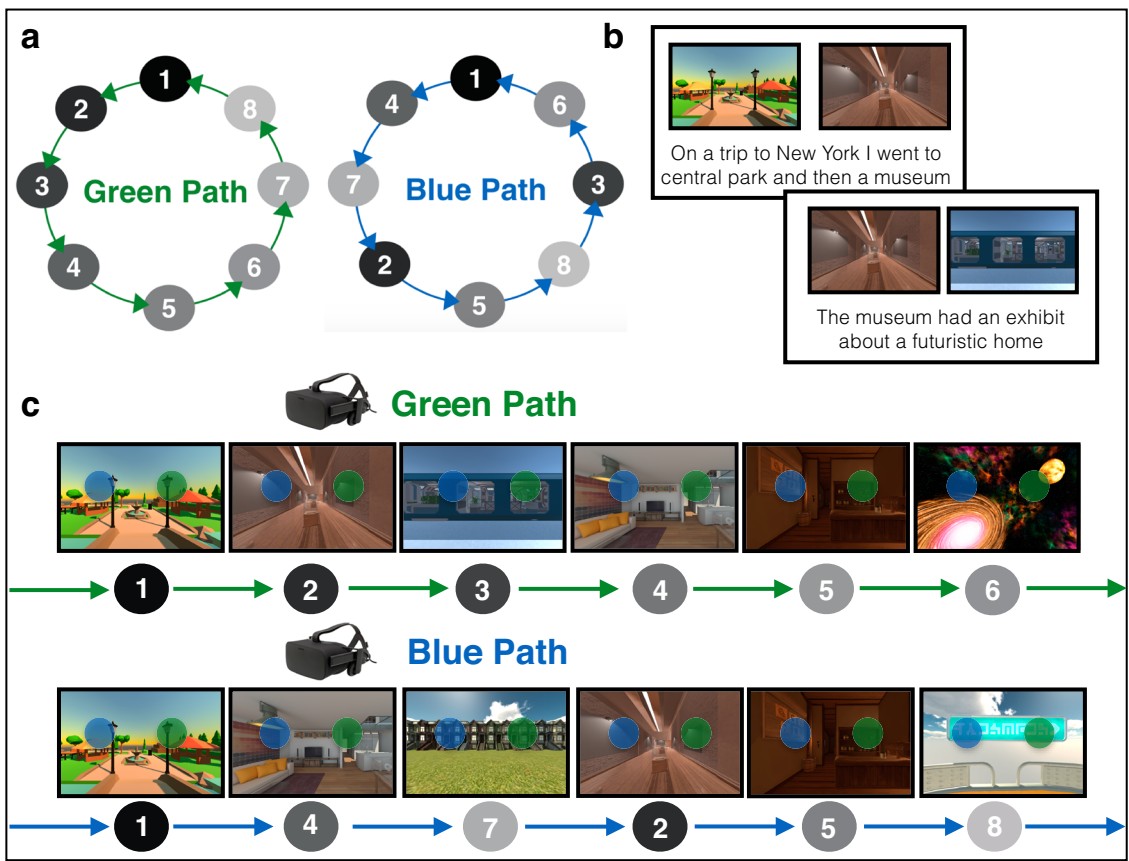

**Fig. 1 | Sequence structure and behavioral training. a** Sequence structure. Participants learned sequences of eight environments, indicated by the gray nodes. The green path and the blue path consisted of the same environments in a different order. The sequences were constructed to be as distinct as possible: for each environment the two preceding and two succeeding environments were different across the sequences. Participants learned four sequences in total: one green and blue path with a set of eight environments, and another green and blue path with a different set of eight environments. Only one green and one blue path are depicted here for illustrative purposes. **b** Story Generation. To learn the sequence of environments, participants generated stories for each path to link the environments in order. Participants were told to link the final environment back to the first environment to create a loop. **c** Virtual Reality Training. Participants then explored the environments in immersive virtual reality in the green path order and the blue path order while rehearsing their stories. In a given environment, a green and blue sphere would appear. These spheres, when touched, teleported the participant to the next environment in the corresponding (green or blue) sequence. Participants then recalled the order of each of the four sequences (not shown). Environment images are screenshots of 3D environments created in the game engine Unity from assets available for commercial use.

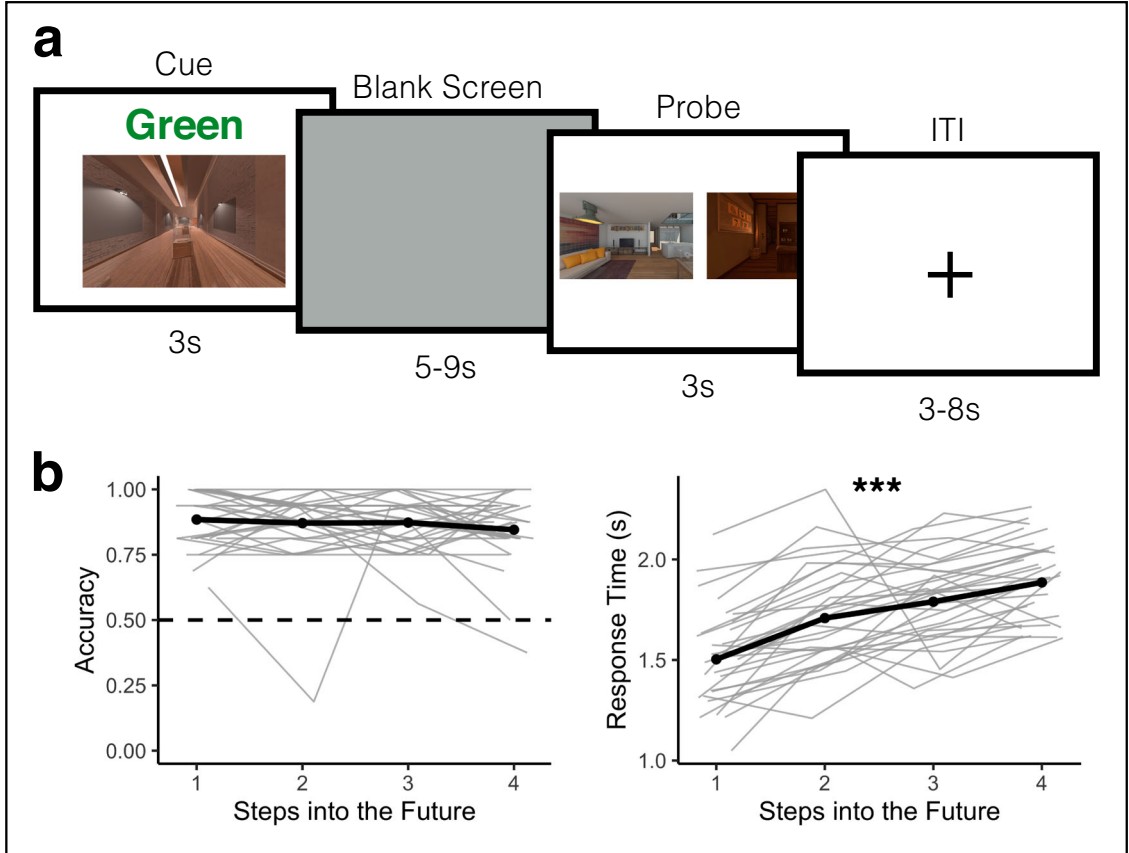

**Fig. 2 | Anticipation task and behavioral performance. a** Anticipation Task. Participants returned one day after behavioral training and completed the Anticipation Task inside the MRI scanner. Participants were cued with a 2D image of an environment from one of the sequences along with a path cue (Green or Blue) for 3 s. They then saw a blank screen for a variable duration of 5 to 9 s during which they were told to anticipate upcoming environments. Participants were then probed with two images of upcoming environments and had 3 s to indicate which of the two environments was coming up sooner in the cued sequence, relative to the cue image. The correct answer could be 1 to 4 steps away from the cue image.

**b** Behavioral Performance. Participants accurately anticipated upcoming environments in the cued sequence. Accuracy did not significantly differ across steps into the future (left). Response time, however, was significantly slower for further steps into the future (right). Pale gray lines indicate data for individual participants; the black line is the group mean. Source data are provided as a Source Data file. $N = 32$ participants. Statistical tests are based on a generalized linear mixed effects model (left) and a linear mixed effects model (right); significance threshold is two-tailed $p < 0.05$; *** $p < 2.2e\text{-}16$.

Sequences were circular, such that environments were temporally predictable multiple steps into the future and the past regardless of location in the sequence. This allowed us to test whether temporal structure in both the prospective and retrospective direction is automatically represented in the brain even if only future states are task-relevant.

Following sequence learning, participants were scanned with fMRI as they completed an Anticipation Task, in which they anticipated upcoming environments one to four steps into the future in a given (cued) sequence (Fig. 2). Following the Anticipation Task, participants completed a Localizer Task in which we obtained template patterns of brain activity for each environment (Fig. 3). We used these templates to conduct multivoxel pattern similarity analyses in visual cortex, hippocampus, insula, and across the brain. Specifically, while participants viewed a given cue environment and attempted to anticipate upcoming environments in the Anticipation Task, we looked for multivoxel evidence of surrounding environments in the sequence. Using this approach, we determined the extent to which temporal structure was (1) represented in a graded and bidirectional manner, with simultaneous representations of future and past environments; (2) represented in a hierarchical fashion among lower and higher order brain regions, with further-reaching representations in higher-order regions; and (3) modulated by context, with prioritized representations for the cued vs uncued sequence.

## Results

### Anticipation Task Performance

Participants performed effectively on the Anticipation Task (Fig. 2a), correctly choosing the closer of the two probe images, relative to the cued image and path, 86.86% of the time (sd = 0.08), which was significantly higher than the chance performance of 50% ($t(31) = 61.08$, $p < 0.00001$). There was a trend toward higher accuracy in Map A (i.e. the first learned map) compared to Map B (beta = −0.283, 95% CI = [−0.576, 0.001], $p = 0.058$). Accuracy did not vary by path (Green vs Blue, beta = −0.124, 95% CI = [−0.458, 0.209], $p = 0.464$), nor was there a map by path interaction (beta = 0.268, 95% CI = [−0.274, 0.81], $p = 0.333$). Response times on the Anticipation Task were not significantly influenced by map (beta = 0.029, 95% CI = [−0.014, 0.072], $p = 0.185$), path (beta = 0.038, 95% CI = [−0.009, 0.085], $p = 0.127$), or their interaction (beta = −0.052, 95% CI = [−0.134, 0.029], $p = 0.209$).

We next determined how performance on the Anticipation Task varied by steps into the future (i.e. how many steps the correct probe was from the cue image on the cued path; also see ref. 39). Steps into the future had a trending impact on accuracy (beta = −0.15, 95% CI = [−0.33, 0.027], $p = 0.096$; Fig. 2b), with an average difference in accuracy of 3.9% between one-step and four-step trials. Steps into the future robustly impacted response time (beta = 0.13, 95% CI = [−0.103 0.149], $p < 0.000001$; Fig. 2b). Responses were on average 126 ms slower for each step into the future, with an average difference of

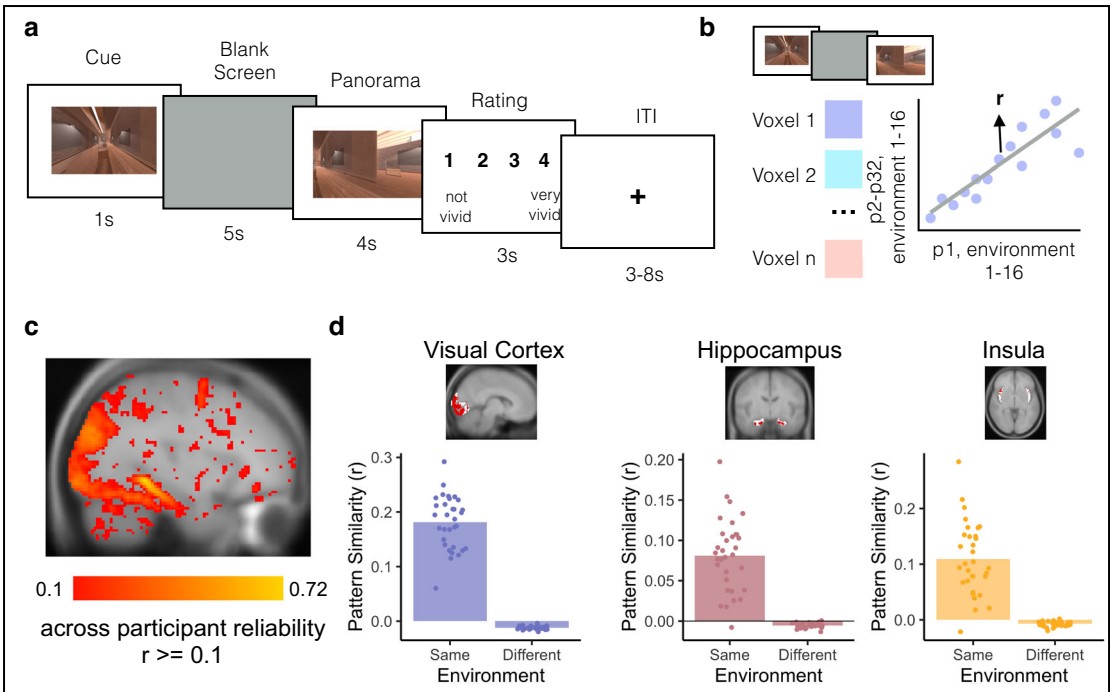

**Fig. 3 | Template brain activity patterns for each environment. a** Localizer Task. Participants completed a Localizer Task inside the MRI scanner at the end of the session. Participants were cued with a 2D image of an environment from the experiment for 1 second. They then saw a blank screen for 5 s during which they were told to imagine being inside the environment in VR. Next, they saw images of the environment from different angles for 4 s and were given 3 s to rate how well their imagination matched the actual images of the environment. **b** Across-participant analysis for identifying voxels that reliably discriminate between environments. We measured the activity of each voxel in each participant during the Localizer Task (combining the cue, blank screen, and panorama phases) for each of the 16 environments. Next, we obtained the Pearson correlation (r) in each voxel between a participant's (e.g., P1's) responses to the 16 environments and the 16 average responses in the remaining participants (e.g., P2-P32). Averaging across

all choices of the left-out participant, this yielded an across-participant reliability score for each voxel. **c** Whole brain map of voxels that reliably discriminate between environments. We only included voxels in subsequent analyses if they had an across-participant environment reliability value of 0.1 or greater and were part of a cluster of at least 10 voxels. **d** Environment representations in ROIs. In early visual cortex (left), hippocampus (middle), and insula (right), we selected the environment-reliable voxels (red) within each anatomically or functionally defined ROI (white). We then confirmed that the analysis successfully identified across-participant patterns of activity within these conjunction ROIs, i.e., activity patterns that were more correlated for the same environment than for different environments. Bars indicate average pattern similarity. Points indicate the pattern similarity between each participant's activity pattern with the mean of all other participants, averaged across all environments. *N* = 32 participants.

380 ms between one-step and four-step trials. Together, this suggests that participants performed accurately on the Anticipation Task, but were slower to anticipate upcoming environments that were further into the future.

## Bidirectional and graded representations of temporal structure in hippocampus

For our MRI analyses, we first created conjunction ROIs by selecting voxels within early visual cortex (V1-4), hippocampus, and insula that reliably responded to distinct environments in the Localizer Task (Fig. 3a–c). Next, we obtained the across-participant multivoxel pattern of brain activity for each environment within each of our ROIs (Fig. 3d). To investigate neural representations of temporal structure during multi-step anticipation, we calculated pattern similarity between (1) multivoxel patterns of brain activity evoked during the Anticipation Task for each trial type (cue and path combination) for each participant and (2) the multivoxel patterns of brain activity evoked during the Localizer Task, averaged across the remaining participants, for each environment on the same map (Fig. 4a; see Methods). We then ordered the resulting correlation values in the sequence of the cued path with the cued environment in the center, successors following the cue to the right of the center, and predecessors to the left of the center (Fig. 4a; see Methods). Importantly, because the order of environments in the sequences was randomized across participants, the across-participant multivoxel pattern of activity during the Localizer Task cannot include information about

successors for individual participants' sequences, resulting in a relatively pure measure of environment representations. Thus, this analysis allows us to determine the extent to which our regions of interest represented upcoming or preceding environments during the Anticipation Task, using activity pattern "templates" for each environment that were constructed to remove information about sequence structure.

We examined whether the sequence order for the cued path was reflected in the brain's representations during the Anticipation Task. We developed an analysis approach that allowed us to: 1) detect representations of past and future environments; 2) test whether such representations were graded as a function of distance and specific to the task-relevant context; and 3) determine if these representations were hierarchically organized across the brain. Our approach involved fitting an asymmetrical Gaussian curve to the pattern similarity values arranged following the order of the cued path (see Methods). The Gaussian similarity model has four parameters: amplitude, asymptote, and forward and backward width (σ) (Fig. 4b). The amplitude of the curve indicates the degree to which a brain region is representing the cue environment while it is on the screen. The forward and backward widths (σ) of the curve indicate how similarity to neighboring environments falls off with the number of steps in the forward and backward directions. Wider (vs. narrower) widths indicate that the brain region represents environments that are further away. If a brain region has a wide forward width but a narrow backward width, this indicates a bias towards representing upcoming environments,

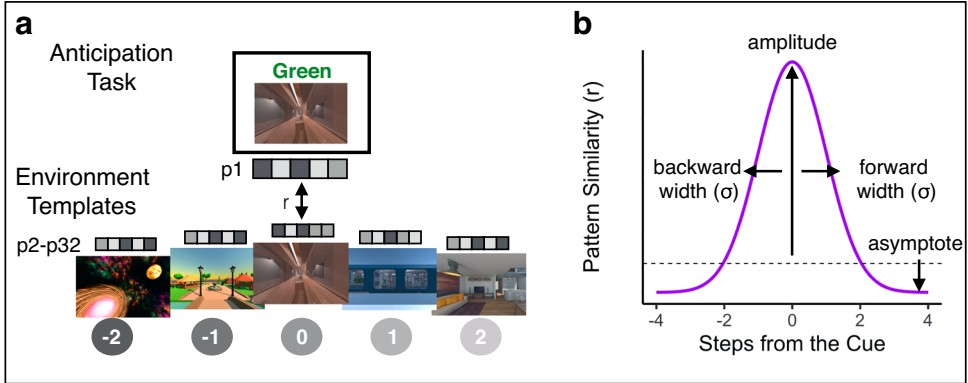

**Fig. 4 | Schematic Depiction of the Gaussian Analysis Approach. a** Pattern similarity analysis for the Anticipation Task. We obtained the correlation between a given participant's (e.g., P1) cue screen activity pattern for each trial of the Anticipation Task and the remaining participants' (e.g., P2-P32) averaged activity patterns for each of the environment templates on the cued path. We then ordered the resulting pattern similarity values with the cue in the center and fit an asymmetrical Gaussian curve. Environment images are screenshots of 3D environments created in the game engine Unity from assets available for commercial use. **b** Gaussian similarity model. The amplitude of the curve indicates the degree to which a brain region is representing the cue environment while it is on the screen. The widths (σ) of the curve indicate how similarity to neighboring environments falls off with the number of steps in the forward and backward directions. Wider (vs. narrower) widths in a brain region indicate further environment representations. The asymptote quantifies the representations of environments that are not captured by the width of the Gaussian; if the asymptote is lower than the dashed line (different-map baseline) this suggests that these environments are suppressed.

indicating anticipation, over retrospective representations of preceding environments. The asymptote is an indication of the representations of environments that are not captured by the width of the Gaussian; if the asymptote is lower than baseline (defined as pattern similarity between the cue screen activity patterns and all environment templates from the other map, henceforth referred to as different-map baseline), this suggests that these environments are suppressed relative to our baseline condition. Importantly, our pattern similarity approach considers independent time points (similarity between our Anticipation Task cue periods and environment templates from the Localizer Task), meaning that more evidence for the cued environment does not necessitate suppression of other environments – suppression need not be observed, and all pattern similarity values could be positive.

We statistically tested the Gaussian fit in two ways. First, we compared the goodness-of-fit for the Gaussian model when applied to the correctly ordered pattern similarity values vs. a shuffled-order version of the pattern similarities, including pattern similarity to the cue environment ("shuffled null including cue"; Fig. 5). This allows us to test the null hypothesis that there was no structure in the similarity values. Second, we removed the pattern similarity to the cue environment and fit the Gaussian model only to the pattern similarities for upcoming and past environments, in both the correct order and the shuffled order ("shuffled null excluding cue"; Fig. 5). If a brain region shows a superior Gaussian fit for the observed vs. shuffled data only when the cue environment is included but not when it is excluded, that would indicate that its Gaussian fit is entirely driven by its representation of the cue environment. If a brain region shows a superior Gaussian fit for the observed vs. shuffled data both when the cue environment is included and when it is excluded, that would indicate that this region represents the cue environment and also has systematically graded representations of nearby environments.

In visual cortex, the Gaussian model provided a significantly better fit to the correctly ordered vs. shuffled pattern similarity values when the cue environment was included in both the observed and shuffled data (correctly ordered data vs. shuffled null including cue, $p < 0.001$, $R^2 = 0.699$). The amplitudes of participants' Gaussian fits were significantly higher than the different-map baseline, indicating that the cue environment was represented while it was on the screen (mean = 0.091, standard deviation = 0.029; t(31) = 18.01, $p < 0.000001$; Fig. 5a). The asymptote was significantly lower than the different-map baseline, suggesting that other environments surrounding the cue

were suppressed relative to our baseline (mean = −0.014, standard deviation = 0.009; t(31) = −5.97, $p = 0.000001$; Fig. 5a). The backward and forward widths (σ) were 0.712 steps and 0.634 steps, respectively, and were not significantly different from each other (V(31) = 195.00, $p = 0.203$), suggesting that representations were not biased toward one direction over the other. However, when pattern similarity to the cue environment was excluded, the Gaussian model was no longer a better fit for correctly ordered vs. shuffled data (correctly ordered data vs. shuffled null excluding cue, $p = 0.848$, $R^2 = -0.006$; Fig. 5a). This indicates that the significance of the Gaussian in visual cortex was driven by the cue environment but not by graded similarity to nearby environments in the sequence.

Turning to representations in hippocampus, the asymmetric Gaussian once again provided better fits to the correctly ordered vs. shuffled pattern similarity values when the cue environment was included (correctly ordered data vs. shuffled null including cue, $p = 0.029$, $R^2 = 0.032$). In the hippocampus, similar to visual cortex, the amplitude of the Gaussian fit was significantly higher than the different-map baseline (mean = 0.0185, standard deviation = 0.030; t(31) = 2.959, $p = 0.005$; Fig. 5b) and the asymptote was significantly lower than the different-map baseline (mean = −0.007, standard deviation = 0.014; t(31) = −3.476, $p = 0.001$; Fig. 5b). Thus, hippocampus represented the cue environment while it was on the screen and suppressed environments further away, relative to the different-map baseline. The backward and forward widths (σ) were 2.313 steps and 1.782 steps, respectively, and were again not significantly different from each other (V(31) = 231.00, $p = 0.548$), suggesting that the hippocampus did not differentially represent the forward vs backward directions.

Critically, even when the cue environment was excluded, the Gaussian was still a better fit to the correctly ordered vs. shuffled data (correctly ordered data vs. shuffled data excluding cue, $p = 0.023$, $R^2 = 0.028$; Fig. 5b). This indicates that the hippocampal Gaussian fit shows significantly graded similarity to nearby environments in the past and future, in addition to representing the cued environment. In contrast to hippocampus and visual cortex, the Gaussian fit was not significantly better for the correctly ordered data vs. the shuffled null (including the cue) in the insula ($p = 0.139$, Supplementary Fig. 1).

We then statistically compared the width parameters in hippocampus and visual cortex, to determine whether hippocampus represented more distant environments surrounding the cue and whether this differed in the forward vs. backward direction. The width

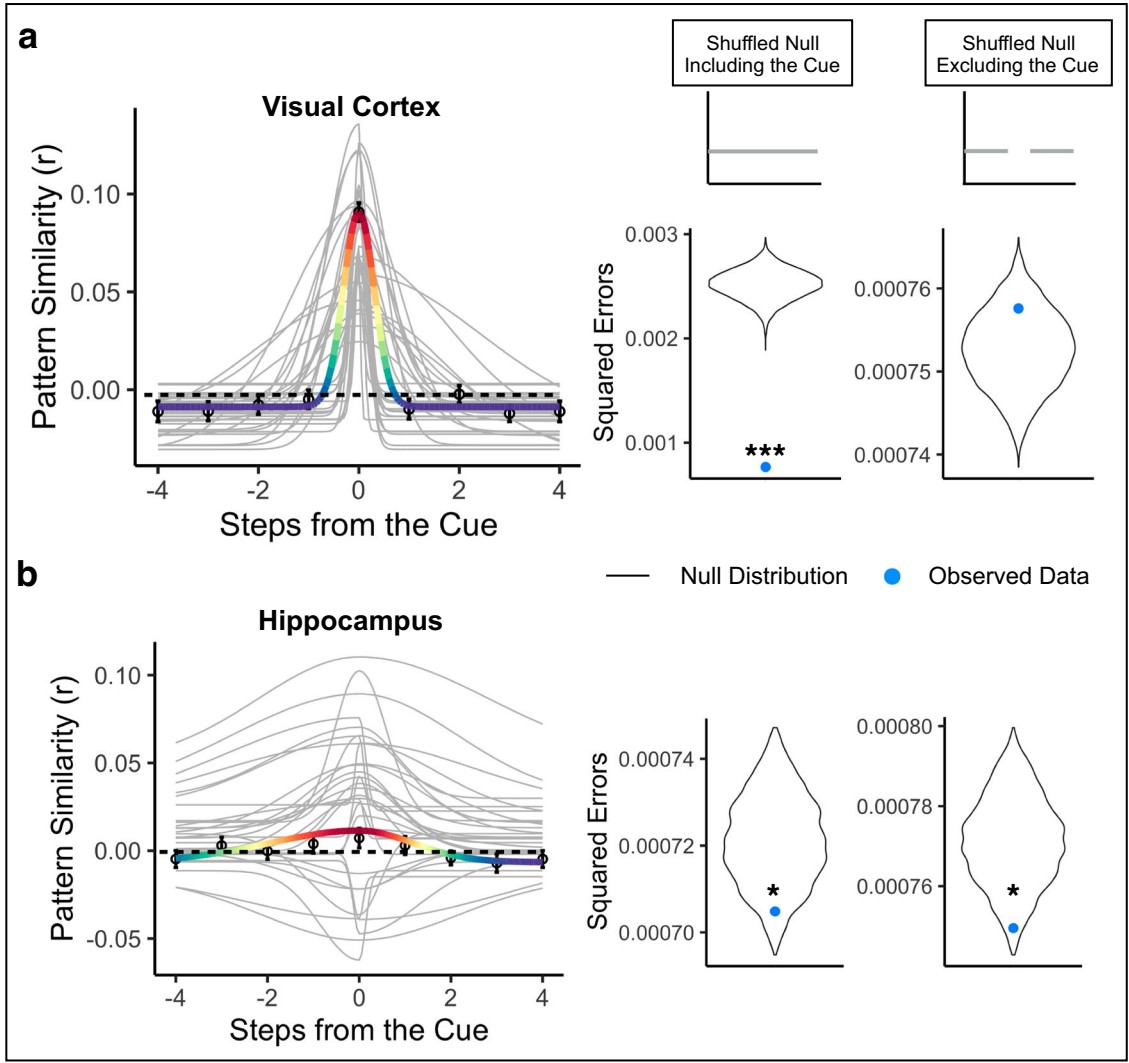

**Fig. 5 | Bidirectional and graded representations of temporal structure in hippocampus but not early visual cortex. a** Gaussian curve in visual cortex for the cued path order. Visual cortex strongly represented the cue environment while it was on the screen (above-baseline amplitude) and did not represent nearby environments (narrow forward and backward widths (σ)), instead showing suppression of environments other than the cue (below-baseline asymptote). The visual cortex Gaussian fit was significantly better than the shuffled null including the cue, but not the shuffled null excluding the cue. *N* = 32 participants. Permutation test (10,000 shuffles) comparing observed data to shuffled null including and excluding cue, two-tailed, *** *p* < 0.001 **b** Gaussian curve in the hippocampus for the cued path order. The hippocampus represented the cue environment while it was on the screen (above-baseline amplitude), represented nearby environments in a

graded manner, in both the forward and backward direction (wide forward and backward widths (σ)), and suppressed environments that were furthest away (below-baseline asymptote). The hippocampus Gaussian fit was significantly better than both the shuffled null including and excluding the cue. Points indicate average pattern similarity at each step from the cue and error bars indicate standard error of the mean. Colored line indicates the average Gaussian fit across participants, with the red end of the rainbow scale indicating higher pattern similarity and the purple end indicating lower pattern similarity, applied to values in each brain region separately. Gray lines indicate each participant's Gaussian curve. *N* = 32 participants. Permutation test (10,000 shuffles) comparing observed data to shuffled null including and excluding cue, two-tailed, Left: * *p* = 0.029, Right: * *p* = 0.023.

of the Gaussian in hippocampus was significantly wider than that in visual cortex (beta = 1.379, 95% CI = [0.559, 2.199], *p* = 0.0023), indicating that hippocampus represented more distant environments. There was no effect of direction (beta = −0.246, 95% CI = [−0.935, 0.443], *p* = 0.489), nor a region by direction interaction (beta = −0.569, 95% CI = [−1.746, 0.606], *p* = 0.346). Further, the widths were significantly larger in hippocampus than visual cortex when separately examining only the forward width (beta = 1.094, 95% CI = [0.111, 2.077], *p* = 0.033) and only the backward width (beta = 1.664, 95% CI = [0.621, 2.707], *p* = 0.003). Together, this suggests that hippocampus had further reaching representations of temporal structure than visual cortex. However, representations were not biased toward the forward, compared to the backward, direction in either region nor were there differential directional biases across regions.

Next, we determined whether the gradedness of sequence representations was specific to the order of the cued path or if it was also present for the uncued path, which contains the same environments but in a different order (see Methods). To investigate this, we repeated our Gaussian analysis with pattern similarity values arranged along the order of the uncued path. If the hippocampus represents the order of surrounding environments in a context-dependent manner, we may not observe statistically reliable Gaussian fits for the uncued path. To test for graded representations of the past and future along the uncued path, we compared the Guassian fit to the shuffled null excluding the cue environment (because the cue environment was the same for both cued and uncued paths). We also repeated this analysis in visual cortex for completeness, but because visual cortex showed no evidence for graded representations of the

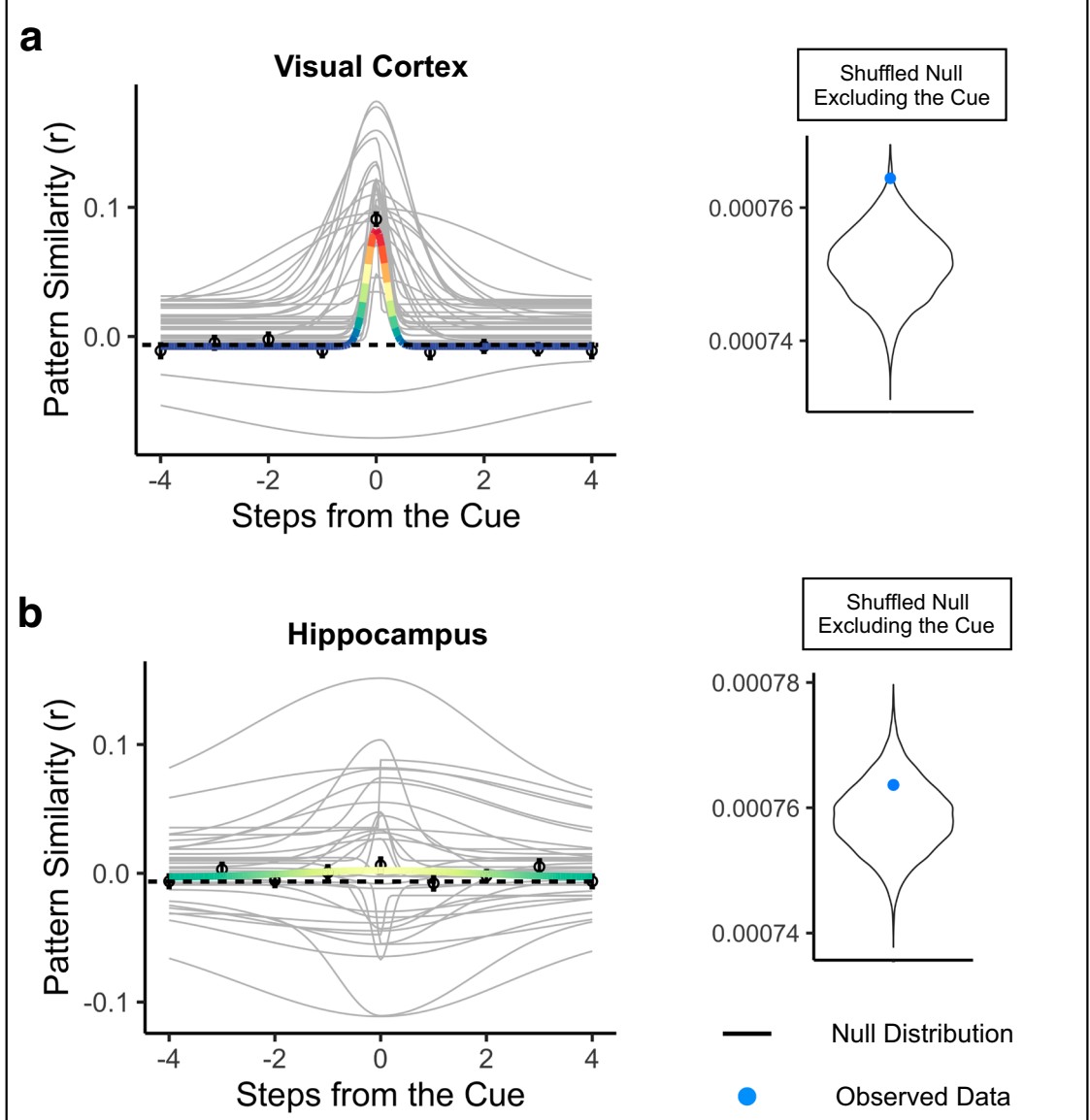

**Fig. 6 | Representations of bidirectional temporal structure are context-specific.** The Gaussian curve for the uncued path order was not significantly better than the shuffled null excluding the cue environment in either visual cortex (**a**) or hippocampus (**b**). Points indicate average pattern similarity at each step from the cue and error bars indicate standard error of the mean. Colored line indicates the average Gaussian fit across participants for the uncued path order, with the rainbow scale indicating pattern similarity values, scaled separately for each brain region along the same scale as used for the cued path Gaussian fit (red end = higher pattern similarity, purple end = lower pattern similarity). Gray lines indicate each participant's Gaussian curve. $N = 32$ participants.

past and future for the cued path, we also expected it to show no gradedness for the uncued path.

As expected, the Gaussian fit in visual cortex for the uncued path was not better than the fit for the shuffled null excluding the cue environment ($p = 0.993$, $R^2 = -0.0173$; Fig. 6a), and the Gaussian fit was not significantly different for the cued path vs the uncued path ($p = 0.132$), indicating that visual cortex did not represent surrounding environments on either the cued or uncued path. In the hippocampus, the Gaussian fit for the uncued path was not significantly better than the fit for the shuffled null excluding the cue environment ($p = 0.915$, $R^2 = -0.012$, Fig. 6b). Further, the-Gaussian fit was significantly better for the order of the cued path compared to the uncued path ($p = 0.005$). Together, these results show that the graded similarity to nearby environments in the past and future in hippocampus was specific to the order of the cued path.

## Hippocampal suppression of environment representations predicts response time costs

We next sought to examine whether neural representations of temporal structure were related to behavioral performance on the Anticipation Task. We reasoned that relative suppression of environments surrounding the cue, indicated by an asymptote that is lower than the different-map baseline, should interfere with the generation of long timescale predictions: more suppression should be associated with more response time costs for accessing future environments. To test this, we obtained the Spearman rank-order correlation between participants' Gaussian asymptote, separately for visual cortex and hippocampus, and the slope of their response times across steps into the future. We hypothesized that a lower (more suppressed) asymptote would be related to steeper response time slopes across steps into the future, indicating a larger response time cost when making judgments about further environments. We expected this relationship to

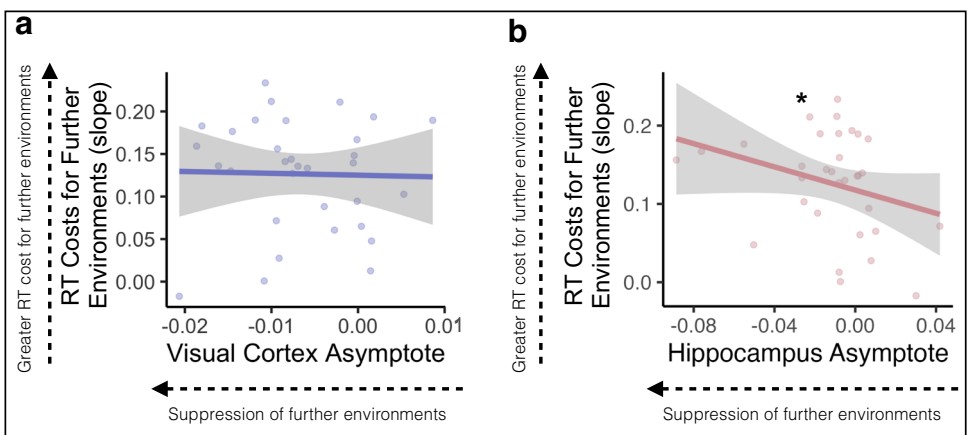

**Fig. 7 | Suppression of further environments in the hippocampus is related to response time costs.** In early visual cortex (**a**), asymptotes, indicating suppression of non-cued environments, were not related to the slope of response times across steps into the future. In the hippocampus (**b**), lower asymptotes were related to steep response time slopes, suggesting that participants were slower to respond to further environments when those environments were relatively suppressed. Lines indicate the correlation and gray error ribbons indicate 95% confidence intervals; points indicate each participant's asymptote and response time slope. Source data are provided as a Source Data file. *N* = 32 participants. Spearman's rank correlation, * *p* = 0.042.

be stronger in hippocampus vs the visual cortex, because visual cortex showed suppression of even the most nearby environments (Fig. 5a).

In visual cortex, there was no relationship between the asymptote of the Gaussian curve and the response time slope across steps into the future (rho = −0.050, *p* = 0.784, Fig. 7a). As hypothesized, in the hippocampus there was a significant negative correlation between the asymptote of the Gaussian curve and response time slope (rho = −0.362, *p* = 0.042, Fig. 7b), indicating that suppression of environments surrounding the cue was related to response time costs for anticipating further environments.

We further tested whether there was a relationship between response time slopes and width of the Gaussian fit, such that narrower widths are associated with steeper response time slopes. There was no relationship between either forward or backward width and response time slope in visual cortex (Forward Width: rho = −0.149, *p* = 0.415; Backward Width: rho = 0.071, *p* = 0.698) or hippocampus (Forward Width: rho = 0.233, *p* = 0.199; Backward Width: rho = −0.008, *p* = 0.966).

### Temporal structure is hierarchically organized within and across visual regions

We next conducted an exploratory searchlight analysis to determine which brain regions outside visual cortex and hippocampus exhibited Gaussian representations (see Methods). Our searchlight analysis revealed significant Gaussian representations across voxels in the visual system (Fig. 8a, Supplementary Fig. 2), including regions that code for scene information such as parahippocampal place area (PPA) and the retrosplenial cortex (RSC)[40]. There were no differences in backward vs forward widths (σ) in any voxel in the searchlight, suggesting bidirectional representations of temporal structure across the visual system.

Next, we conducted an exploratory analysis to test whether temporal structure was represented hierarchically across searchlights that exhibited significant Gaussian fits. We decided to focus our analysis on PPA and RSC, as prior work has shown within-region functional differences in posterior vs anterior parts of these visual regions[41–43]. Specifically, posterior aspects of these regions may play a larger role in scene perception while anterior aspects may represent scene memories. Based on these differences, we hypothesized that there may be hierarchical representations of temporal structure within PPA and RSC, with further reaching representations (as indicated by wider vs narrower widths (σ)) in successively more anterior aspects of these regions. To test for hierarchical organization of temporal structure, we

obtained the correlation for each participant between (1) the averaged forward and backward widths (σ) of the Gaussian curve in each voxel and (2) that voxel's y-coordinate, indicating its position along the posterior-anterior axis. We then tested whether these correlations were different from 0 across participants. We conducted the same analysis for the amplitude and asymptote, to determine if the representation of the cued environment and suppression of nearby environments also changed along the posterior-anterior axis.

There was a significant positive correlation between width (σ) and y-coordinate, indicating that Gaussian fits became progressively wider in progressively more anterior aspects of both RSC (t(31) = 2.638, *p* = 0.013; Fig. 8b) and PPA (t(31 = 2.424, *p* = 0.021; Fig. 8c). In PPA, the correlation between width and y-coordinate remained significant when separately examining the forward and backward widths, suggesting further reaching bidirectional representations in more anterior parts of this region (Forward Width: t(31) = 2.11, *p* = 0.043; Backward Width: t(31) = 2.046, *p* = 0.049, Supplementary Fig. 6). In RSC, the forward width successively increased with y-coordinate (t(31) = 3.931, *p* = 0.0004), but the backward width did not change with y-coordinate (t(31) = −0.256, *p* = 0.799, Supplementary Fig. 6), suggesting that hierarchically ordered representations were driven by progressive changes in the forward width in this region. There was a negative correlation between amplitude and y-coordinate in PPA (t(31) = −2.636, *p* = 0.013; Fig. 8c), but not RSC (t(31) = −0.550, *p* = 0.586; Fig. 8b). To determine whether the change in width along the posterior-anterior axis was related to the change in amplitude, such that wide Gaussian fits are associated with lower amplitudes, we ran a partial correlation between width and y-coordinate, controlling for amplitude. The negative correlation between width and y-coordinate remained significant in RSC (t(31) = 2.050, *p* = 0.049), but not PPA (t(31) = 0.08, *p* = 0.936). Thus, the amplitude and width of the Gaussian fit can vary independently in some regions, but are coupled in others. We further conducted simulations of BOLD data to confirm the lack of an inherent, systematic relationship between the width and amplitude of the Gaussian fits (see Supplementary Materials, Supplementary Fig. 4 for details). Finally, there was no correlation between asymptote and y-coordinate in either PPA (t(31) = 1.721, *p* = 0.095) or RSC (t(31) = 1.047, *p* = 0.303). Overall, this suggests a within-region hierarchical organization of representations in the visual system, such that more anterior (vs posterior) aspects of PPA and RSC represent environments that are further away in the past and future. Importantly, the partial correlation between width and y-coordinate while controlling for amplitude in RSC, as well as our simulations, suggests that further

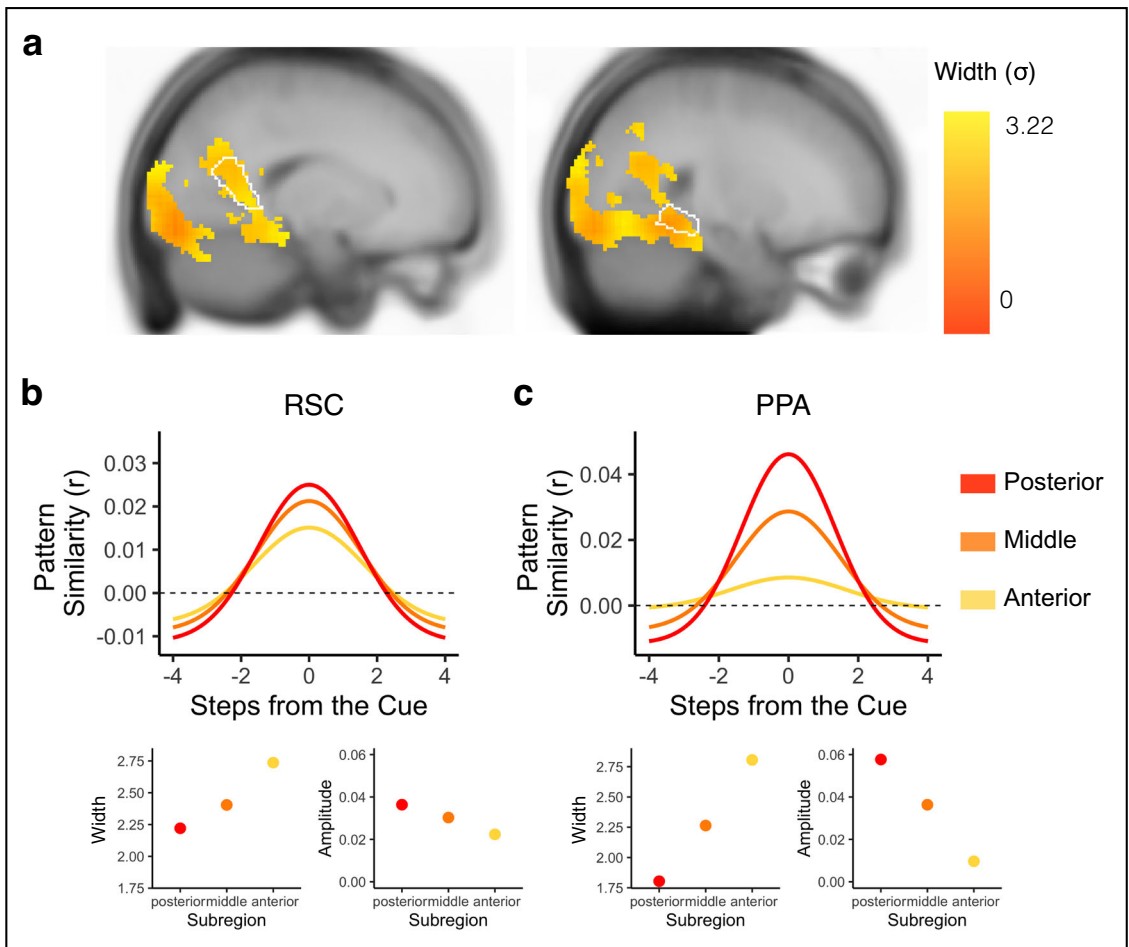

**Fig. 8 | Bidirectional and graded representations of temporal structure reveal within-region hierarchies in the visual system. a** Searchlight results revealed statistically reliable Gaussian representations in voxels across visual regions. Forward and backward widths (σ) of the Gaussian curves were hierarchically organized within visual regions (e.g., RSC and PPA), with narrow widths (indicated in red) in more posterior aspects of the region and progressively wider widths (indicated in yellow) in progressively more anterior aspects of the region. Gaussian fits of sample voxels are shown from RSC (**b**) and PPA (**c**). Voxels in progressively more anterior (indicated in yellow) compared to posterior (indicated in red) aspects of RSC and PPA had progressively wider widths (bottom left in (**b**) and (**c**)) and progressively lower amplitudes in PPA (bottom right in (**c**)) but not RSC (bottom right in (**b**)).

reaching representations are not necessarily a consequence of reduced processing of the present.

We additionally tested whether there were across-region differences in width along the visual hierarchy. We hypothesized that Gaussian fits would be successively wider in later visual cortex regions, compared to earlier ones[12,44]. We computed the average width of the Gaussian fit across all voxels separately for V1, V2, V3, V4, and PPA for each participant, and then obtained the correlation between 1) average width of the Gaussian fit in a region and 2) that region's order along the visual hierarchy. We then tested whether these correlations were different from 0 across participants. We found a significant positive correlation, indicating that successively later visual cortex regions had successively wider Gaussian fits ($t(31) = 2.136$, $p = 0.041$). This correlation remained significant when only examining the forward direction ($t(31) = 2.337$, $p = 0.026$), and when only examining the backward direction ($t(31) = 3.025$, $p = 0.005$, Supplementary Fig. 6). It also remained significant when only considering V1, V2, V3, and V4 ($t(31) = 4.322$, $p = 0.0001$). Taken together, this suggests that past and future states were represented hierarchically both within and across regions in visual cortex.

## Discussion

We examined how extended temporal structure is represented in the brain during context-dependent anticipation of future events.

Participants anticipated multiple steps into the future accurately but were slower to anticipate far vs near events. Multivoxel fMRI analyses revealed bidirectional and context-specific representations of temporal structure in hippocampus, with graded representations of environments in the forward and backward direction for the cued, but not the uncued context. Hippocampal representations of temporal structure were relevant for behavior: suppression of distant environments (relative to the different-map baseline) was linked to response time costs for anticipating further events. Beyond hippocampus, a hierarchy of temporal structure was also apparent within and across visual regions: successively more anterior aspects of PPA and RSC represented further environments into the past and future, and later regions in the visual hierarchy (e.g., V4) had further reaching representations than earlier ones (e.g., V1).

Our results build upon influential theories of prediction in the brain. Graded coding of upcoming events is consistent with successor representation models[2,24,45,46], which propose that information about future states becomes cached into the representation of the current state in a temporally discounted manner. These models have been extended to account for multiple timescales of prediction by incorporating different scales of temporal discounting[24]. In line with these theories, recent work has shown that multiple timescales of prediction are represented simultaneously in the brain, with progressively further-reaching predictions in progressively more anterior brain

regions[12] and with relatively less evidence for far away vs nearby predictions[22,47]. Converging with this past work, in our prediction task we found that patterns for nearby vs far away environments (defined in an independent task) were activated at multiple scales in a hierarchical manner across brain regions. Strikingly, although our asymmetric Gaussian analysis was designed to allow differential coding of the future vs the past, representations were not uniquely biased toward future states. Instead, the hippocampus and visual system represented temporal structure bidirectionally, with graded representations into the past and future. This finding is seemingly at odds with successor representation models[45,48], which assume that representations should be future-oriented. Taken together with prior work showing that hippocampal representations of temporal sequences[47,49,50] can be flexibly biased in either the forward or backward direction based on task demands[34], our findings suggest that representations of the past and future can exist simultaneously within the hippocampus, even though the task demands were to anticipate future states.

Why were both past and future states represented in the brain in our task? Our finding of bidirectional representations might be consistent with event segmentation and temporal context models, which suggest that a whole event is brought online during behavior, including other memories nearby in time from the same event[36]. Therefore, we believe one possible explanation for our findings is the proximity to an event boundary: if cued with the beginning of an event, individuals may exhibit future oriented predictions, but not retrospective representations. But, if cued with the middle of an event, individuals may need to bring online representations of past states to access their memory for the whole event[51]. An important distinction between our experiment and past studies of prediction is that our sequences were circular and temporally extended, whereas sequences in prior studies tended to have a clear end point (i.e. were linear instead of circular)[12,22,38,47,52] or were shorter[19]. Because our sequences were circular, environments were neither at the beginning nor at the end of the sequence, potentially explaining our finding of bidirectional rather than future-oriented representations. Thus, our findings present an intriguing avenue for future research to disentangle when bidirectional representations might be present, in line with temporal context models, rather than just forward ones, in line with successor representations.

In addition to representing nearby environments in the past and future, we also found that the hippocampus suppressed more distant environments, showing deactivation of these environments' patterns relative to an unrelated-environment baseline. To our knowledge, prior work has not looked at suppression of far away environments in a sequence during prediction of upcoming events[12,22]. It is possible that, in this previous work, suppression of further events was present but went undetected. Another possibility for such suppression is a result of the overlapping paths in the current study: individuals may have suppressed further environments on the cued path if they were coming up sooner on the uncued path. Thus, suppressing environments that were far away in the cued path but nearby in the uncued path may have been useful in avoiding confusion between the overlapping paths, and contributed to the findings observed here. Although suppressing distant environments can be beneficial for responding to imminent events, it can also lead to behavioral costs. For example, if a more distant environment appears as a probe, its representation may have to be reactivated, and this reactivation will take more time if it was initially suppressed – leading to response time costs. Indeed, hippocampal suppression of distant environments was related to response times costs for anticipating further events. This highlights a trade-off between prioritizing nearby events and being able to quickly respond to upcoming events further in the future. However, it is important to note that 1) the observed suppression was calculated relative to the different-map baseline, and therefore only indicates relative suppression compared to representations of other environments in the experiment, and 2) the observed individual-differences correlation was observed with a relatively small sample size, requiring replication in future work. With respect to the first point, environments may be suppressed relative to the different-map baseline but not suppressed compared to ongoing task-irrelevant thoughts or experiences; if so, the environments represented by the asymptote may be considered to be more *weakly* represented, rather than suppressed in an absolute sense. Future work could add additional comparisons or use alternative neuroimaging techniques to determine the level of suppression, and test whether the widths of brain regions' predictive horizons influence behavioral performance in studies specifically powered for individual differences analyses.

Representations of temporal structure extended beyond hippocampus. In an exploratory whole-brain searchlight analysis, we found representations of temporal structure across the visual system, including PPA and RSC, regions that play an important role in spatial cognition[40]. Both PPA and RSC represented the cued environment but also represented the temporal structure of surrounding environments in the sequence in both the forward and backward direction. Our findings therefore extend prior work showing that PPA responses can be modulated by temporal context[53] and prior contextual associations more generally[54–57]. Notably, our findings go beyond this prior work by showing a gradual progression of sequence coding within PPA and RSC, with progressively more anterior regions representing more of the future and past and less of the present. This is broadly consistent with prior work suggesting a posterior vs. anterior division within PPA, with posterior aspects playing a larger role in scene perception and anterior aspects playing a larger role in scene memory[41–43]. This within-region hierarchy was complemented by an across-region hierarchy, with regions higher up the visual hierarchy, such as V4 or PPA, representing further states into the past and future than regions earlier in the visual hierarchy, such as V1. Thus, we show that, within a context, visual regions may balance representations of perception and memory, gradually incorporating less information from perception and more information about learned temporal structure along a posterior to anterior hierarchy.

To investigate bidirectional, context-dependent sequence representations, we carefully manipulated overlapping sequences. Pairs of sequences contained the same environments in a different order. They were structured such that, for each environment, the environments one and two steps into the future and the past were all different in the two sequences. We found that prospective and retrospective representations in hippocampus were context-dependent, emerging only for the cued but not the uncued sequence. This dovetails with prior work showing context-sensitive representations of future events in hippocampus[38], and extends this work to graded and bidirectional sequence representations within a context. Strikingly, we observed such effects in hippocampus even with environment templates that were identified across participants. The finding of reliable hippocampal activity patterns for individual environments across participants adds to an emerging body of work demonstrating shared hippocampal representations across individuals, representations that were previously difficult to detect[58,59]. We speculate that these shared representations may include information about the perceived and/or imagined spatial layout of the scenes, given prior work linking the hippocampus to representations of attended spatial configurations[60–65]. Further, these hippocampal representations of visual scenes are consistent with a burgeoning line of work linking this region to visual representations more broadly[66–68]. To be detectable across individuals, these shared hippocampal representations are likely to be fairly coarse, similar to other across-individual representations that have been identified across the cortex[69].

Across visual cortex, the width of the Gaussian curve (when the cue environment was included) changed systematically, becoming progressively wider within early visual cortex from V1-V4 and from V1

to PPA. However, unlike hippocampus, the Gaussian curve in early visual cortex as a whole was not superior for the correctly ordered data vs. shuffled null when the cue environment was excluded. Bidirectional representations of the future and the past were more robust in higher-order visual areas (PPA, RSC) than early visual cortex. This result is seemingly in contrast to a rich literature showing predictive representations in early visual regions[11]. One possibility for why we were unable to detect predictions in early visual cortex is due to the nature of our stimuli. Past work decoding predictions from early visual regions has tended to use relatively simple stimuli such as gratings[13,14], fractals[16], or specific spatial locations[70]. In contrast, we used rich, naturalistic scenes that were experienced in immersive virtual reality from multiple viewpoints, making it unlikely that participants were generating predictions of low-level visual features tied to specific retinotopic locations. Instead, individuals in our study may have been predicting whole scenes at a relatively coarse (vs detailed) level, leading to the predictive representations observed in higher order scene-specific visual regions such as PPA and RSC[40]. We were also unable to find evidence for long-timescale predictions in the insula, unlike our prior work investigating predictive hierarchies across the brain during movie watching[12]. The relative paucity of environment-sensitive voxels in the insula may have hurt our ability to detect sequence representations in this region. Alternatively, the environment sequences used in our current study may not have engaged the insula as strongly as a continuously unfolding audiovisual movie stimulus that allowed the generation of social or emotional predictions[23].

In our study, individuals were asked to generate narratives to tie together the environments in each sequence to help learning, memory, and prediction. These stories were, however, idiosyncratic, meaning that the content of a given person's story could have influenced the conceptualization and representation of the environments and the extent to which an individual reactivated past or future states. Our analyses, however, relied on activity patterns for each environment that were obtained from across-participant templates; because the only shared information across people was the environments themselves and not the sequences or the generated stories, this allowed us to test for group-level similarities in the graded activation of environment representations. Thus, our findings show that despite the idiosyncrasies of individual stories, participants were nevertheless still representing the environments in a reliable and consistent way during the anticipation task – a way that was systematic enough that we could detect evidence for bidirectional and graded representations of the future and the past across participants. Nevertheless, it remains the case that the stories that individuals generated may have differed in their social predictions, goals, and motivations, and this may have influenced our findings. Such narrative differences across individuals may have led to differential involvement of the insula across participants[23], making it difficult for us to replicate long-timescale predictions in this region[12]. Future work could assess the participants' stories in greater detail to determine whether the stories with stronger social narratives were related to far-reaching predictions in the insula.

Broadly, it may be advantageous to represent temporal structure bidirectionally, rather than only prioritizing future states. For example, representing past states and future states could be a useful strategy when events surrounding ongoing experience differ based on context. Activating links toward past states as well as future ones may allow individuals to contextualize their current location within the sequence[51]. This possibility is consistent with our prior work showing that individuals represent sequences in terms of context-specific links between environments[39]: when an environment is cued, its associated links in both directions may be brought to mind so that the entire context is prioritized. An alternative possibility is that representing temporal structure into the past and future happens automatically: activating a particular moment within a temporally extended experience could cause activation to spread to the entire event

representation[36,71,72], which may comprise both past and future. Future work could disentangle these possibilities and further investigate the circumstances under which future and past states are simultaneously represented.

Overall, the results presented here show that temporal structure is represented bidirectionally in the hippocampus and visual system. Future and past representations of temporal structure were graded, with less evidence for further environments in both the forward and backward direction, and were organized along a posterior to anterior hierarchy within and across regions. Our results further our understanding of how temporal structure is represented in the brain: such bidirectional representations could allow integration of past events from memory alongside anticipation of future ones, which could support adaptive behavior during complex, temporally extended experiences.

## Methods

### Participants

All participants in the current study gave written, informed consent in accordance with the Institutional Review Board at Columbia University. Thirty-five healthy younger adults from the Columbia University community participated in the experiment. Participants were compensated $15 per hour for the behavioral training session and $20 per hour for the fMRI session (approximately $80 combined across both sessions). Three participants were excluded for technical issues with data collection, excessive motion (10% of TRs across all runs of the experiment marked as motion outliers by fMRIprep output), and dizziness inside the MRI scanner. Applying these exclusions resulted in a final sample of 32 participants (21 self-reported female/11 self-reported male, 19–35 years old, mean = 24.17, sd = 4.11, 13–29 years of education, mean = 16.85, sd = 3.76).

### Overview

Participants learned two sequences ("Green Path" and "Blue Path") within each of two maps (Map A and Map B; Fig. 1a). Map A and Map B contained eight distinct environments each. Within each map, the Green Path and Blue Path contained the same environments in a different order.

Participants first learned the order of the four sequences of environments by generating stories (Fig. 1b) and then experiencing the environment sequences in immersive virtual reality using an Oculus Rift (Fig. 1c). Participants returned 1 day later and completed the Anticipation Task in the MRI scanner (two runs, 32 trials per run). In the Anticipation Task, participants used their memory for the four sequences to anticipate upcoming environments. They then completed a Localizer Task to obtain multivoxel patterns of brain activity for each environment (four runs, 16 trials per run).

### Stimuli and sequence structure

Stimuli consisted of 16 3D virtual reality environments in the Unity game engine. Environments were obtained from asset collections in the Unity Asset Store. Half of the environments were indoor and half of the environments were outdoor. Using Unity, we created 2D images of each environment by rotating a virtual camera to eight different angles, 45 degrees apart. One angle was selected to be used as the cue and probe images throughout the task and the other angles were used for the panorama phase of the Localizer Task.

The 16 environments were used to form four sequences (Map A Green Path, Map A Blue Path, Map B Green Path, Map B Blue Path). Eight of the environments were assigned to Map A (i.e., the first learned set of environments) and the other eight environments were assigned to Map B (i.e., the second learned set of environments). Then, within each map (A or B), the Green Path and the Blue Path consisted of the same eight environments in a different order. The final environment in each sequence connected back to the first environment, forming a

circle. The Green and Blue Paths were designed to be as distinct as possible: for a given environment the two preceding and two succeeding environments were different across the paths (Fig. 1a). The environment-to-map assignment and the order of the environments within a sequence was randomized across participants, although the Green Path was always shuffled in the same way to create the Blue Path, as described above.

## Procedure

Participants first completed a training phase outside the MRI scanner. They returned one day later and completed a sequence refresher task outside the MRI scanner before taking part in the fMRI session. During fMRI, they completed an Anticipation Task (two runs), an Integration Task (four runs, data not included in the current manuscript) and a Localizer Task (four runs). In the training session, stimuli were presented on a computer screen with PsychoPy version 2[73] and in virtual reality with an Oculus Rift and Unity, using a mixture of custom code and OpenMaze[74]. In the fMRI session, PsychoPy was used to present the stimuli, which were projected onto a screen in the scanner bore and viewed via a mirror mounted on the head coil.

**Training phase.** In the training phase (one day before the fMRI scan), participants were instructed to learn the order of the four sequences (Map A Green, Map A Blue, Map B Green, Map B Blue; see *Stimuli and Sequence Structure)*. Participants always began by learning the Map A Green Path, because Map A was defined as the first set of environments that participants learned and the Green Path was defined as the first sequence within each map.

Participants were instructed to learn the sequences by generating a story to link the environments in order. They first saw 2D renderings of all the environments in the Map A Green Path order displayed on a computer screen. They were told to generate a detailed story to link the environments in order, and that the final environment should loop back to the first environment in the sequence to create a circle. Participants indicated that they were finished generating a story by pressing a button. Then, they were shown the sequence as pairs of adjacent environments with an empty text entry box displayed underneath (e.g., environments #1 and #2, then environments #2 and #3, etc). Participants were told to write down the story that they had generated (Fig. 1b; see Supplementary Material for story examples). Participants were given unlimited time to generate and write down their story. Once they had finished, participants verbally repeated the story back to the experimenter.

Following story generation, participants then experienced the Map A Green Path in virtual reality using an Oculus Rift (Fig. 1c). Participants were initially placed in the first environment in the sequence. After five seconds, a floating green sphere and blue sphere appeared in a random location within reaching distance of the participant. Participants were told that touching the spheres would teleport them to the next environment in the correspondingly colored sequence: they were told to touch the green sphere on the Green Path and the blue sphere on the Blue Path. After being teleported to the next environment in the corresponding sequence, participants were again given five seconds to explore the environment before the spheres would appear. After 20% of trials ("test trials"), instead of teleporting to the next environment in the sequence, participants were teleported to a black environment in which they were shown two images of upcoming environments and were told to indicate which of those two environments was coming up sooner in the sequence they were currently "traversing", relative to the preceding environment. Participants had ten seconds to respond using the Y and B buttons on the left and right Oculus Rift controllers. They were given feedback about whether their answer was correct or incorrect. As participants were exploring the environments in virtual reality, they were also told to rehearse their stories

to ensure the sequence was learned. Participants rehearsed the Map A Green Path sequence in virtual reality three times following this procedure.

Participants then repeated the exact same procedure, but learned the Map A Blue Path, which consisted of the same environments as the Map A Green Path in a different order. Participants were told to make their Blue Path story distinct from their Green Path story to avoid confusing the two paths. They then followed the same virtual reality procedure as noted above, but were instructed to touch the blue spheres instead of the green spheres to teleport between environments.

Following Map A Green and Blue Path learning, participants were exposed to each sequence three more times (including test trials) in virtual reality in an interleaved fashion (i.e., one presentation of Green Path then one presentation of Blue path, repeated three times). Participants then recalled the order of the Map A Green and Blue Paths. The above procedure was then repeated for the Map B Green and Blue Paths. In total, the training phase took between one and a half and two hours to complete. All participants performed at ceiling by the end of the training phase.

**Sequence refresher task.** Participants returned one day later. Before the fMRI scan, they completed a sequence refresher task to ensure they maintained memory for all four sequences learned during the Training Phase. Participants viewed 2D renderings of all the environments from virtual reality, one at a time, in the order of each of the four sequences (Map A Green, Map A Blue, Map B Green, Map B Blue). Participants saw each sequence in order three times. In the first presentation, participants were told to verbally repeat the stories they had generated for each sequence. In the subsequent two presentations, participants were told to verbally recall the environment that came after the currently presented environment in the current sequence.

**Anticipation task.** During the fMRI scan, participants first completed the Anticipation Task, for which there were two runs with 32 trials each (Fig. 2a). On each trial in the Anticipation task, participants were cued with an environment and a path cue ("Green" or "Blue") for 3 s. This cue indicated the starting point and sequence on that trial. Participants then viewed a blank (gray) screen for a variable duration (five to nine seconds). Then, participants were presented with two images of upcoming environments and were told to judge which of the two environments was coming up sooner in the cued sequence, relative to the cue image. Participants were given three seconds to make this judgment. This relatively short response deadline was implemented to encourage participants to use the blank screen period to generate predictions along the cued path in preparation for the forced choice decision. The correct answer could be one to four steps away from the cue image. The incorrect answer could be a maximum of five steps away from the cue image. Because the sequences were circular, every environment could be used as a cue with successors up to five steps away. There was a uniformly sampled three to eight second jittered inter-trial interval (ITI), during which participants viewed a fixation cross. At the end of each run, there was a 60 second rest period during which participants viewed a blank screen.

In each run, participants were cued with every environment from Map A and B on the Blue and Green Paths (eight environments per sequence) for a total of 32 trials per run (64 trials total). In the probe phase, the correct answer was equally distributed across steps into the future (one to four). The incorrect answer was randomly sampled to be one to four steps away from the correct answer (two to five steps away from the cue). Within a run, sequences were presented in blocks (i.e., participants completed the Anticipation Task for all the environments in the Map A Green Path in one block), but the order of the cues was randomized within a block. The order of the sequence

blocks was also randomized across runs and participants. A single run of the Anticipation Task was approximately 11 min, for a total of 22 min across both runs.

**Integration task.** Following the Anticipation Task, participants completed an Integration Task, in which they were told that one of the environments in Map A was now connected to one of the environments in Map B on either the Green or Blue Path. One of the environments in Map B also connected back to Map A, creating a single integrated path encompassing all the environments in both maps. The integrated path (Green or Blue) was counterbalanced across participants, with the other path serving as a control non-integrated path. For example, if a participant learned that Map A and Map B were integrated on the Green Path, the Blue Path would be the non-integrated path. The environments that connected Map A to Map B were randomly selected, while the environments that connected Map B back to Map A were always the preceding environments in the sequence, allowing the integrated path to form a circle. Participants then completed a version of the Anticipation Task (see above) in which they anticipated upcoming environments in the non-integrated and integrated paths (four runs, 24 trials per run). The Integration Task is not analyzed in the current manuscript.

**Localizer task.** Participants then completed four runs of a Localizer Task used to obtain environment-specific patterns of brain activity across participants (Fig. 3a, see *Environment Templates*). In the Localizer Task, participants were cued with an environment from Map A or B on the screen for one second. The cue in the Localizer Task did not include a path cue (Green or Blue), allowing us to obtain a context-independent pattern of brain activity for each environment. Specifically, the lack of a context cue should disincentivize participants from consistently activating one sequence (Green or Blue path) over the other while viewing the environments – allowing us to obtain activity patterns for each environment relatively uncontaminated by associated information. Following the cue, participants saw a blank gray screen for five seconds, during which they were told to imagine being inside the environment in virtual reality. Participants then viewed images of the cued environment from different angles, 45 degrees apart, for four seconds. They were then given three seconds to rate how well their imagination matched the actual images of the environment, on a scale from one to four (one = not well, four = very well). There was a three to eight-second jittered ITI, during which participants viewed a fixation cross.

In each run, participants were cued with every environment from Map A and B for a total of 16 trials per run (64 trials total across all four runs). The order of the environments was randomized across runs and participants. A single run of the Localizer Task was approximately five and a half minutes, for a total of 22 min across all four runs.

**Behavioral Analysis.** We conducted analyses on the behavioral data in the R programming language using generalized linear and linear mixed effects models (GLMMs and LMMs, glmer and lmer functions in the *lme4* package, version 1.1.35.3[75]). For analyses that modeled multiple observations per participant, such as accuracy or response time on a given trial, models included random intercepts and slopes for all within-participant effects. All response time models examined responses on correct trials only.

To ensure that participants performed effectively during the Anticipation Task, we first tested whether accuracy during the Probe screen (see Fig. 2a) was better than chance performance (50%) using a one-sample t-test.

We next determined whether accuracy and response time differed across the Maps (A and B) and Paths (Green and Blue). To examine sequence effects, we fit separate models for accuracy (a GLMM) and response time (an LMM) as a function of Map (A = −0.5, B = 0.5), Path

(Green = −0.5, Blue = 0.5), and their interaction. We used the following R-based formulas (where "participant" indicates participant number):

$$glmer(correct \sim map*path + (1 + map*path|participant), family = ''binomial'', data)$$
(1)

$$lmer(RT \sim map*path + (1 + map*path|participant), data, subset = (correct = = 1))$$
(2)

Next, we determined whether accuracy and response time differed across steps into the future. We fit separate models for accuracy (a GLMM) and response time (an LMM) as a function of steps into the future (−0.75 = 1 step, −0.25 = 2 steps, 0.25 = 3 steps, 0.75 = 4 steps). We used the following R-based formulas (where "participant" indicates participant number):

$$glmer(correct \sim steps + (1 + steps|participant), family = ''binomial'', data)$$
(3)

$$lmer(RT \sim steps + (1 + steps|participant), data, subset = (correct = = 1)) \quad (4)$$

## MRI acquisition
Whole-brain data were acquired on a 3 Tesla Siemens Magnetom Prisma scanner equipped with a 64-channel head coil at Columbia University. Whole-brain, high-resolution (1.0 mm iso) T1 structural scans were acquired with a magnetization-prepared rapid acquisition gradient-echo sequence (MPRAGE) at the beginning of the scan session. Functional measurements were collected using a multiband echo-planar imaging (EPI) sequence (repetition time = 1.5 s, echo time = 30 ms, in-plane acceleration factor = 2, multiband acceleration factor = 3, voxel size = 2 mm iso). Sixty-nine oblique axial slices were obtained in an interleaved order. All slices were tilted approximately −20 degrees relative to the AC-PC line. There were ten functional runs in total: two runs of the Anticipation Task, four runs of an Integration Task (not analyzed here), and four runs of the Localizer Task. Field maps were collected after the final functional scan to aid registration (TR = 679 ms, TE = 4.92 ms/7.38 ms, flip angle = 60°, 69 slices, 2 mm isotropic).

## Preprocessing
Results included in this manuscript come from preprocessing performed using fMRIPrep 1.5.2 (Esteban, Markiewicz, et al. (2018)[76]; Esteban, Blair, et al. (2018)[77]; RRID:SCR_016216), which is based on Nipype 1.3.1 (Gorgolewski et al.[78]; Gorgolewski et al. (2018)[77]; RRID:SCR_002502).

**Anatomical data preprocessing.** The T1-weighted (T1w) image was corrected for intensity non-uniformity (INU) with N4BiasFieldCorrection (Tustison et al.)[79], distributed with ANTs 2.2.0 (Avants et al.[80], RRID:SCR_004757), and used as T1w-reference throughout the workflow. The T1w-reference was then skull-stripped with a Nipype implementation of the antsBrainExtraction.sh workflow (from ANTs), using OASIS30ANTs as target template. Brain tissue segmentation of cerebrospinal fluid (CSF), white-matter (WM) and gray-matter (GM) was performed on the brain-extracted T1w using fast (FSL 5.0.9, RRID:SCR_002823, Zhang, Brady, and Smith[81]). Brain surfaces were reconstructed using recon-all (FreeSurfer 6.0.1, RRID:SCR_001847, Dale et al.,[82]), and the brain mask estimated previously was refined with a custom variation of the method to reconcile ANTs-derived and FreeSurfer-derived segmentations of the cortical gray-matter of Mindboggle (RRID:SCR_002438, Klein et al.[83]). Volume-based spatial normalization to one standard space (MNI152NLin2009cAsym) was performed through nonlinear registration with antsRegistration (ANTs 2.2.0), using brain-extracted versions of both T1w reference

and the T1w template. The following template was selected for spatial normalization: ICBM 152 Nonlinear Asymmetrical template version 2009c [Fonov et al.[84], RRID:SCR_008796; TemplateFlow ID: MNI152NLin2009cAsym].

**Functional data preprocessing.** For each of the 10 BOLD runs found per subject (across all tasks and sessions), the following preprocessing was performed. First, a reference volume and its skull-stripped version were generated using a custom methodology of fMRIPrep. A deformation field to correct for susceptibility distortions was estimated based on a field map that was co-registered to the BOLD reference, using a custom workflow of fMRIPrep derived from D. Greve's epidewarp.fsl script and further improvements of HCP Pipelines (Glasser et al.)[85]. Based on the estimated susceptibility distortion, an unwarped BOLD reference was calculated for a more accurate co-registration with the anatomical reference. The BOLD reference was then co-registered to the T1w reference using bbregister (FreeSurfer) which implements boundary-based registration (Greve and Fischl 2009). Co-registration was configured with six degrees of freedom. Head-motion parameters with respect to the BOLD reference (transformation matrices, and six corresponding rotation and translation parameters) are estimated before any spatiotemporal filtering using mcflirt (FSL 5.0.9, Jenkinson et al.)[86]. The BOLD time-series, were resampled to surfaces on the following spaces: fsaverage6. The BOLD time-series (including slice-timing correction when applied) were resampled onto their original, native space by applying a single, composite transform to correct for head-motion and susceptibility distortions. These resampled BOLD time-series will be referred to as preprocessed BOLD in original space, or just preprocessed BOLD. The BOLD time-series were resampled into standard space, generating a preprocessed BOLD run in ['MNI152NLin2009cAsym'] space. First, a reference volume and its skull-stripped version were generated using a custom methodology of fMRIPrep. Several confounding time-series were calculated based on the preprocessed BOLD: framewise displacement (FD), DVARS and three region-wise global signals. FD and DVARS are calculated for each functional run, both using their implementations in Nipype (following the definitions by Power et al.)[87]. The three global signals are extracted within the CSF, the WM, and the whole-brain masks. Additionally, a set of physiological regressors were extracted to allow for component-based noise correction (CompCor, Behzadi et al.)[88]. Principal components are estimated after high-pass filtering the preprocessed BOLD time-series (using a discrete cosine filter with 128 s cut-off) for the two CompCor variants: temporal (tCompCor) and anatomical (aCompCor). tCompCor components are then calculated from the top 5% variable voxels within a mask covering the subcortical regions. This subcortical mask is obtained by heavily eroding the brain mask, which ensures it does not include cortical GM regions. For aCompCor, components are calculated within the intersection of the aforementioned mask and the union of CSF and WM masks calculated in T1w space, after their projection to the native space of each functional run (using the inverse BOLD-to-T1w transformation). Components are also calculated separately within the WM and CSF masks. For each CompCor decomposition, the k components with the largest singular values are retained, such that the retained components' time series are sufficient to explain 50 percent of variance across the nuisance mask (CSF, WM, combined, or temporal). The remaining components are dropped from consideration. The head-motion estimates calculated in the correction step were also placed within the corresponding confounds file. The confound time series derived from head motion estimates and global signals were expanded with the inclusion of temporal derivatives and quadratic terms for each (Satterthwaite et al. 2013). Frames that exceeded a threshold of 0.5 mm FD or 1.5 standardized DVARS were annotated as motion outliers. All resamplings can be performed with a single interpolation step by composing all the pertinent transformations (i.e. head-motion transform matrices,

susceptibility distortion correction when available, and co-registrations to anatomical and output spaces). Gridded (volumetric) resamplings were performed using antsApplyTransforms (ANTs), configured with Lanczos interpolation to minimize the smoothing effects of other kernels (Lanczos)[89]. Non-gridded (surface) resamplings were performed using mri_vol2surf (FreeSurfer).

Many internal operations of fMRIPrep use Nilearn 0.5.2 (Abraham et al.[90], RRID:SCR_001362), mostly within the functional processing workflow. For more details of the pipeline, see the section corresponding to workflows in fMRIPrep's documentation.

**Copyright waiver.** The above boilerplate text was automatically generated by fMRIPrep with the express intention that users should copy and paste this text into their manuscripts unchanged. It is released under the CC0 license.

## fMRI analysis

After preprocessing, all fMRI analyses were performed in Python and R. Pattern similarity analyses were performed using custom code in Python 3. Statistical analysis comparing pattern similarity values across conditions, correlations between fMRI results and behavior, and visualizations were performed using custom code in R version 4.4.0.

## Localizer task analyses

We conducted GLMs predicting whole-brain univariate BOLD activity from task and nuisance regressors from the Localizer Task using custom scripts in Python. For each participant, we first concatenated the fMRI data across runs of the Localizer Task and modeled BOLD activity for each environment (1 to 16) with a boxcar regressor combined across the cue, blank screen, and panorama periods. We also included nuisance regressors in the same model (translation and rotation along the X, Y, and Z axes and their derivatives, motion outliers as determined by fMRIprep, CSF, white matter, framewise displacement, and discrete cosine-basis regressors for periods up to 125 s).

We next looked across the whole brain for voxels that showed reliable, environment-specific patterns of activity during the Localizer Task. We used an approach that identifies voxels that respond reliably to different conditions across runs of an experiment[91], here measuring reliability across different participants[92]. For each voxel, we obtained a 16-element vector of beta weights from the whole-brain GLM, reflecting the beta weight for each of the 16 environments for each participant (e.g., Participant #1 or P1). Next, we obtained the Pearson correlation (r) between each participant's 16-element vector in each voxel and the averaged 16-element vector from the remaining participants (e.g., P2-P32). Finally, we calculated an environment reliability score by averaging the r values across all iterations of the held-out participant (Fig. 3b). Voxels that had an *r* value of 0.1 or greater ("environment-reliable voxels") were then included in subsequent steps (Fig. 3c). We selected 0.1 as our cutoff following the threshold used in prior work to detect reliable across-participant representations[69,93]. This threshold resulted in reasonable spatial coverage while maintaining voxel reliability, including in our a priori regions of interest[91].

## Conjunction ROI definition

Three a priori regions of interest (ROIs) were defined using environment-reliable voxels (see above) within anatomical or functional areas of interest. The V1-4 ROI was obtained from the probabilistic human visual cortex atlas provided in Wang et al.[94] (threshold: $p = 0.50$). The hippocampus and insular cortex ROIs were both defined from the Harvard-Oxford probabilistic atlas in FSL (threshold: $p = 0.50$). We resampled the three ROIs onto the same MNI grid as the functional data (MNI152NLin2009cAsym), and then intersected them with our map of environment-reliable voxels ($r > 0.1$, see description above) to create conjunction ROIs in visual cortex, hippocampus, and

insula (Fig. 3d). There were 2931 environment-reliable voxels in visual cortex, 156 environment-reliable voxels in hippocampus, and 84 environment-reliable voxels in insula. We also included PPA and RSC as exploratory ROIs based on our searchlight analyses (see below). There were 631 voxels in PPA and 888 in RSC.

We then obtained the spatial pattern of activity across voxels in each conjunction ROI by averaging the pattern of activity for that environment across all participants. These across-participant environment-specific activity patterns were then used as "template" activity patterns for subsequent analyses. Because the Localizer Task did not include a path cue (Green or Blue), participants should not have been differentially and consistently activating one path as they viewed each environment; thus, the pattern of activity obtained for each environment should be context-independent and should not prioritize past or upcoming environments in a given context. Importantly, this approach yielded the expected result of producing ROIs with environment-specific patterns of activity: activity patterns for the same environment were more correlated than activity patterns for different environments within each conjunction ROI (Fig. 3d). These environment-specific patterns (hereafter referred to as "environment templates") are a necessary precursor for investigating prediction along each sequence (see below).

### Anticipation task analyses

We conducted GLMs predicting whole-brain univariate BOLD activity from behavioral and nuisance regressors from the Anticipation Task using Python. For each participant, we modeled BOLD activity concatenated across both runs of the Anticipation Task with separate regressors for the cue, blank screen, and probe periods for each environment in Map A and B (1 to 16) and for each path (Green Path and Blue Path). This resulted in a total of 32 task regressors for each phase (cue, blank screen, probe) of the Anticipation Task (16 environments across Map A and Map B, with each environment modeled separately for the Green Path and the Blue Path). We also included nuisance regressors in the same model (the same as those used for the Localizer Task Analyses). For all subsequent analyses (except the searchlight analysis), the resulting beta weights were examined within our conjunction ROIs.

**Asymmetrical gaussian analysis.** To assess evidence for multivoxel representations of temporal structure, we obtained the correlation between (1) a given participant's (e.g., P1) cue screen activity pattern for each trial type (a given environment cued on a given path) in the Anticipation Task and (2) the remaining participants' (e.g., P2-P32) averaged patterns of activity for each of the environment templates from the corresponding map (Fig. 4a). For example, if Participant 1 was cued with environment one from Map A on the Green Path, we obtained the correlation between: (1) Participant 1's cue screen activity pattern for that environment and path and (2) each environment template (averaged across Participants 2-32) from Map A. This yielded eight separate pattern similarity values (because there are eight environments per map) for each trial type (a given environment cued on a given path). Because each participant had a different ordering of environments, these across-participant templates cannot contain any reliable information about the successors or predecessors of a given environment for a given participant. We then ordered the resulting pattern similarity values according to the cued Map and Path (in this example, the Map A Green Path order), with the cue in the center (Fig. 4a). Thus, successors following the cue would be to the right of the center and predecessors would be to the left of the center. Because, in an eight-environment map, four steps away is an equal distance from the cue in both the past and the future, we included the pattern similarity value four steps away from the cue in both the forward and the backward direction. We also obtained the correlation between (1) a given participant's (e.g., P1) cue screen activity pattern

for each trial type (a given environment cued on a given path) in the Anticipation Task and (2) the remaining participants' (e.g., P2-P32) averaged patterns of activity across all environment templates in the different map (in this example, Map B). This single value served as the different-map baseline.

Next, we fit an asymmetrical Gaussian curve to the resulting pattern similarity values arranged on the order of the cued path. We chose to use a Gaussian curve because we hypothesized that brain regions would represent upcoming (or past) environments in a graded manner, with stronger representations for nearby environments[24]. The asymmetrical Gaussian has four parameters: amplitude, asymptote, and forward and backward widths (σ) (Fig. 4b). The amplitude controls the height of the peak of the Gaussian curve, and indicates the extent to which a brain region is representing the cue environment presented on the screen. The asymptote controls the vertical shift of the Gaussian curve. This asymptote was compared to a baseline value consisting of pattern similarity between the cue and environments from the other map (different-map baseline), because these other-map environments are never accurate predictions from this cue. If the asymptote is lower than this baseline, that would reflect relative suppression of some environment templates in the current map (relative to environments that are currently irrelevant). The widths (σ) control the slope of the fall off from the amplitude to the asymptote. Wider Gaussians indicate activation of environment patterns further away from the cue. Because we fit an asymmetrical Gaussian, we obtained different widths in the forward and backward direction; this allows brain regions to potentially represent more environments in one direction (e.g., upcoming environments) than another (e.g., past environments). This in turn enables us to detect if some brain areas anticipate the future but do not represent the past. We constrained the widths to be a maximum of 10 and applied L2 regularization to the amplitude and intercept (with strength = 0.01) to ensure the model did not return uninterpretable parameter values.

To test whether the parameters of the Gaussian curve were consistent across participants, we fit the asymmetrical Gaussian curve on all but one participant's data (e.g. P2-32) and then measured the sum of squared errors (observed vs. predicted pattern similarity values) when using this curve to predict the held-out participant's data (e.g. P1). We repeated this procedure for each choice of held-out participant to obtain an average error value. We conducted two tests. First, we compared the Gaussian fit to correctly ordered data and the Gaussian fit to shuffled pattern similarity values, including pattern similarity to the cue environment ("shuffled null including cue"). Second, we conducted the same comparison but removed pattern similarity to the cue environment from both the correctly ordered and shuffled data ("shuffled null excluding cue"). In both cases, we fit the Gaussian to shuffled pattern similarity values 10000 times and obtained the goodness of fit each time to create a null distribution. The goodness of fit for the correctly ordered data vs. null distributions were then compared. If, for a given brain region, the Gaussian model provides a better fit to the correctly ordered vs. shuffled data both when the cue is included and when it is excluded, that would indicate that the brain region's representations of both the cue and graded representations of nearby environments contribute to a significant Gaussian fit. If, however, the Gaussian model provides a better fit to the correctly ordered vs. shuffled data only when the cue is included and not when it is excluded, that would indicate that its Gaussian fit was driven by its representation of the cue environment without any evidence for graded representations of the past or future.

We obtained a p-value by calculating the fraction of 10,000 shuffles that produced Gaussian fits with lower error than the fit for the correctly ordered data. We only proceeded with tests of the model parameters (i.e., amplitude, asymptote, width) if the Gaussian fit for the correctly ordered data vs. the shuffled null was statistically significant. We also calculated $R^2$ as 1 minus the sum of squared errors of

the Gaussian fit for the correctly ordered data divided by the mean sum of squared errors of the Gaussian fits for the shuffled data. $R^2$ values greater than 0 indicate smaller squared errors for the correctly ordered data than the shuffled data.

We also performed statistical tests on the parameters of the correctly ordered Gaussians. We compared the amplitude and asymptote to a baseline which consisted of the pattern similarity values between the cue screen activity pattern and the activity pattern templates for the eight environments that were not in the cued map (different-map baseline). We tested whether the amplitude was significantly above the different-map baseline, and whether the asymptote was significantly below the different-map baseline, using one-sample t-tests across participants. A brain region with an asymptote that is significantly lower than the different-map baseline would be interpreted as having relatively suppressed representations of those environments, compared to our baseline condition. We also tested whether the widths (σ) differed by brain region (visual cortex vs hippocampus), direction (forward vs backward), and their interaction using the following R-based formula (where "participant" indicates participant number):

$$\text{lmer(width} \sim \text{region*direction} + (1|\text{participant}), \text{data}) \quad (5)$$

Finally, to test whether the Gaussian representations were context dependent, we repeated the same Gaussian fitting procedure as above but with the pattern similarity values arranged on the order of the uncued path. Importantly, we only tested whether the Gaussian fit for the uncued path was significantly better than the shuffled null excluding the cue because 1) the cue environment was the same across paths and 2) this comparison specifically tests evidence for graded representations of the past and future. We also directly assessed whether the cued order Gaussian fit was significantly better than the uncued order Gaussian fit. We computed the difference between the sum of squared errors for 1) the correctly ordered data along the cued path vs the correctly ordered data along the uncued path, excluding the cue environment from both and 2) the same cued vs uncued comparison for each of the shuffled nulls. We then tested whether the difference in fit for the cued vs uncued path in the real (unshuffled) data was larger than the difference in fit for the cued vs uncued path in the shuffled nulls. This allowed us to statistically test whether there was more evidence for graded representations of the past and future along the cued vs. uncued path in the absence of similarity to the cue environment.

**Searchlight**. We conducted a whole-brain searchlight analysis with custom Python code to test whether brain regions beyond our ROIs represented temporal structure in the hypothesized asymmetrical Gaussian format. We looked for significant Gaussian representations in cubes with a side length of 7 voxels, moved throughout the whole brain volume with a step size of 2 voxels. We included only environment-reliable voxels within each cube and only proceeded with the analysis of a cube if it contained at least 64 environment-reliable voxels. The parameters of fitted Gaussians within each searchlight, along with the goodness of fit, were assigned to each voxel in the searchlight. For voxels that were included in more than one searchlight, the final Gaussian parameters and goodness-of-fit were obtained by averaging the results across all the searchlights in which the voxel was included.

To determine which voxels exhibited significant Gaussian representations across participants, we first obtained a measure of goodness of fit by dividing the squared errors of the correctly ordered Gaussian by the average of the squared errors of the permuted Gaussians and then subtracting the resulting value from 1 for each voxel included in the searchlight for each participant. Numbers above 0 indicate better fits to the correctly ordered vs permuted

data. We then statistically tested whether the goodness of fit values were greater than 0 in each voxel, using FSL's randomize function with threshold free cluster enhancement, which generates null distributions using 10,000 permutations and performs a one-sample t-test while enhancing clusters of significant voxels[95]. We then corrected for multiple comparisons using the family-wise error rate correction ($p < 0.05$).

To determine whether Gaussian widths (σ) were organized hierarchically within brain regions, we first averaged the forward and backward widths for each voxel for each participant. We opted to compute the average of the forward and backward widths because we did not find evidence for directional asymmetry in any brain region, in both our ROI and searchlight analyses. Next, we determined whether the averaged widths became increasingly wider in more anterior, compared to posterior, voxels in the parahippocampal place area (PPA) and the retrosplenial cortex (RSC). We created PPA and RSC ROIs using pre-defined anatomical ROIs[96], which we then resampled onto the same MNI grid as the functional data (MNI152NLin2009cAsym) and intersected with our map of environment-reliable voxels (r > 0.1). We chose PPA and RSC because (1) the searchlight revealed significant Gaussian representations in the majority of voxels in these regions and (2) they have previously been implicated in both scene perception and memory[40,42]. In each region, for each participant, we obtained the Spearman rank-order correlation between the averaged forward and backward widths (σ) and the y coordinate (indicating a voxel's position on the posterior-anterior axis) across voxels. We then determined whether the correlation was significant at the group level by comparing the participant-specific r values to 0 using a one-sample t-test. A significantly positive r value would indicate that Gaussian curves become increasingly wider in successively anterior aspects of regions. Additionally, we repeated the same analysis as above assessing the correlation between width and y-coordinate while controlling for the amplitude of the Gaussian fit.

Finally, we determined whether there was a timescale hierarchy across regions in visual cortex, with average widths becoming increasingly wider in later visual cortex regions. We first computed the average width across voxels separately for V1, V2, V3, V4, and PPA for each participant. We then ordered each visual region based on their location along the visual hierarchy (V1-V4, then PPA). For each participant, we then obtained the Spearman rank-order correlation between the average forward and backward widths (σ) and the region's location along the visual hierarchy. We determined whether the correlation was significant at the group level by comparing the participant-specific r values to 0 using a one-sample t-test. A significantly positive r value would indicate that Gaussian curves become increasingly wider in successively later visual cortex regions.

**Relationship to behavior**. We determined whether an individual's asymptote from their Gaussian model, indicating suppression of environments not captured by the Gaussian's width, was related to response time costs for further environments. We also tested whether an individual's forward and backward width from their Gaussian model was related to response time costs. Response time costs were quantified with participant-specific regressions that predicted response time as a function of steps into the future. We then performed an individual differences analysis by obtaining the Spearman rank-order correlation between participants' response time costs and their asymptotes in (1) the hippocampus and (2) the visual cortex. We repeated this analysis for the forward and backward width instead of the asymptote.

**Reporting summary**
Further information on research design is available in the Nature Portfolio Reporting Summary linked to this article.

## Data availability

The data presented in figures in this study are provided in the source data file. fMRI data used in this study are available in the openneuro database under accession code https://openneuro.org/datasets/ds005125. Source data are provided with this paper.

## Code availability

Code to reproduce all figures and statistical analyses in the manuscript and supplement is available at https://github.com/hannahtarder-stoll/predNav.

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

## Acknowledgements

This work was funded by a National Institutes of Health Research Project Grant (R01EY034436) and a Zuckerman Institute Seed Grant for MR Studies (CU-ZI-MR-S-0016) to M.A. and C.B. We would like to thank the Alyssano Group for helpful advice on this project.

## Author contributions

Conceptualization, H.T.S., C.B., and M.A.; Methodology, H.T.S., C.B., and M.A.; Software, H.T.S.; Formal Analysis, H.T.S.; Investigation, H.T.S.; Writing – Original Draft, H.T.S.; Writing – Review & Editing, C.B. and M.A.; Visualization, H.T.S., C.B., and M.A.; Supervision, C.B. and M.A.; Funding Acquisition, C.B. and M.A.

## Competing interests

The authors declare no competing interests.
