## [Peer Review File · Nature Communications]

The brain hierarchically represents the past and future during
multistep anticipationReviewer #1 (Remarks to the Author):

Tarder-Scholl and colleagues investigate how the brain represents the temporal order of sequences of environments, focusing on the hippocampus and visual cortex. They find that hippocampal representations extend further backward and forward in time than visual cortical representations do, and that forward representations may be relevant for behavioural performance in an anticipation task. The experimental manipulation is elegant, the immersive VR training of the sequences is naturalistic, and the analyses are novel. However, I have some serious concerns about the analyses and the interpretation of the results, which I detail below.

Major

1. It does not seem like a fitted asymmetrical Gaussian curve is the best model for the data. The forward-width of the Gaussian for the visual cortex (Fig.4c) seems to descend more sharply than the data for any of the participants and the mean over participants do. Further model comparisons (other than comparing to the null model) would be more convincing. In particular, the Gaussian for hippocampus (Fig.4d) could be compared to the flat baseline, and error bars should be added.
2. Does the analysis in Figure 7b suggest that the uncued sequences were replayed just as much as the cued ones during the anticipation task in visual cortex and hippocampus? So would the results in Figure 4 look the same if the uncued rather than the cued path were used for the analysis? (The concern is somewhat alleviated for visual cortex by the blank screen analysis in Figure 7c, but not for hippocampus.) This may be a misunderstanding, but if not, that seems a very important point that should be emphasised more strongly from the start. At a minimum, the results in Figure 4 should be shown for the uncued path as well. Also, with vast literature showing that the hippocampus holds contextual information and may be involved in pattern completion processes, why not look at the blank screen period for the hippocampus as well?
3. The interpretation of the Gaussian curve results is unclear. The Gaussian for visual cortex shows that steps -1 and +1 have below zero pattern similarity (Fig.4c), so can the Gaussian really be said have a meaningfully non-zero width, or is just this an artefact of the analysis?
4. More justification is needed for the claim on p. 11: "if the asymptote is lower than baseline (defined as pattern similarity between the cue and all environments from the other map, henceforth referred to as different-map baseline), this suggests that these environments are suppressed." Given that pattern similarity for all 'different' environments (Fig.3d) seems to be below baseline, it seems more likely to just be an artefact of the analysis that unrelated environments have below zero similarity. And, if more distant environments are suppressed, how are they nonetheless still activated afterwards, to allow the participant to respond correctly?
5. The authors relate the asymptote to behaviour (Fig.5), which is an interesting analysis. It might be interesting (and more intuitive) to do the same for the backward and (especially) forward widths of the Gaussians, i.e., the more forward-looking the hippocampus is, the better people do at the anticipation task. Of course, correlating multiple variables to individual differences in behaviour poses a multiple comparison problem, especially with a small sample

(by individual difference analysis standards). Perhaps this is a limitation that should be mentioned in either case: brain-behaviour correlations with small samples are notoriously tricky.

Minor

1. When participants made errors or had longer response times in the anticipation task, were these more-often-than-not congruent with the order of environments in the uncued path?
2. Were representations of upcoming environments weaker on trials in which participants made errors in the anticipation task? And if so, was there any evidence for representations of the uncued path, especially on those error trials where participants' response was congruent with the order of the scenes in that alternative path? Of course there may not be sufficient numbers of error trials to do these analyses, but it would be great if you could.
3. It is unclear why the authors have only done across-subject analyses. I appreciate the argument on p.9 regarding randomised sequences for individual participants, but this concern might already be sufficiently taken care of by having different (blue and green) paths of the same scenes within participants. Within participant analyses might be expected to be more sensitive, and given that the hippocampus representations vary quite a lot across subjects (Fig.4d), it would be interesting to see the results when comparing within-subjects.
4. You could make it clearer that results in Fig. 4c-e combines the cued and uncued paths, rather than only the cued path.

Reviewer #2 (Remarks to the Author):

In this paper, Tarder-Stoll et al. examine the neural representations that are activated when people are anticipating spatial environments that unfold in a typical sequence. In regions including the hippocampus and insular, they observed neural signatures of environments that occurred both earlier and later elements than the current sequence elements, while participants prepared to make a judgment about upcoming sequence elements. They also observed an interaction between the position of brain regions and the sequence activations, with more anterior brain regions activating patterns associated with more distant sequence elements.

The main strengths of this paper are: (i) that it uses a sophisticated and careful experimental design that enables the measurement of both forward and backward sequence representations without horizon effects, while allowing for context cues; and (ii) the paper address a question of current interest in memory research, as sequence replay and prediction are currently of great interest in the study of human memory. The manuscript is well-written and well-illustrated, and the Discussion does a good job situating these findings in the broader literature.

The main area for improving this manuscript will be in methodologically confirming (via further analysis and/or simulations) that some of the key claims (i.e. regional gradients in the properties of the Gaussian-fits) cannot be explained by other forms of spatial variation, such as variation in signal-to-noise or variation in the effect of preprocessing/normalization prior to computing the

pattern similarity.

MAIN POINTS

** The manuscript should demonstrate that the Gaussian model is actually a good model of the PatternSimilarity-by-Steps data.(Figure 3C). The fit could be measured using, e.g. the cross-validated R^2 of the Gaussian model, but any method is fine as long as it quantifies overall performance in held-out data. It is also crucial to check whether the “goodness” of the Gaussian model fit varies by ROI, and to ensure that this goodness-of-fit is not confounded with other ROI-based variation in the results.

** The manuscript should more carefully consider the properties of the stories that participants generated during encoding, and how they might affect the subsequent reactivation effects. Story generation is described as: “Participants were instructed to learn the sequences by generating a story to link the environments in order.” There are only two example stories shown in the Supplementary Information but they are quite different. The first example is quite passive, with a person simply being transported from one place to the next, but the second story contains a much richer motivational structure. Indeed, in the second story the participant meets a character (a man in a tophat) who actually accompanies him to subsequent rooms and then later betrays the participant. Could such narrative differences induce differences in the inter-environment reactivation patterns? For example, when the participant enters the pub and meets the monocle character, will they not be reminded of their ultimate betrayal back into prison by that same character? In light of this, it seems important to understand more about the stories. How often did participants imagine that they themselves were traversing the environment? How often did they generate goals and associate those goals with specific rooms in the loop? Were the goal locations (and the causal connections between locations) equally distributed across rooms? This kind of information seems important to understand the possible basis for the environment-reactivation effects that are described in Figure 4.

** It is important to show that the inter-regional [or inter-voxel] variation in the Gaussian widths [as reported in Figure 6 and Figure 4] cannot be explained by gradients of peak pattern similarity (i.e. signal-to-noise). Looking at the pale gray curves in 4C/D and the warm-colored curves in Figure 6C, it seems that wider Gaussians tend to also have lower peaks (i.e. smaller maximum pattern similarity). Because there are various normalizations applied to the BOLD data, and because there may be biases in the Gaussian fit procedure as one varies the signal-to-noise ratio in the data, it is important to show that these do not introduce a systematic relationship between peak-pattern-similarity and Gaussian width. Probably the best way to achieve this is to simulate BOLD data with a fixed set of underlying pattern similarities, then add a fixed amount of noise, then apply the same preprocessing steps and Gaussian-fitting as applied in the manuscript, and demonstrate that the sigma values remain constant even as you vary the strength of the pattern similarities (i.e. the signal-to-noise ratio). Alternatively, you could demonstrate that there is no empirical relationship between the peak similarity and the Gaussian sigma — for example, by plotting a variant of Figure 6A with the peak pattern similarity plotted instead of sigma, and demonstrating that this map lacks any of the gradients shown in Figure 6A. Relatedly, one could also just make a scatter plot of Gaussian-peak-vs-Gaussian-width and show that the two parameters are uncorrelated.

** It is important to show that the negative pattern similarities (described as “suppression” effects) are not induced by preprocessing steps. On page 11, the manuscript describes how the asymptotes of the Gaussians are interpreted: “The asymptote is an indication of the representations of environments that are not captured by the width of the Gaussian; if the asymptote is lower than baseline (defined as pattern similarity between the cue and all environments from the other map, henceforth referred to as different-map baseline), this suggests that these environments are suppressed.” However, if voxel time courses are normalized to have a zero mean, then when a pattern at time X is positively correlated with a pattern at time Y, then necessarily there will have to be another timepoint Z whose pattern has a negative correlations X and/or Y. This phenomenon is analogous to the “global signal regression” phenomenon in functional connectivity, where the presence of positive functional connectivity between some pairs of nodes necessarily implies negative functional connectivity between other nodes, because of the normalization procedure that precedes the correlation, e.g. Weissenbacher et al., 2009. If the analogous effect is present in these environment-to-environment correlation maps, then one might expect that greater-magnitude positive pattern similarity (at zero / one steps from the cue) will be associated with greater-magnitude negative pattern similarity at other step-numbers. More generally, describing a negative pattern similarity result as “suppression” is not valid unless it can be clearly established that the baseline correlation value is truly zero — it is easy for random patterns to exhibit a non-zero correlation on average with a target vector, depending on how the data are normalized before the correlation is computed. This concern could be ruled out using simulated data subjected to the same preprocessing and normalization steps as used in the text.

Ref: Weissenbacher, A., Kasess, C., Gerstl, F., Lanzenberger, R., Moser, E., & Windischberger, C. (2009). Correlations and anticorrelations in resting-state functional connectivity MRI: a quantitative comparison of preprocessing strategies. *Neuroimage*, 47(4), 1408-1416.

** Could the within-region hierarchical gradient phenomenon (Figure 6) reflect a gradient of intrinsic neural timescales across voxels (e.g. Raut et al., 2020)? In other words, suppose that Region X has an “activation response” function that ramps to its peak in 1 second and then decays, while Region Y generates a response that ramps to its peak in 2 seconds and then decays. Even if Region X and Region Y have the same activation peak, Region Y will exhibit a stronger signal-to-noise ratio for its representations of later materials [near its peak] while Region X will exhibit a stronger signal-to-noise ratio for its representation of earlier content. So even if both Region X and Region Y are representing all the same content, differences in the intrinsic timescales of population activity could act as a kind of filter, which magnifies the effective strength of neural states that are earlier or later. One way to rule this out would be to show that measurements of intrinsic timescales [e.g. autocorrelation width, as in Raut et al., 2020] are uncorrelated with the gradients shown in Figure 6.

Ref: Raut, R. V., Snyder, A. Z., & Raichle, M. E. (2020). Hierarchical dynamics as a macroscopic organizing principle of the human brain. *Proceedings of the National Academy of Sciences*, 117(34), 20890-20897.

** Could the observation of bidirectional sequence activation which was found here, but not in some prior studies, be a consequence of the story-based training that was used during the

environment decoding? If participants in prior studies were not encouraged to use stories at encoding then perhaps this is another factor that could be mentioned in the Discussion paragraph beginning “An important distinction between our experiment and past studies of prediction ...”. More generally, the manuscript would be stronger if it commented on how the story generation factor is expected to have impacted any of the results. (To what extent is this study even about a sequences of “environments” if there is a self-generated story involved... is the anticipation then not really anticipation of “situations” which happen to have taken place in environments?)

SMALLER POINTS

* The manuscript is missing a summary (either at the end of the Intro or start of the Results) of the basic experimental design, which briefly describes the experimental design and logic, and the previewing the names of the task phases. Currently, the Results section (which is located before the Methods in the Nature Communications format) begins describing the outcomes of the Anticipation Task, even though this task and its purpose have not been previously described. Similarly, it could be helpful to provide a one-sentence textual description of the Localizer Task when it is first mentioned in the Results. If the Localizer corresponds to the “training data” (Figure 1), this should be made clear in the text, and not left to a figure caption or to the Methods.

* On page 9 the basic method for pattern-matching is described as a between-subjects method: “we calculated pattern similarity between (1) multivoxel patterns of brain activity evoked during the Anticipation Task for each trial type (cue and path combination) for each participant and (2) the multivoxel patterns of brain activity evoked during Localizer Task, averaged across the remaining participants.” It seems worth discussing why the pattern match is not computed within-subject. Is there not enough held-out data for this to be feasible? Does the cross-subject method simply generate more consistent results? There are other settings in fMRI where within-subject pattern match seems to perform better than between-subject pattern match (e.g. in single-subject language encoding models from the Gallant lab or Huth lab) and so it’s worth commenting on the logic behind this between-subject method.

* It could strengthen the manuscript if there was more discussion earlier-on (e.g. in the Introduction) regarding the functional significance of bidirectional representations. On page 4 the manuscript states: “Although this work suggests that the brain may represent both anticipated and past events, these prior studies did not test whether forward and backward representations of temporally extended structure existed simultaneously in the same brain regions.” I think that one or two of the key ideas from the Discussion could be mentioned in the Introduction could help to motivate the interest in the directionality of the anticipatory sequence-representations. Similarly, at the bottom of page 4 the text states: “However, it remains unknown whether contextual modulation of temporal structure representations is specific to planning trajectories in the forward direction or if contextual relevance also modulates representations of the past.” It seems clear why you would want to have forward trajectories that are context-specific (because you want accurate context-specific predictions that can enable you to plan and prepare for action). But it seems worth laying out some of the functional considerations for why one would/would-not want to have context specific backward trajectories. Is it the case that you don’t need to have an accurate backward representation of

where you were previously, at least in some specific brain regions? Again, some motivating considerations on these points could strengthen the Introduction.

* Related to the point above, the Discussion might be strengthened by mentioning any specific theories of cortical prediction which are consistent with (or, better, inconsistent with) the hierarchical and bidirectional sequence reactivation data reported here.

* I was surprised that the hippocampus did not exhibit a context effect [i.e. different representations of the same regions within different Paths]. If the participants told different stories for Path A and Path B, then should we perhaps expect to see context effects in regions of the brain that represent "situational" content, such as regions in the default-mode network? Some more discussion of this point could be fruitful.

Reviewer #3 (Remarks to the Author):

This paper is aimed at investigating representations of future and past environments in different brain regions and how these are modulated by context. This is a topic that is currently receiving a lot of attention in the field and there are several related recent papers that have just been published. The current study used a clever design in which participants learned the order of a set of virtual environments and then anticipated future environments in the scanner. By comparing the activity patterns in this anticipation task, to the patterns elicited in other participants during the localization task, it is possible to get an estimate of the representation of future environments that is not contaminated with the learned order of environments, as the order is different across participants. While the experimental setup seems to be very well considered, I am concerned about the analyses that are used to investigate the evidence for representations of future and past environments and the conclusions that are drawn from those. My main concern is about the amount of evidence that there is an anticipation signal in the brain regions that are studied. In the first section of the paper, anticipation is studied by investigating the width of a gaussian fit to the pattern similarities of environment with varying steps into the future and the past. However, there are several issues with this approach, which I detail below. The other important concern is the number of different analyses approaches that are used in the paper in different sections, even though it would have been possible to study all questions within the same basic analysis framework. In addition, some rather weak findings seem to be taken at face value without considering the borderline significance of the results and the conclusions are phrased too strongly in relation to the quality of the evidence.

1. Looking at the raw data in figure 4, there does not seem to be a lot of evidence for representations of future or past environments in visual cortex or hippocampus. The pattern similarities are close to zero. In the hippocampus, the representations of both the current and future/past environments are very weak. This might be because the analyses rely on across-participant pattern similarity analyses. Improved matching of fine-grained patterns across participants might be necessary to see these effects more clearly, for example by hyperaligning the data first.

2. The pattern similarity between participants for one and two steps in the past and future provide a very clear and straightforward metric that can be used to directly test the evidence for representation of future and past environments. However, in the first section of the paper, the

authors chose a more convoluted approach to look at this question, which involves fitting a gaussian similarity model and then investigating the parameters of this model. It is not clear why this approach is needed and the investigation of these parameters is not straightforward. Currently, the authors test the gaussian fit in 3 ways: 1. comparing the full model against a shuffled environment baseline to test if the parameters are consistent across participants 2. comparing the amplitude and asymptote of the gaussian fit with a different-environment baseline and 3. comparing the width estimates across different brain regions. Below I detail the limitations of the overarching approach and for each of these tests.

a. Test 1 does not provide any information about which parameters of the model are driving the consistency across participants. Most likely, it is the amplitude the gaussian that is driving the observed consistency. Therefore, it is not very informative about representation of future and past environments.

b. Test 2 compared the amplitude and asymptote of the gaussian fit to the shuffled environment baseline but does not investigate the width of the gaussian and therefore cannot say anything about representation of future and past environments. It is not clear why the width was not compared to the shuffled environment baseline. I would imagine that directly comparing the width to the shuffled environment baseline would allow for a direct test of whether there are more representations of future and past environments than expected by chance.

c. In test 3, the width of the fits of different brain regions are compared. However, this means that for the brain region with the narrowest width (the visual ROI), representation of future and past states cannot be tested.

d. A bigger issue with test 3 is that is that the estimates of width might be affected by the estimates of the other parameters. The parameters of the Gaussian similarity model are fit simultaneously and are therefore not independent. I also expect that data with lower signal to noise will tend to have a lower amplitude and a wider gaussian fit. This would mean that the differences between hippocampus and the visual ROI, do not reflect increased anticipation, but decreased signal in the hippocampus. Indeed, visual inspection of the results in hippocampus suggests that there is not much evidence for representation of future and past states in that ROI. Simulations could be used to establish whether this is an issue. For example, you might simulate data with a given width and amplitude of representation and vary the signal-to-noise ratio. The question would be whether the estimate of the width co-varies with the signal-to-noise ratio. I am aware that the current draft of the manuscript already gives one example where there was a significant change in width, but not amplitude (in RSC), but I don't think the absence of a significant effect in a single case is sufficient evidence. A more systematic exploration with simulations would be necessary.

3. A related point to the previous is that analysis approaches are quite different per subsection of the results. This gives the impression that the analyses were selected based on what provided the optimal results for each sub question. I will shortly list the three main approaches here: 1. To study the evidence for anticipation in different brain regions, the analyses are based on the gaussian fit described above. 2. To study context dependent predictions during the cue representation, one and two steps in the future/past are taken together in the analyses to compare backward and forward representations. 3. To study context dependent predictions during the blank screen, one and two steps in the future are analysed separately (see figure 7), but only the results for one step are reported. It is not clear why the analyses were set up so differently in each case.

4. Given all the issues listed in point 2 and the discrepancies I described in point 3, I would

suggest that the authors use a consistent approach throughout the manuscript. In all analyses, the pattern similarity could be investigated, separately for 1 and 2 steps, without fitting a gaussian. Then the effect of direction, step-distance and context could all be based on these dependent variables. To test whether there is anticipation of future/past environments, it would be most straightforward to see whether the pattern similarity for steps 1 and 2 is significantly higher than zero (or higher than the average over the baseline timepoints (-4,-3,3 & 4)).

5. In the hippocampus there was a significant negative correlation between the asymptote of the Gaussian curve and response time slope. However, the p-value for this association was close to 0.05, suggesting that the evidence for this effect is very weak. This should be considered in the interpretation of this result. It is also not clear why the association with the asymptote was investigated but not with the width. I would expect that especially the width of the gaussian fit would relate to the speed of anticipation of nearby timepoints.

6. The reasoning for selecting specifically the PPA and RSC to look at the association between width and y-coordinate is unclear. Based on previous work, I would expect to see a hierarchy of prediction (i.e. more anticipation in more anterior/higher-level areas) across the entire (visual) cortex. Looking at the results in figure 6a, that pattern does not seem to be present more generally.

7. For the RSC and PPA, only the fits are shown and not the data. I think it's important to also show the data.

8. Related to point 2d above, I wonder if there is a correlation between the RSC and PPA y-coordinate and the width of the gaussian fit if the amplitude is used as a covariate (i.e. is there an independent association?).

9. It is not clear what the theoretical reason was for selecting the insula as an a priori ROI. The ROI also seems to be covered by relatively few reliable voxels and the voxels that do show a reliable response seem to be only in the anterior insula.

10. For the visual ROI, context-dependent anticipation is more clearly visible during the blank screen period. Therefore, I would think that it also makes sense to run the initial analyses that test for anticipation in hippocampus and visual cortex, on this blank screen period. That might give more convincing evidence for representation of future and past states.

11. The evidence for context-dependent predictions in visual cortex is very limited, the p-value is only just below 0.05 uncorrected. This should be acknowledged explicitly, and any conclusions based on this result should be phrased with caution.

12. If the insula was an a priori ROI, why was it not used as an ROI in the analyses of figure 4?

13. The results in this manuscript are quite different from what would be expected based on previous work, as far as I have seen. I think the manuscript would benefit from a more thorough discussion about similarities and differences both with respect to the study setup and outcomes. For example, regarding the finding that environments further away from the current environment are suppressed.

14. This sentence in the discussion is unclear: 'In line with these theories, recent work has shown that multiple timescales of prediction are represented simultaneously in the brain¹², with less evidence for further predictions^{22,41}.'

15. The conclusions are somewhat contradictory. First it is stated that: 'Visual cortex (V1-4) primarily represented the current environment' then 'visual cortex and insula exhibited context-dependent representations in the forward but not backward direction'. This confusion about whether/not there is representation of future environments in visual cortex, might be due to the mix-up of different analysis approaches. For the initial analyses in figure 4, there was no formal test for representation of future environments in visual cortex.

16. The conclusion that 'graded retrospection of past states can occur alongside prediction of future ones.' is based only on the analyses of the Gaussian fits. Given the limitations I mentioned above, I think that there is not sufficient evidence to support this.

17. In the the null models in which the order of the pattern similarity values was shuffled, was the order of all environments shuffled, or only those in the past and the future?

18. The ROIs should be visualized more clearly and descriptive statistics about each ROI should be provided. How many voxels did each ROI contain?

19. Voxels included in the ROIs showed across participant reliability $r > 0.1$. Which of these voxels showed statistically significant representations of the environments?

REVIEWER COMMENTS

Reviewer #1 (Remarks to the Author):

Tarder-Scholl and colleagues investigate how the brain represents the temporal order of sequences of environments, focusing on the hippocampus and visual cortex. They find that hippocampal representations extend further backward and forward in time than visual cortical representations do, and that forward representations may be relevant for behavioural performance in an anticipation task. The experimental manipulation is elegant, the immersive VR training of the sequences is naturalistic, and the analyses are novel. However, I have some serious concerns about the analyses and the interpretation of the results, which I detail below.

Major

1. It does not seem like a fitted asymmetrical Gaussian curve is the best model for the data. The forward-width of the Gaussian for the visual cortex (Fig.4c) seems to descend more sharply than the data for any of the participants and the mean over participants do. Further model comparisons (other than comparing to the null model) would be more convincing. In particular, the Gaussian for hippocampus (Fig.4d) could be compared to the flat baseline, and error bars should be added.

Response: Thank you for the valuable suggestions. We agree that, in the original submission, the overall Gaussian in visual cortex looked different than the individual participants' Gaussians. These differences were due to different sampling procedures when visualizing the overall vs individual Gaussian fits. Specifically, the overall fit was sampled with higher resolution along the x-axis than the fits for individual participants. We have now sampled the overall and individual Gaussians at the same resolution and believe that this shows much closer convergence between the fits, as can be seen in Figures 4 and 5.

We also appreciate your request to ensure that the Gaussian model fits our hippocampus data better than it fits different baselines. To test this, we created a shuffled baseline condition by randomly reordering the pattern similarity values (where the pattern similarity values indicate the correlation between the brain's activity pattern for a given trial and the environment templates). This shuffling eliminates any systematicity in pattern similarity values based on the order of the environments, allowing us to test whether the Gaussian fits our hippocampal data better than null permuted data. For this comparison, which we reported in the initial manuscript, we shuffled the order of the pattern similarity values for all environments (including the cue environment) 10000 times ("shuffled null including cue"). Thus we showed that there are significant Gaussian fits in hippocampus and visual cortex for the correctly ordered environments compared to the Gaussian fits for this shuffled baseline. We agree with you, however, that additional comparisons would be useful for better understanding how the observed fits in hippocampus and visual cortex differ from a shuffled null. In particular, a

superior Gaussian fit to correctly ordered pattern similarity values vs. shuffled pattern similarity values could occur because 1) the correctly ordered data has a peak at the cue environment whereas the shuffled baseline does not and/or because 2) the correctly ordered data has graded tails, corresponding to graded evidence for nearby environments, whereas the shuffled baseline does not.

We therefore created a second test to specifically assess whether there were graded representations for future and past environments. To do so, we removed pattern similarity to the cue environment from the correctly ordered and shuffled data and fit the Gaussian model to these no-cue datasets. For the shuffled dataset, therefore, we shuffled the order of pattern similarity values (excluding pattern similarity to the cue environment) 10000 times to create a second baseline condition that specifically tests for gradedness for past and future environments (“shuffled null excluding cue”). Specifically, if the Gaussian model provides a better fit to correctly ordered data vs. shuffled data *even when pattern similarity to the cue environment is removed from both*, that would indicate that there is significant evidence for graded similarity between the activity pattern on a given trial and nearby environments in the sequence. That is, by removing pattern similarity to the cue environment from both the observed data and the shuffled baseline, this analysis allows us to see how much of the original Gaussian fits were due to representation of the cue environment vs. representations of the upcoming and past environments. If a brain region shows a significant Gaussian fit compared to the shuffled null when the cue environment is included but *not* when the cue environment is excluded, that would indicate that its Gaussian fit is entirely driven by its representation of the cue environment. If a brain region shows a significant Gaussian fit compared to the shuffled baseline both when the cue environment is included and when it is excluded, that would indicate that its representations of both the cue and nearby environments contribute to a significant Gaussian fit.

These analyses revealed that the Gaussian fit in visual cortex was no longer significant compared to the shuffled baseline excluding the cue environment from the analysis ($p = 0.848$). This indicates that the significance of the Gaussian in visual cortex was driven by the cue environment but not by graded similarity to nearby environments. In contrast, the Gaussian fit in hippocampus was still significant compared to the shuffled baseline excluding the cue environment from the analysis ($p = 0.023$), indicating that the hippocampal Gaussian fit shows significantly graded similarity to nearby environments, in addition to representations of the cued environment.

Finally, we also fit the Gaussian model to the order of the uncued path and showed that the Gaussian fit in the hippocampus is specific to the order of the cued path (see response to Comment #2). Taken together, all of these results offer additional insights into the nature of representations in hippocampus and visual cortex. The Gaussian fit in visual cortex was primarily driven by the representation of the cue environment whereas the Gaussian fit in hippocampus was additionally driven by graded similarity to nearby environments. Thus, this shows that visual cortex only represented the current environment whereas hippocampus represented both the current environment and showed graded similarity with past and future environments. Further, these hippocampal representations were selective to the cued path

and not found for the uncued path. These analyses and results are now described in the manuscript as follows:

“We statistically tested the Gaussian fit in two ways. First, we compared the goodness-of-fit for the Gaussian model when applied to the correctly ordered pattern similarity values vs. a shuffled-order version of the pattern similarities, including pattern similarity to the cue environment (“shuffled null including cue”; **Figure 4b**). This allows us to test the null hypothesis that there was no structure in the similarity values. Second, we removed the pattern similarity to the cue environment and fit the Gaussian model only to the pattern similarities for upcoming and past environments, in both the correct order and the shuffled order (“shuffled null excluding cue”; **Figure 4b**). If a brain region shows a superior Gaussian fit for the observed vs. shuffled data only when the cue environment is included but not when it is excluded, that would indicate that its Gaussian fit is entirely driven by its representation of the cue environment. If a brain region shows a superior Gaussian fit for the observed vs. shuffled data both when the cue environment is included and when it is excluded, that would indicate that this region represents the cue environment and also has systematically graded representations of nearby environments.

In visual cortex, the Gaussian model provided a significantly better fit to the correctly ordered vs. shuffled pattern similarity values when the cue environment was included in both the observed and shuffled data (correctly ordered data vs. shuffled null including cue, $p < 0.001$, $R^2 = 0.699$). The amplitudes of participants’ Gaussian fits were significantly higher than the different-map baseline, indicating that the cue environment was represented while it was on the screen (mean = 0.091, standard deviation = 0.029; $t(31) = 18.01$, $p < 0.000001$; **Figure 4c**). The asymptote was significantly lower than the different-map baseline, suggesting that other environments surrounding the cue were suppressed (mean = -0.014, standard deviation = 0.009; $t(31) = -5.97$, $p = 0.000001$; **Figure 4c**). The backward and forward widths (σ) were 0.712 steps and 0.634 steps, respectively, and were not significantly different from each other ($V(31) = 195.00$, $p = 0.203$), suggesting that representations were not biased toward one direction over the other. However, when pattern similarity to the cue environment was excluded, the Gaussian model was no longer a better fit for correctly ordered vs. shuffled data (correctly ordered data vs. shuffled null excluding cue, $p = 0.848$, $R^2 = -0.006$; **Figure 4c**). This indicates that the significance of the Gaussian was driven by the cue environment but not by graded similarity to nearby environments in the sequence.

Turning to representations in hippocampus, the asymmetric Gaussian once again provided better fits to the correctly ordered vs. shuffled pattern similarity values when the cue environment was included (correctly ordered data vs. shuffled null including cue, $p = 0.029$, $R^2 = 0.032$). In the hippocampus, similar to visual cortex, the amplitude of the Gaussian fit was significantly higher than the different-map baseline (mean = 0.0185, standard deviation = 0.030; $t(31) = 2.959$, $p = 0.005$; **Figure 4d**) and the asymptote was significantly lower than the different-map baseline (mean = -0.007, standard deviation = 0.014; $t(31) = -3.476$, $p = 0.001$; **Figure 4d**). Thus, hippocampus represented the cue environment while it was on the screen and suppressed environments further away. The backward and forward widths (σ) were 2.313 steps and 1.782 steps, respectively, and were again not significantly different from each other ($V(31) = 231.00$, $p = 0.548$), suggesting that the hippocampus represented forward and backward environments similarly. Critically, even when the cue environment was

excluded, the Gaussian was still a better fit to the correctly ordered vs. shuffled data (correctly ordered data vs. shuffled data excluding cue, $p = 0.023$, $R^2 = 0.028$; **Figure 4d**). This indicates that the hippocampal Gaussian fit shows significantly graded similarity to nearby environments in the past and future, in addition to representing the cued environment. In contrast to hippocampus and visual cortex, the Gaussian fit was not significantly better for the correctly ordered data vs. the shuffled null (including the cue) in the insula ($p = 0.139$, **Supplementary Figure 1**.)” (p.11-13, Results)

“To test whether the parameters of the Gaussian curve were consistent across participants, we fit the asymmetrical Gaussian curve on all but one participant’s data (e.g. P2-32) and then measured the sum of squared errors (observed vs. predicted pattern similarity values) when using this curve to predict the held-out participant’s data (e.g. P1). We repeated this procedure for each choice of held-out participant to obtain an average error value. We conducted two tests. First, we compared the Gaussian fit to correctly ordered data and the Gaussian fit to shuffled pattern similarity values, including pattern similarity to the cue environment (“shuffled null including cue”). Second, we conducted the same comparison but removed pattern similarity to the cue environment from both the correctly ordered and shuffled data (“shuffled null excluding cue”). In both cases, we fit the Gaussian to shuffled pattern similarity values 10000 times and obtained the goodness of fit each time to create a null distribution. The goodness of fit for the correctly ordered data vs. null distributions were then compared. If, for a given brain region, the Gaussian model provides a better fit to the correctly ordered vs. shuffled data both when the cue is included and when it is excluded, that would indicate that the brain region’s representations of both the cue and graded representations of nearby environments contribute to a significant Gaussian fit. If, however, the Gaussian model provides a better fit to the correctly ordered vs. shuffled data only when the cue is included and not when it is excluded, that would indicate that its Gaussian fit was driven by its representation of the cue environment without any evidence for graded representations of the past or future.” (p.47, Methods)

We have also added error bars to Figures 4c and d, as well as to Figure 5a and 5b:

Figure 4 | Bidirectional and graded representations of temporal structure in hippocampus but not early visual cortex. (a) Schematic depiction of Gaussian analysis. We obtained the correlation between a given participant's (e.g., P1) cue screen activity pattern for each trial of the Anticipation Task and the remaining participants' (e.g., P2-P32) averaged patterns of activity for each of the environment templates on the cued path. We then ordered the resulting pattern similarity values with the cue in the center and fit an asymmetrical Gaussian curve. (b) Gaussian similarity model. The amplitude of the curve is an indication of the degree to which a brain region is representing the cue environment while it is on the screen. The widths (σ) of the curve indicate how similarity to neighboring environments falls off with the

number of steps in the forward and backward directions. Wider (vs. narrower) widths indicate that the brain region represents environments that are further away. The asymptote quantifies the representations of environments that are not captured by the width of the Gaussian; if the asymptote is lower than the dashed line (different-map baseline) this suggests that these environments are suppressed. The Gaussian fit for the correctly ordered data was compared to the Gaussian fit for shuffled data including the cue environment (left) and shuffled data excluding the cue environment (right). A significant Gaussian fit for the correctly ordered vs. shuffled data only for the shuffled null including the cue would indicate a brain region is only representing the cue environment. A significant Gaussian fit for the correctly ordered vs. shuffled data for both the shuffled null including and excluding the cue would indicate systematically graded representations of nearby environments in addition to representations of the cue environment. **(c)** Gaussian curve in visual cortex for the cued path order. Visual cortex strongly represented the cue environment while it was on the screen (above-baseline amplitude) and did not strongly represent nearby environments (narrow forward and backward widths (σ)), instead showing suppression of environments other than the cue (below-baseline asymptote). The visual cortex Gaussian fit was significantly better than the shuffled null including the cue, but not the shuffled null excluding the cue. **(d)** Gaussian curve in the hippocampus for the cued path order. The hippocampus represented the cue environment while it was on the screen (above-baseline amplitude), represented nearby environments in a graded manner, in both the forward and backward direction (wide forward and backward widths (σ)), and suppressed environments that were furthest away (below-baseline asymptote). The hippocampus Gaussian fit was significantly better than the shuffled null including the cue and the shuffled null excluding the cue. Points indicate average pattern similarity at each step from the cue and error bars indicate standard error of the mean. Colored line indicates the average Gaussian fit across participants, with the red end of the rainbow scale indicating higher pattern similarity and the purple end indicating lower pattern similarity. The rainbow scale is applied to values in each brain region separately. Gray lines indicate each participant's Gaussian curve. * $p < .05$, *** $p < 0.001$

Figure 5 | Representations of bidirectional temporal structure are context-specific. The Gaussian curve for the uncued path order was not significantly better than the shuffled null excluding the cue environment in either visual cortex (**a**) or hippocampus (**b**). Points indicate average pattern similarity at each step from the cue and error bars indicate standard error of the mean. Colored line indicates the average Gaussian fit across participants for the uncued path order, with the rainbow scale indicating pattern similarity values, scaled separately for each brain region along the same scale as used for the cued path Gaussian fit (red end = higher pattern similarity, purple end = lower pattern similarity). Gray lines indicate each participant's Gaussian curve.

2. Does the analysis in Figure 7b suggest that the uncued sequences were replayed just as much as the cued ones during the anticipation task in visual cortex and hippocampus? So would the results in Figure 4 look the same if the uncued rather than the cued path were

used for the analysis? (The concern is somewhat alleviated for visual cortex by the blank screen analysis in Figure 7c, but not for hippocampus.) This may be a misunderstanding, but if not, that seems a very important point that should be emphasised more strongly from the start. At a minimum, the results in Figure 4 should be shown for the uncued path as well. Also, with vast literature showing that the hippocampus holds contextual information and may be involved in pattern completion processes, why not look at the blank screen period for the hippocampus as well?

Response: Thank you for the valuable suggestion. We agree that the paper would be strengthened by investigating the pattern similarity values in our regions of interest when they are arranged based on the order of the uncued path as well as the cued path. We have done as suggested by measuring the Gaussian fit for the uncued path in the hippocampus and visual cortex using the same analysis as that reported in Figure 4. If visual cortex and the hippocampus represent the order of surrounding environments in a context-dependent manner, we may not observe statistically reliable Gaussian fits for the uncued path. Because we wanted to test whether graded representations of past and future states were context-sensitive, and because the cue environment was the same on the cued and the uncued path, we only tested the Gaussian fit for the uncued path by comparing the correctly ordered data to the shuffled null excluding the cue.

The hippocampus showed evidence for context-dependent representations: the uncued path Gaussian fit was not significantly better for the correctly ordered data vs. the shuffled null excluding the cue ($p = 0.915$). We then assessed whether the cued path Gaussian fit was significantly better than the uncued path Gaussian fit. The evidence for graded past and future representations along both the cued and uncued paths was measured by comparing each fit to the shuffled null excluding the cue, and then testing whether the difference in fit for the correctly ordered vs shuffled data was larger for the cued vs. uncued path. Indeed, the Gaussian fit was significantly better for the order of the cued path compared to the uncued path (correctly ordered data vs. shuffled null excluding cue for cued vs uncued path, $p = 0.005$). Together, these results show that the graded similarity to nearby environments was specific to the order of the cued path.

We also conducted the same analyses for visual cortex, but did not expect to see context-dependent representations of the future and past, given that visual cortex did not show evidence for future and past environments even along the cued path (see response to comment #1). Indeed, when environments were ordered along the uncued path, the visual cortex Gaussian fit was not better than the Gaussian fit to the shuffled null excluding the cue ($p = 0.993$). Further, unlike hippocampus, the Gaussian fit in visual cortex was not significantly different for the cued path vs the uncued path (correctly ordered data vs. shuffled null excluding cue for cued vs uncued path, $p = 0.132$). This supports the perspective that visual cortex represented the cue environment, but did not represent surrounding environments in either the cued or uncued path. These new analyses and interpretations are now included in the manuscript as follows:

“Next, we determined whether the gradedness of sequence representations was specific to the order of the cued path or if it was also present for the uncued path, which contains the same environments but in a different order (see Methods). To investigate this, we repeated our Gaussian analysis with pattern similarity values arranged along the order of the uncued path. If the hippocampus represents the order of surrounding environments in a context-dependent manner, we may not observe statistically reliable Gaussian fits for the uncued path. To test for graded representations of the past and future along the uncued path, we compared the Gaussian fit to the shuffled null excluding the cue environment (because the cue environment was the same for both cued and uncued paths). We also repeated this analysis in visual cortex for completeness, but because visual cortex showed no evidence for graded representations of the past and future for the cued path, we also expected it to show no gradedness for the uncued path.

As expected, the Gaussian fit in visual cortex for the uncued path was not better than the fit for the shuffled null excluding the cue environment ($p = 0.993$, $R^2 = -0.0173$; **Figure 5a**), and the Gaussian fit was not significantly different for the cued path vs the uncued path ($p = 0.132$), indicating that visual cortex did not represent surrounding environments on either the cued or uncued path. In the hippocampus, the Gaussian fit for the uncued path was not significantly better than the fit for the shuffled null excluding the cue environment ($p = 0.915$, $R^2 = -0.012$, **Figure 5b**). Further, the Gaussian fit was significantly better for the order of the cued path compared to the uncued path ($p = 0.005$). Together, these results show that the graded similarity to nearby environments in the past and future in hippocampus was specific to the order of the cued path.” (p. 16–17, Results)

“Finally, to test whether the Gaussian representations were context dependent, we repeated the same Gaussian fitting procedure as above but with the pattern similarity values arranged on the order of the uncued path. Importantly, we only tested whether the Gaussian fit for the uncued path was significantly better than the shuffled null excluding the cue because 1) the cue environment was the same across paths and 2) this comparison specifically tests evidence for graded representations of the past and future. We also assessed whether the cued order Gaussian fit was significantly better than the uncued order Gaussian fit when compared to the shuffled null excluding the cue. This allowed us to specifically test whether there was more evidence for graded representations of the past and future along the cued vs. uncued path in the absence of similarity to the cue environment. We computed the difference between the sum of squared errors for 1) the correctly ordered data along the cued path vs. the shuffled null, excluding the cue from both; and 2) the correctly ordered data along the uncued path vs. the shuffled null, excluding the cue from both. We then tested how often the difference in errors for the cued path was larger than the difference for the uncued path.” (Methods, p. 48-49)

We also now visualize these results in Figure 5:

Figure 5 | Representations of bidirectional temporal structure are context-specific. The Gaussian curve for the uncued path order was not significantly better than the shuffled null excluding the cue environment in either visual cortex (**a**) or hippocampus (**b**). Points indicate average pattern similarity at each step from the cue and error bars indicate standard error of the mean. Colored line indicates the average Gaussian fit across participants for the uncued path order, with the rainbow scale indicating pattern similarity values, scaled separately for each brain region along the same scale as used for the cued path Gaussian fit (red end = higher pattern similarity, purple end = lower pattern similarity). Gray lines indicate each participant's Gaussian curve.

We also appreciate your request to run the Gaussian analysis for hippocampus during the blank screen period as well. The Gaussian fit in hippocampus during the blank screen was not significantly better than the fit for the shuffled null including the cue ($p = 0.604$, $R^2 = -0.003$) or the shuffled null excluding the cue ($p = 0.570$, $R^2 = -0.002$). This suggests that the systematic

representations of temporal structure were specific to the cue screen period. However, the lack of significant findings during the blank screen period should be interpreted with caution because the multivoxel patterns of activity were obtained from GLMs that included the cue screen and the blank screen periods in a single model, with different regressors. Brain activity triggered by the cue period that persisted into the blank period may have been accounted for by the regressor for the cue period, complicating interpretation of the blank period. Indeed, pattern similarity to the cue environment was negative during the blank screen period, consistent with the interpretation that cue-triggered responses were explained by the cue regressor and led to an anticorrelation with the blank period. Therefore, it's unclear whether hippocampus did not contain sequence representations during the blank screen or whether the results were confounded with the cue screen results; for this reason, we did not include the blank screen Gaussian analysis in the main manuscript.

3. The interpretation of the Gaussian curve results is unclear. The Gaussian for visual cortex shows that steps -1 and +1 have below zero pattern similarity (Fig.4c), so can the Gaussian really be said have a meaningfully non-zero width, or is just this an artefact of the analysis?

Response: Thank you for requesting this clarification. We agree that visual cortex is primarily representing the current moment, with little evidence for representations of the surrounding environments during the cue period. The Gaussian function must return a width parameter that is greater than 0; for this reason, we never compare the widths to 0. Indeed, as noted in our response to your first comment, we found that the Gaussian fit in visual cortex was primarily driven by its representation of the cue environment. This was because when we excluded the cue environment from the analysis, the Gaussian fit for visual cortex was not superior to the shuffled null – indicating no evidence for graded representations of past or future environments.

For these reasons, we were careful in saying that visual cortex primarily represented the current moment; but we have gone through the manuscript thoroughly to ensure that we do not indicate any evidence for future or past environments in visual cortex based on the Gaussian analysis of the cue period. Our main comparisons of interest regarding the Gaussian widths were between visual cortex and hippocampus. This comparison showed that the widths in hippocampus are greater than in visual cortex, suggesting comparatively further reaching representations in the hippocampus. We have clarified our text throughout to make sure we do not leave readers with the impression that the Gaussian in visual cortex shows evidence for future or past environments. We have also added a small section to emphasize this, as follows:

“Unlike hippocampus, we were unable to detect bidirectional temporal structure representations in visual cortex. This result is seemingly in contrast to a rich literature showing predictive representations in early visual regions¹¹. One possibility for why we were unable to detect predictions in early visual cortex is due to the nature of our stimuli. Past work decoding predictions from early visual regions has tended to use relatively simple stimuli such as gratings^{13,14}, fractals¹⁶, or specific spatial locations⁵⁹. In contrast, we used rich, naturalistic scenes that were experienced in immersive virtual

reality from multiple viewpoints, making it unlikely that participants were generating predictions of low-level visual features tied to specific retinotopic locations. Instead, individuals in our study may have been predicting whole scenes at a relatively coarse (vs detailed) level, leading to the predictive representations observed in higher order scene-specific visual regions such as PPA and RSC³⁹.” (p. 28, Discussion)

4. More justification is needed for the claim on p. 11: "if the asymptote is lower than baseline (defined as pattern similarity between the cue and all environments from the other map, henceforth referred to as different-map baseline), this suggests that these environments are suppressed." Given that pattern similarity for all 'different' environments (Fig.3d) seems to be below baseline, it seems more likely to just be an artefact of the analysis that unrelated environments have below zero similarity. And, if more distant environments are suppressed, how are they nonetheless still activated afterwards, to allow the participant to respond correctly?

Response: Thank you for your comment; we now realize we may not have been clear about our approach. The negative correlations shown in Figure 3d, from the Localizer Task, are across all different environments, regardless of which map they belong to. The baseline for the Gaussian analysis consists of the pattern similarity values between the cue screen activity pattern and the activity pattern templates for the eight environments *that are not in the cued map* (different-map baseline). The asymptote is suppressed not just below zero, but relative to this different-map baseline. Thus, even though there is a bias for different environments to be slightly negatively correlated with each other during the Localizer Task, that does not explain why some same-map environments are suppressed *even further than* environments in the different map during the Anticipation Task. We have clarified this in the manuscript as follows:

“The asymptote is an indication of the representations of environments that are not captured by the width of the Gaussian; if the asymptote is lower than baseline (defined as pattern similarity between the cue screen activity patterns and all environment templates from the other map, henceforth referred to as different-map baseline), this suggests that these environments are suppressed. Importantly, our pattern similarity approach considers independent time points (similarity between our Anticipation Task cue periods and environment templates from the Localizer Task), meaning that more evidence for the cued environment does not necessitate suppression of other environments – suppression need not be observed, and all pattern similarity values could be positive, such as in Figure 7c.” (p. 11, Results)

“We also performed statistical tests on the parameters of the correctly ordered Gaussians. We compared the amplitude and asymptote to a baseline which consisted of the pattern similarity values between the cue screen activity pattern and the activity pattern templates for the eight environments that were not in the cued map (different-map baseline). We tested whether the amplitude was significantly above the different-map baseline, and whether the asymptote was suppressed below the different-map baseline, using one-sample t-tests across participants.” (p. 48, Methods)

We also thank you for your question about how individuals make correct predictions when the representations of further environments are suppressed. Suppression of further environments

may be useful during the cue and blank screen as individuals anticipate upcoming nearby environments, but these far-away environment representations may not be suppressed if they subsequently appear as probes. If they appear as probes, their representations may have to be reactivated, and this reactivation will take more time if they were initially suppressed – leading to response time costs. Indeed, this is consistent with what we observed in our data. The more an individual suppressed further environment representations in the hippocampus during the cue period, the greater their response time cost for further vs closer environments during the probe screen. Thus, participants may need to take more time when responding to further environments during the probe screen because those environment representations were more suppressed during the cue screen and needed to be reactivated. We expand on this interpretation in the Discussion:

“In addition to representing nearby environments in the past and future, we also found that the hippocampus suppressed more distant environments, showing deactivation of these environments' activity patterns relative to an unrelated-environment baseline. To our knowledge, prior work has not looked at suppression of far away environments in a sequence during prediction of upcoming events^{12,22}. It is possible that, in this previous work, suppression of further events was present but went undetected. Another possibility for such suppression is a result of the overlapping paths in the current study: individuals may have suppressed further environments on the cued path if they were coming up sooner on the uncued path. Thus, suppressing environments that were far away in the cued path but nearby in the uncued path may have been useful in avoiding confusion between the overlapping paths, and contributed to the findings observed here. Although suppressing distant environments can be beneficial for responding to imminent events, it can also lead to behavioral costs. For example, if a more distant environment appears as a probe, its representation may have to be reactivated, and this reactivation will take more time if it was initially suppressed – leading to response time costs. Indeed, hippocampal suppression of distant environments was related to response times costs for anticipating further events. This highlights a trade-off between prioritizing nearby events and being able to quickly respond to upcoming events further in the future. However, it is important to note that this individual-differences correlation was observed with a relatively small sample size, requiring replication in future work. Future work could also test whether the widths of brain regions' predictive horizons influence behavioral performance in studies specifically powered for individual differences analyses.” (p. 26, Discussion)

5. The authors relate the asymptote to behaviour (Fig.5), which is an interesting analysis. It might be interesting (and more intuitive) to do the same for the backward and (especially) forward widths of the Gaussians, i.e., the more forward-looking the hippocampus is, the better people do at the anticipation task. Of course, correlating multiple variables to individual differences in behaviour poses a multiple comparison problem, especially with a small sample (by individual difference analysis standards). Perhaps this is a limitation that should be mentioned in either case: brain-behaviour correlations with small samples are notoriously tricky.

Response: Thank you for this suggestion. There were no significant relationships between forward and backward Gaussian widths and response time or accuracy on the Anticipation

Task in any brain region. We now report these null correlations in the Results and discuss the limitations of interpreting these correlations given our small sample size in the Discussion, as mentioned above and copied here for your convenience:

“We further tested whether there was a relationship between response time slopes and width of the Gaussian fit, such that narrower widths are associated with steeper response time slopes. There was no relationship between either forward or backward width and response time slope in visual cortex (Forward Width: $\rho = -0.149$, $p = 0.415$; Backward Width: $\rho = 0.071$, $p = 0.698$) or hippocampus (Forward Width: $\rho = 0.233$, $p = 0.199$; Backward Width: $\rho = -0.008$, $p = 0.966$).” (p. 19-20, Results)

“Indeed, hippocampal suppression of distant environments was related to response times costs for anticipating further events. This highlights a trade-off between prioritizing nearby events and being able to quickly respond to upcoming events further in the future. However, it is important to note that this individual-differences correlation was observed with a relatively small sample size, requiring replication in future work. Future work could also test whether the widths of brain regions’ predictive horizons influence behavioral performance in studies specifically powered for individual differences analyses.” (p. 26, Discussion)

Minor

1. When participants made errors or had longer response times in the anticipation task, were these more-often-than-not congruent with the order of environments in the uncued path?

Thank you for this excellent suggestion. We examined whether errors on the Anticipation Task were more likely to occur when the incorrect probe would have been the correct answer on the uncued path, controlling for steps into the future on the cued path. We found that trials on which the incorrect answer would have been the correct answer for the uncued path (vs. incorrect answer for both paths) were not associated with slower response times ($\beta = -0.029$, $p = 0.267$) or higher errors, although there was a trend toward lower accuracy on trials in which the cued and uncued paths led to different answers ($\beta = -0.352$, $p = 0.065$). However, it is important to note two caveats to this analysis. Firstly, there were few trials in this analysis as participants were, on average, incorrect only 13% of the time. Secondly, because of the predetermined path structure, trials in which the incorrect answer was correct on the uncued path (as opposed to incorrect on both paths) had more steps between the cue and the closest environment on the cued path. Indeed, the sign of the effect noted above flipped depending on whether the steps into the future regressor was included or not, suggesting potentially problematic collinearity between the regressors in the model. Future studies could disentangle steps into the future from congruency of the incorrect answer with the uncued path, and decrease participant training to increase error rates overall, to more cleanly investigate interference between the two paths.

2. *Were representations of upcoming environments weaker on trials in which participants made errors in the anticipation task? And if so, was there any evidence for representations of the uncued path, especially on those error trials where participants' response was congruent with the order of the scenes in that alternative path? Of course there may not be sufficient numbers of error trials to do these analyses, but it would be great if you could.*

Response: Thank you for this excellent suggestion. We tried running the Gaussian analysis only on incorrect trials to determine whether there were diminished representations of past and future states. Indeed, when considering only incorrect trials, the Gaussian fit in hippocampus was not significantly better for the correctly ordered data vs. the shuffled null including the cue ($p = 0.518$, $R^2 = -0.011$) or vs. the shuffled null excluding the cue ($p = 0.7635$, $R^2 = -0.02$). The Gaussian fit in hippocampus for the uncued path was also not better than the fit for the shuffled null excluding the cue when considering only incorrect trials ($p = 0.939$, $R^2 = -0.033$). However, it is important to note that participants were, on average, incorrect only 13% of the time, meaning that there were very few trials in these analyses. It is therefore unclear whether participants were not representing temporal structure on these incorrect trials, or whether we were underpowered to detect such representations. For these reasons, we opted not to include this analysis in the manuscript.

3. *It is unclear why the authors have only done across-subject analyses. I appreciate the argument on p.9 regarding randomised sequences for individual participants, but this concern might already be sufficiently taken care of by having different (blue and green) paths of the same scenes within participants. Within participant analyses might be expected to be more sensitive, and given that the hippocampus representations vary quite a lot across subjects (Fig.4d), it would be interesting to see the results when comparing within-subjects.*

Response: Thank you for this suggestion. We did try to conduct within-participant analyses and created participant-specific templates for each environment by using the four runs from the Localizer Task. However, these individual-participant environment templates were noisier than the across-participant environment templates. Specifically, we conducted the same analyses as described in the *Localizer Task Analyses* section of the manuscript, but using held-out runs of the Localizer Task within a participant. We obtained the within-participant multivoxel pattern of brain activity for each environment within each of our ROIs for a single run of the Localizer Task (e.g. P1, Localizer Task Run 1). We then compared that pattern to the averaged within-participant multivoxel patterns for the remaining runs of the Localizer Task (e.g. P1, Localizer Task Runs 2-4). Akin to Figure 3d, we then compared within-participant activity patterns for the same environment to the activity patterns for different environments within each ROI. However, the resulting within-participant pattern similarity values were noisier than the across-participant analysis: there was not a clear difference in pattern similarity values for the same environment vs different environments. We believe that there are two potential explanations for noisier within- vs across-participant environment patterns: 1) a given individual's template for a given environment could contain predictive information about upcoming environments if they are activating one path more than the other during the Localizer Task, or activating one path sometimes and the other path at other times during the

Localizer Task; and 2) we may have been underpowered to detect participant-specific patterns of activity for all 16 environments from just four runs of the Localizer Task.

The across-participant approach we took may have therefore helped us identify hippocampal activity patterns that reflected the features of each environment uncontaminated by any prediction or other forms of participant-specific cognition that may have occurred during the Localizer Task. We think this is a particular strength because 1) the across-participant templates are cleaner measures of individual environments, uncontaminated by future environments, than within-participant templates and therefore offer a stronger test of our hypotheses; and 2) the finding of reliable hippocampal activity patterns for individual environments *across participants* adds to an emerging body of work demonstrating shared hippocampal representations across individuals, which previously went undetected (Koch et al., 2020; Chen et al., 2021). One possible reason for our ability to detect across person representations in the hippocampus could be due to the immersiveness of our virtual reality environments during encoding. While speculative, it's possible that participants were more strongly reactivating environment characteristics that were learned in immersive VR compared to environments or images viewed on a desktop computer, which is the standard approach in other cognitive neuroscience studies. We now discuss the implications of these across-participant findings in the Discussion as follows:

“To investigate bidirectional, context-dependent sequence representations, we carefully manipulated overlapping sequences. Pairs of sequences contained the same environments in a different order. *They were structured such that, for each environment, the environments one and two steps into the future and the past were all different in the two sequences. We found that prospective and retrospective representations in hippocampus were context-dependent, emerging only for the cued but not the uncued sequence. This dovetails with prior work showing context-sensitive representations of future events in hippocampus³⁷, and extends this work to graded and bidirectional sequence representations within a context. Strikingly, we observed such effects in hippocampus even with environment templates that were identified across participants. The finding of reliable hippocampal activity patterns for individual environments across participants adds to an emerging body of work demonstrating shared hippocampal representations across individuals, representations that were previously difficult to detect^{57,58}.*” (p. 27-28, Discussion)

4. *You could make it clearer that results in Fig. 4c-e combines the cued and uncued paths, rather than only the cued path.*

Response: Thank you for this suggestion. The results in Figure 4c-e are only for the order of the cued path. We have made the following changes to the manuscript to clarify this analysis:

“We examined whether the sequence *order for the cued path* was reflected in the brain’s representations during the Anticipation Task. *We developed an analysis approach that allowed us to: 1) detect representations of past and future environments; 2) test whether such representations were graded as a function of distance and specific to the task-relevant context; and 3) determine if these representations were hierarchically organized across the brain. Our approach involved fitting an asymmetrical Gaussian curve to the pattern similarity values arranged*

following the order of the cued path (see Methods). The Gaussian similarity model has four parameters: amplitude, asymptote, and forward and backward width (σ) (**Figure 4b**). The amplitude of the curve indicates the degree to which a brain region is representing the cue environment while it is on the screen. The forward and backward widths (σ) of the curve indicate how similarity to neighboring environments falls off with the number of steps in the forward and backward directions. Wider (vs. narrower) widths indicate that the brain region represents environments that are further away. If a brain region has a wide forward width but a narrow backward width, this indicates a bias towards representing upcoming environments, indicating anticipation, over retrospective representations of preceding environments. The asymptote is an indication of the representations of environments that are not captured by the width of the Gaussian; if the asymptote is lower than baseline (defined as pattern similarity between the cue screen activity patterns and all environment templates from the other map, henceforth referred to as different-map baseline), this suggests that these environments are suppressed. Importantly, our pattern similarity approach considers independent time points (similarity between our Anticipation Task cue periods and environment templates from the Localizer Task), meaning that more evidence for the cued environment does not necessitate suppression of other environments – suppression need not be observed, and all pattern similarity values could be positive, such as in Figure 7c.” (p. 11-12, Results)

“Next, we fit an asymmetrical Gaussian curve to the resulting pattern similarity values arranged on the order of the cued path.” (p. 46, Methods)

We also now run the Gaussian analysis on the uncued path order, visualize the results in Figure 5, and report them in the Results section. This new analysis should further clarify that Figure 4 represents the cued path analysis and the new Figure 5 represents the uncued path analysis. We have made the following changes to reflect this new analysis, as previously discussed in our response to your comment #2 and copied here for your convenience:

“Next, we determined whether the gradedness of sequence representations was specific to the order of the cued path or if it was also present for the uncued path, which contains the same environments but in a different order (see Methods). To investigate this, we repeated our Gaussian analysis with pattern similarity values arranged along the order of the uncued path. If the hippocampus represents the order of surrounding environments in a context-dependent manner, we may not observe statistically reliable Gaussian fits for the uncued path. To test for graded representations of the past and future along the uncued path, we compared the Gaussian fit to the shuffled null excluding the cue environment (because the cue environment was the same for both cued and uncued paths). We also repeated this analysis in visual cortex for completeness, but because visual cortex showed no evidence for graded representations of the past and future for the cued path, we also expected it to show no gradedness for the uncued path.

As expected, the Gaussian fit in visual cortex for the uncued path was not better than the fit for the shuffled null excluding the cue environment ($p = 0.993$, $R^2 = -0.0173$; **Figure 5a**), and the Gaussian fit was not significantly different for the cued path vs the uncued path ($p = 0.132$), indicating that visual cortex did not represent surrounding environments on either the cued or uncued path. In the hippocampus, the Gaussian fit

for the uncued path was not significantly better than the fit for the shuffled null excluding the cue environment ($p = 0.915$, $R^2 = -0.012$, **Figure 5b**). Further, the Gaussian fit was significantly better for the order of the cued path compared to the uncued path ($p = 0.005$). Together, these results show that the graded similarity to nearby environments in the past and future in hippocampus was specific to the order of the cued path.” (p. 16-17, Results)

“Finally, to test whether the Gaussian representations were context dependent, we repeated the same Gaussian fitting procedure as above but with the pattern similarity values arranged on the order of the uncued path. Importantly, we only tested whether the Gaussian fit for the uncued path was significantly better than the shuffled null excluding the cue because 1) the cue environment was the same across paths and 2) this comparison specifically tests evidence for graded representations of the past and future. We also assessed whether the cued order Gaussian fit was significantly better than the uncued order Gaussian fit when compared to the shuffled null excluding the cue. This allowed us to specifically test whether there was more evidence for graded representations of the past and future along the cued vs. uncued path in the absence of similarity to the cue environment. We computed the difference between the sum of squared errors for 1) the correctly ordered data along the cued path vs. the shuffled null, excluding the cue from both; and 2) the correctly ordered data along the uncued path vs. the shuffled null, excluding the cue from both. We then tested how often the difference in errors for the cued path was larger than the difference for the uncued path.” (p. 48-49, Methods)

Reviewer #2 (Remarks to the Author):

In this paper, Tarder-Stoll et al. examine the neural representations that are activated when people are anticipating spatial environments that unfold in a typical sequence. In regions including the hippocampus and insular, they observed neural signatures of environments that occurred both earlier and later elements than the current sequence elements, while participants prepared to make a judgment about upcoming sequence elements. They also observed an interaction between the position of brain regions and the sequence activations, with more anterior brain regions activating patterns associated with more distant sequence elements.

The main strengths of this paper are: (i) that it uses a sophisticated and careful experimental design that enables the measurement of both forward and backward sequence representations without horizon effects, while allowing for context cues; and (ii) the paper address a question of current interest in memory research, as sequence replay and prediction are currently of great interest in the study of human memory. The manuscript is well-written and well-illustrated, and the Discussion does a good job situating these findings in the broader literature.

The main area for improving this manuscript will be in methodologically confirming (via further analysis and/or simulations) that some of the key claims (i.e. regional gradients in the properties of the Gaussian-fits) cannot be explained by other forms of spatial variation, such

as variation in signal-to-noise or variation in the effect of preprocessing/normalization prior to computing the pattern similarity.

MAIN POINTS

*** The manuscript should demonstrate that the Gaussian model is actually a good model of the PatternSimilarity-by-Steps data.(Figure 3C). The fit could be measured using, e.g. the cross-validated R^2 of the Gaussian model, but any method is fine as long as it quantifies overall performance in held-out data. It is also crucial to check whether the “goodness” of the Gaussian model fit varies by ROI, and to ensure that this goodness-of-fit is not confounded with other ROI-based variation in the results.*

Response: Thank you for your comment. We agree that we should report the goodness-of-fit of the Gaussian model to the held-out data. For each of our ROIs and our searchlight analysis, we calculated R^2 as 1 minus the sum of squared errors of the Gaussian fit for the correctly ordered data divided by the mean sum of squared errors of the Gaussian fits for the shuffled data. R^2 values greater than 0 indicate smaller squared errors for the correctly ordered data than the shuffled data. Interestingly, we found that the goodness-of-fit varies across ROI depending on whether the cue environment was included or excluded from the correctly ordered and shuffled data. When testing the cued path Gaussian fit compared to the shuffled null including the cue environment (see response to Reviewer #1), the R^2 was higher in visual cortex ($R^2 = 0.699$) than in hippocampus ($R^2 = 0.032$). However, when testing the cued path Gaussian fit compared to the shuffled null excluding the cue environment (see response to Reviewer #1), the R^2 was higher in hippocampus ($R^2 = 0.028$) than in visual cortex ($R^2 = -0.006$). This suggests that (1) the Gaussian model is a good fit to the data, as suggested by better performance on the correctly ordered data compared to the shuffled nulls (with the exception of the visual cortex Gaussian fit compared to the shuffled null excluding the cue environment), and (2) R^2 does not systematically vary by ROI, instead flipping direction depending on whether the analysis includes or excludes the cue environment. We now report the R^2 of the Gaussian fit in each ROI:

“In visual cortex, the Gaussian model provided a significantly better fit to the correctly ordered vs. shuffled pattern similarity values when the cue environment was included in both the observed and shuffled data (correctly ordered data vs. shuffled null including cue, $p < 0.001$, $R^2 = 0.699$). The amplitudes of participants’ Gaussian fits were significantly higher than the different-map baseline, indicating that the cue environment was represented while it was on the screen (mean = 0.091, standard deviation = 0.029; $t(31) = 18.01$, $p < 0.000001$; **Figure 4c**). The asymptote was significantly lower than the different-map baseline, suggesting that other environments surrounding the cue were suppressed (mean = -0.014, standard deviation = 0.009; $t(31) = -5.97$, $p = 0.000001$; **Figure 4c**). The backward and forward widths (σ) were 0.712 steps and 0.634 steps, respectively, and were not significantly different from each other ($V(31) = 195.00$, $p = 0.203$), suggesting that representations were not biased toward one direction over the other. However, when pattern similarity to the cue environment was excluded, the Gaussian model was no longer a better fit for correctly ordered vs. shuffled data (correctly ordered data vs. shuffled null excluding cue, $p = 0.848$, $R^2 = -0.006$; **Figure 4c**). This indicates that the significance of the Gaussian was

driven by the cue environment but not by graded similarity to nearby environments in the sequence.

Turning to representations in hippocampus, the asymmetric Gaussian once again provided better fits to the correctly ordered vs. shuffled pattern similarity values when the cue environment was included (correctly ordered data vs. shuffled null including cue, $p = 0.029$, $R^2 = 0.032$). In the hippocampus, similar to visual cortex, the amplitude of the Gaussian fit was significantly higher than the different-map baseline (mean = 0.0185, standard deviation = 0.030; $t(31) = 2.959$, $p = 0.005$; **Figure 4d**) and the asymptote was significantly lower than the different-map baseline (mean = -0.007, standard deviation = 0.014; $t(31) = -3.476$, $p = 0.001$; **Figure 4d**). Thus, hippocampus represented the cue environment while it was on the screen and suppressed environments further away. The backward and forward widths (σ) were 2.313 steps and 1.782 steps, respectively, and were again not significantly different from each other ($V(31) = 231.00$, $p = 0.548$), suggesting that the hippocampus represented forward and backward environments similarly.

Critically, even when the cue environment was excluded, the Gaussian was still a better fit to the correctly ordered vs. shuffled data (correctly ordered data vs. shuffled data excluding cue, $p = 0.023$, $R^2 = 0.028$; **Figure 4d**). This indicates that the hippocampal Gaussian fit shows significantly graded similarity to nearby environments in the past and future, in addition to representing the cued environment. In contrast to hippocampus and visual cortex, the Gaussian fit was not significantly better for the correctly ordered data vs. the shuffled null (including the cue) in the insula ($p = 0.139$, **Supplementary Figure 1**).” (p. 11-13, Results)

“Next, we determined whether the gradedness of sequence representations was specific to the order of the cued path or if it was also present for the uncued path, which contains the same environments but in a different order (see Methods). To investigate this, we repeated our Gaussian analysis with pattern similarity values arranged along the order of the uncued path. If the hippocampus represents the order of surrounding environments in a context-dependent manner, we may not observe statistically reliable Gaussian fits for the uncued path. To test for graded representations of the past and future along the uncued path, we compared the Gaussian fit to the shuffled null excluding the cue environment (because the cue environment was the same for both cued and uncued paths). We also repeated this analysis in visual cortex for completeness, but because visual cortex showed no evidence for graded representations of the past and future for the cued path, we also expected it to show no gradedness for the uncued path.

As expected, the Gaussian fit in visual cortex for the uncued path was not better than the fit for the shuffled null excluding the cue environment ($p = 0.993$, $R^2 = -0.0173$; **Figure 5a**), and the Gaussian fit was not significantly different for the cued path vs the uncued path ($p = 0.132$), indicating that visual cortex did not represent surrounding environments on either the cued or uncued path. In the hippocampus, the Gaussian fit for the uncued path was not significantly better than the fit for the shuffled null excluding the cue environment ($p = 0.915$, $R^2 = -0.012$, **Figure 5b**). Further, the Gaussian fit was significantly better for the order of the cued path compared to the uncued path ($p = 0.005$). Together, these results show that the graded similarity to nearby environments in the past and future in hippocampus was specific to the order of the cued path.” (p. 16-17, Results)

“We obtained a p-value by calculating the fraction of 10,000 shuffles that produced Gaussian fits with lower error than the fit for the correctly ordered data. We also calculated R^2 as 1 minus the sum of squared errors of the Gaussian fit for the correctly ordered data divided by the mean sum of squared errors of the Gaussian fits for the shuffled data. R^2 values greater than 0 indicate smaller squared errors for the correctly ordered data than the shuffled data.” (p. 47-48, Methods)

As per your suggestion, we also investigated whether variation in signal-to-noise and R^2 was confounded with the width of the Gaussian fit. To investigate this, we systematically simulated datasets by adding varying amounts of noise to visual cortex data and assessing changes in the resulting Gaussian fits for each dataset. See response to comment 3 as well.

We added random noise to each voxel’s activity in the visual cortex patterns of activity elicited during the cue screen of the Anticipation Task and the across-participant environment templates elicited during the Localizer Task (random values drawn from a normal distribution centered at 0). To create datasets with different amounts of noise, we varied the standard deviation of this normal distribution from 0.1 to 2 in increments of 0.1. We then ran the same Gaussian analysis as reported in the manuscript on each of these simulated datasets.

The amplitude of the Gaussian fits became successively smaller in datasets with successively more noise. Additionally, R^2 became successively smaller in datasets with successively more noise. Critically, however, the forward and backward sigmas did not become wider in datasets with successively more noise. Instead, as amplitude and R^2 decreased, the forward and backward sigmas remained similar. See the figure below for a visualization of these results. Therefore, our simulations reveal that (1) decreased signal to noise ratio does not explain the variation in Gaussian widths and (2) variation in goodness of fit does not account for changes in Gaussian width, as R^2 and width vary independently in our simulations.

Overall, we have shown that the Gaussian model is a good fit to the data, that the goodness-of-fit is not confounded by region, and that changes in noise and goodness-of-fit do not account for ROI-based differences in the Gaussian widths.

We report the noise simulation in the manuscript and Supplementary Materials as follows:

“We further conducted simulations of BOLD data to confirm the lack of an inherent, systematic relationship between the width and amplitude of the Gaussian fits (see Supplementary Materials, Supplementary Figure 4 for details). Finally, there was no correlation between asymptote and y-coordinate in either PPA ($t(31) = 1.721$, $p = 0.095$) or RSC ($t(31) = 1.047$, $p = 0.303$). Overall, this suggests a within-region hierarchical organization of representations in the visual system, such that more anterior (vs posterior) aspects of PPA and RSC represent environments that are further away in the past and future. Importantly, the partial correlation between width and y-coordinate while controlling for amplitude in RSC, as well as our simulations, suggests that further reaching representations are not necessarily a consequence of reduced processing of the present.” (p. 22, Results)

“Simulation of BOLD Data

We conducted simulations of BOLD data to investigate whether variation in signal-to-noise and R^2 was confounded with the width of the Gaussian fit across regions. To investigate this, we systematically simulated datasets by adding varying amounts of noise to visual cortex data and assessing changes in the resulting Gaussian fits for each dataset.

We added random noise to the visual cortex patterns of activity elicited during the cue screen of the Anticipation Task and to the across-participant environment templates elicited during the Localizer Task (random values drawn from a normal distribution centered at 0). To create datasets with different amounts of noise, we varied the standard deviation of this normal distribution from 0.1 to 2 in increments of 0.1. We then ran the same Gaussian analysis as reported in the manuscript on each of these simulated datasets. We simulated five datasets at each noise level.

The amplitude of the Gaussian fits became successively smaller in datasets with successively more noise (Supplementary Fig 4a and 4d). Additionally, R^2 became successively smaller in datasets with successively more noise (Supplementary Fig 4b). Critically, however, the forward and backward sigmas did not become wider in datasets with successively more noise (Supplementary Fig 4c and 4d). Instead, as amplitude and R^2 decreased, the forward and backward sigmas remained similar. Therefore, our simulations reveal that (1) decreased signal to noise ratio does not explain the variation in Gaussian widths and (2) variation in goodness of fit does not account for changes in Gaussian width, as R^2 and width vary independently in our simulations.” (p. 2-3, Supplementary Materials)

Supplementary Figure 4 | Results of BOLD Data Simulation. The amplitude (a) and R^2 (b) of the Gaussian fit decreased for simulated datasets with successively higher noise levels, but there was no systematic change in the width of the Gaussian fit (c). Black lines indicate the average value across five simulated datasets at each noise level and transparent gray ribbons indicate the 95% confidence interval. Note that the error ribbon is also shown in panel (a) but difficult to detect due to very low variability. (d) Gaussian curve at three sample noise levels.

** The manuscript should more carefully consider the properties of the stories that participants generated during encoding, and how they might affect the subsequent reactivation effects. Story generation is described as: “Participants were instructed to learn the sequences by generating a story to link the environments in order.” There are only two example stories shown in the Supplementary Information but they are quite different. The first example is quite passive, with a person simply being transported from one place to the next, but the second story contains a much richer motivational structure. Indeed, in the second story the participant meets a character (a man in a tophat) who actually accompanies him to subsequent rooms and then later betrays the participant. Could such narrative differences induce differences in the inter-environment reactivation patterns? For example, when the participant enters the pub and meets the monocle character, will they not be reminded of their ultimate betrayal back into prison by that same character? In light of this, it seems

important to understand more about the stories. How often did participants imagine that they themselves were traversing the environment? How often did they generate goals and associate those goals with specific rooms in the loop? Were the goal locations (and the causal connections between locations) equally distributed across rooms? This kind of information seems important to understand the possible basis for the environment-reactivation effects that are described in Figure 4.

Response: Thank you for bringing up these excellent questions for consideration. We agree that there is certainly variation in the stories that participants generated, and these narrative differences could lead to interesting neural differences when anticipating upcoming events in their stories. This is a very interesting area for future research, and we are currently running other studies that score the stories in more detail to allow us to investigate these questions.

Despite the individual differences in the stories, our across-participant analyses were focused on information that was shared across individuals: the environments themselves. Our results suggest that participants were consistently and reliably reactivating environments that were nearby in the past and the future, regardless of differences in their individual narratives.

We have added a paragraph to the discussion about these important questions regarding differences that could arise based on story-specific goals or situations, which we are actively investigating in other work:

“Alternatively, the environment sequences used in our current study may not have engaged the insula as strongly as a continuously unfolding audiovisual movie stimulus that allowed the generation of social or emotional predictions²³.

In our study, individuals were asked to generate narratives to tie together the environments in each sequence to help learning, memory, and prediction. These stories were, however, idiosyncratic, meaning that the content of a given person’s story could have influenced the conceptualization and representation of the environments and the extent to which an individual reactivated past or future states. Our analyses, however, relied on activity patterns for each environment that were obtained from across-participant templates; because the only shared information across people was the environments themselves and not the sequences or the generated stories, this allowed us to test for group-level similarities in the graded activation of environment representations. Thus, our findings show that despite the idiosyncrasies of individual stories, participants were nevertheless still representing the environments in a reliable and consistent way during the anticipation task – a way that was systematic enough that we could detect evidence for bidirectional and graded representations of the future and the past across participants.

Nevertheless, it remains the case that the stories that individuals generated may have differed in their social predictions, goals, and motivations, and this may have influenced our findings. Such narrative differences across individuals may have led to differential involvement of the insula across participants²³, making it difficult for us to replicate long-timescale predictions in this region¹². Future work could assess the participants’ stories in greater detail to determine whether the stories with stronger

social narratives were related to far-reaching predictions in the insula.” (p. 28-29, Discussion)

We think that thoroughly addressing your question, in a way that does justice to the richness of the narrative strategies used by participants, requires in-depth analyses and detailed scoring of individual stories. We therefore think this would be better suited for a stand-alone paper than an addition to the current work, given the focus of the current work on similarities in sequential representations across individuals. We hope to be able to fully answer your question in upcoming papers!

*** It is important to show that the inter-regional [or inter-voxel] variation in the Gaussian widths [as reported in Figure 6 and Figure 4] cannot be explained by gradients of peak pattern similarity (i.e. signal-to-noise). Looking at the pale gray curves in 4C/D and the warm-colored curves in Figure 6C, it seems that wider Gaussians tend to also have lower peaks (i.e. smaller maximum pattern similarity). Because there are various normalizations applied to the BOLD data, and because there may be biases in the Gaussian fit procedure as one varies the signal-to-noise ratio in the data, it is important to show that these do not introduce a systematic relationship between peak-pattern-similarity and Gaussian width. Probably the best way to achieve this is to simulate BOLD data with a fixed set of underlying pattern similarities, then add a fixed amount of noise, then apply the same preprocessing steps and Gaussian-fitting as applied in the manuscript, and demonstrate that the sigma values remain constant even as you vary the strength of the pattern similarities (i.e. the signal-to-noise ratio). Alternatively, you could demonstrate that there is no empirical relationship between the peak similarity and the Gaussian sigma — for example, by plotting a variant of Figure 6A with the peak pattern similarity plotted instead of sigma, and demonstrating that this map lacks any of the gradients shown in Figure 6A. Relatedly, one could also just make a scatter plot of Gaussian-peak-vs-Gaussian-width and show that the two parameters are uncorrelated.*

Response: Thank you for bringing up this very important point. We agree that it is important to determine whether there is a systematic relationship between amplitude and width in our Gaussian fitting procedure that may be caused by variations in signal-to-noise ratios. To investigate this, we systematically simulated datasets by adding varying amounts of noise to visual cortex data and assessing changes in the resulting Gaussian fits for each dataset (as noted in response to your main point #1, and reiterated below for convenience).

To this end, we added random noise to each voxel’s activity in the visual cortex patterns of activity elicited during the cue screen of the Anticipation Task and the across-participant environment templates elicited during the Localizer Task (random values drawn from a normal distribution centered at 0). To create datasets with different amounts of noise, we varied the standard deviation of this normal distribution from 0.1 to 2 in increments of 0.1. We then ran the same Gaussian analysis as reported in the manuscript on each of these simulated datasets.

The amplitude and R^2 of the Gaussian fits became successively smaller in datasets with successively more noise. This confirms that adding noise reduced peak pattern similarity in

our Gaussian fitting procedure. Critically, however, the forward and backward sigmas did not become wider in datasets with successively more noise. Instead, as amplitude and R^2 decreased, the forward and backward sigmas remained similar, and even became slightly narrower compared to datasets with higher amplitude. See the figure below for a visualization of these results. Therefore, our simulations reveal that decreased signal to noise ratio does not explain the variation in Gaussian widths.

Additionally, we reran the correlation between the Gaussian width and y-coordinate when controlling for amplitude as a covariate. We found that the partial correlation between width and y-coordinate with amplitude as a covariate was significant in RSC ($t(31) = 2.050$, $p = 0.049$). However, the partial correlation was not significant in PPA ($t(31) = 0.08$, $p = 0.936$). This suggests that, while the amplitude and width of the Gaussian fit can vary independently and systematically in some brain regions, they are negatively correlated with one another in others, such that more anterior parts of these regions show both a lower amplitude and a larger width. See also Reviewer #3, Comment #8.

The negative correlation between amplitude and width within PPA suggests that there is a trade-off between representing the present versus associated future and past environments, because voxels with broader timescales are also those that are less responsive to the cue. Within RSC, however, the increased timescales in more anterior voxels are not tightly related

to decreases in visual responsiveness. This suggests that there is no inherent conflict between representing the current stimulus and the learned sequence in some brain regions. It further demonstrates that our Gaussian modeling approach can detect independent variation in amplitude and width.

We have made the following changes to the main text to reflect these findings, as described above and pasted here for your convenience:

“There was a significant positive correlation between width (σ) and y-coordinate, indicating that Gaussian fits became progressively wider in progressively more anterior aspects of both RSC ($t(31) = 2.638$, $p = 0.013$; **Figure 7b**) and PPA ($t(31) = 2.424$, $p = 0.021$; **Figure 7c**). There was a negative correlation between amplitude and y-coordinate in PPA ($t(31) = -2.636$, $p = 0.013$; **Figure 7c**), but not RSC ($t(31) = -0.550$, $p = 0.586$; **Figure 7b**). To determine whether the change in width along the posterior-anterior axis was related to the change in amplitude, such that wide Gaussian fits are associated with lower amplitudes, we ran a partial correlation between width and y-coordinate, controlling for amplitude. The negative correlation between width and y-coordinate remained significant in RSC ($t(31) = 2.050$, $p = 0.049$), but not PPA ($t(31) = 0.08$, $p = 0.936$). Thus, the amplitude and width of the Gaussian fit can vary independently in some regions, but are coupled in others. We further conducted simulations of BOLD data to confirm the lack of an inherent, systematic relationship between the width and amplitude of the Gaussian fits (see **Supplementary Materials, Supplementary Figure 4** for details). Finally, there was no correlation between asymptote and y-coordinate in either PPA ($t(31) = 1.721$, $p = 0.095$) or RSC ($t(31) = 1.047$, $p = 0.303$). Overall, this suggests a within-region hierarchical organization of representations in the visual system, such that more anterior (vs posterior) aspects of PPA and RSC represent environments that are further away in the past and future. Importantly, the partial correlation between width and y-coordinate while controlling for amplitude in RSC, as well as our simulations, suggests that further reaching representations are not necessarily a consequence of reduced processing of the present.” (p. 21-22, Results)

We have made the following changes to the Supplementary Materials to reflect these findings, as described above and pasted here for your convenience:

“Simulation of BOLD Data

We conducted simulations of BOLD data to investigate whether variation in signal-to-noise and R^2 was confounded with the width of the Gaussian fit across regions. To investigate this, we systematically simulated datasets by adding varying amounts of noise to visual cortex data and assessing changes in the resulting Gaussian fits for each dataset.

We added random noise to the visual cortex patterns of activity elicited during the cue screen of the Anticipation Task and to the across-participant environment templates elicited during the Localizer Task (random values drawn from a normal distribution centered at 0). To create datasets with different amounts of noise, we varied the standard deviation of this normal distribution from 0.1 to 2 in increments of 0.1. We then ran the same Gaussian analysis as reported in the manuscript on each of these simulated datasets. We simulated five datasets at each noise level.

The amplitude of the Gaussian fits became successively smaller in datasets with successively more noise (Supplementary Fig 4a and 4d). Additionally, R^2 became successively smaller in datasets with successively more noise (Supplementary Fig 4b). Critically, however, the forward and backward sigmas did not become wider in datasets with successively more noise (Supplementary Fig 4c and 4d). Instead, as amplitude and R^2 decreased, the forward and backward sigmas remained similar. Therefore, our simulations reveal that (1) decreased signal to noise ratio does not explain the variation in Gaussian widths and (2) variation in goodness of fit does not account for changes in Gaussian width, as R^2 and width vary independently in our simulations.” (p. 2-3, Supplementary Materials)

Supplementary Figure 4 | Results of BOLD Data Simulation. The amplitude (a) and R^2 (b) of the Gaussian fit decreased for simulated datasets with successively higher noise levels, but there was no systematic change in the width of the Gaussian fit (c). Black lines indicate the average value across five simulated datasets at each noise level and transparent gray ribbons indicate the 95% confidence interval. Note that the error ribbon is also shown in panel (a) but difficult to detect due to very low variability. (d) Gaussian curve at three sample noise levels.

** It is important to show that the negative pattern similarities (described as “suppression” effects) are not induced by preprocessing steps. On page 11, the manuscript describes how

the asymptotes of the Gaussians are interpreted: “The asymptote is an indication of the representations of environments that are not captured by the width of the Gaussian; if the asymptote is lower than baseline (defined as pattern similarity between the cue and all environments from the other map, henceforth referred to as different-map baseline), this suggests that these environments are suppressed.” However, if voxel time courses are normalized to have a zero mean, then when a pattern at time X is positively correlated with a pattern at time Y, then necessarily there will have to be another timepoint Z whose pattern has a negative correlations X and/or Y. This phenomenon is analogous to the “global signal regression” phenomenon in functional connectivity, where the presence of positive functional connectivity between some pairs of nodes necessarily implies negative functional connectivity between other nodes, because of the normalization procedure that precedes the correlation, e.g. Weissenbacher et al., 2009. If the analogous effect is present in these environment-to-environment correlation maps, then one might expect that greater-magnitude positive pattern similarity (at zero / one steps from the cue) will be associated with greater-magnitude negative pattern similarity at other step-numbers. More generally, describing a negative pattern similarity result as “suppression” is not valid unless it can be clearly established that the baseline correlation value is truly zero — it is easy for random patterns to exhibit a non-zero correlation on average with a target vector, depending on how the data are normalized before the correlation is computed. This concern could be ruled out using simulated data subjected to the same preprocessing and normalization steps as used in the text.

Ref: Weissenbacher, A., Kasess, C., Gerstl, F., Lanzenberger, R., Moser, E., & Windischberger, C. (2009). Correlations and anticorrelations in resting-state functional connectivity MRI: a quantitative comparison of preprocessing strategies. Neuroimage, 47(4), 1408-1416.

Response: Thank you for bringing up this point. We also agree that it would be incorrect to describe a negative correlation as indicating suppression without an appropriate baseline. In our Gaussian analysis, we compare the asymptote of the Gaussian fit to the *different-map baseline*, not to zero, to determine whether environments were suppressed. The different-map baseline consists of the pattern similarity values between the activity pattern during the cue screen period and the activity pattern templates of the eight environments that are not present in the cued map. Therefore, suppression of further environments is determined not by comparison to zero, but to pattern similarity with other environments that are irrelevant for the current trial (those from the different map). We have edited the manuscript, as described below, to clarify this analysis.

It is also certainly the case that this would be an issue if we were comparing time points that have been normalized to have a zero mean. However, our analyses compare pattern similarity between all the environment templates from the Localizer Task and a single time point from the Anticipation Task—the cue screen period on a given trial. In this case, there is no mathematical reason for a trade-off in which some positive correlations necessarily lead to some negative correlations. In theory, all of the correlations could be positive: there was no time course or set of values that was normalized to have a zero mean. For example, in anterior PPA (Figure 7c), all of the pattern similarity values (between the cue screen activity

pattern and the relevant environment templates) are indeed positive. We have clarified these points in the manuscript as follows:

“The asymptote is an indication of the representations of environments that are not captured by the width of the Gaussian; if the asymptote is lower than baseline (defined as pattern similarity between the cue screen activity patterns and all environment templates from the other map, henceforth referred to as different-map baseline), this suggests that these environments are suppressed. Importantly, our pattern similarity approach considers independent time points (similarity between our Anticipation Task cue periods and environment templates from the Localizer Task), meaning that more evidence for the cued environment does not necessitate suppression of other environments – suppression need not be observed, and all pattern similarity values could be positive, such as in Figure 7c.” (p. 11-12, Results)

“We also performed statistical tests on the parameters of the correctly ordered Gaussians. We compared the amplitude and asymptote to a baseline which consisted of the pattern similarity values between the cue screen activity pattern and the activity pattern templates for the eight environments that were not in the cued map (different-map baseline). We tested whether the amplitude was significantly above the different-map baseline, and whether the asymptote was suppressed below the different-map baseline, using one-sample t-tests across participants.” (p. 48, Methods)

*** Could the within-region hierarchical gradient phenomenon (Figure 6) reflect a gradient of intrinsic neural timescales across voxels (e.g. Raut et al., 2020)? In other words, suppose that Region X has an “activation response” function that ramps to its peak in 1 second and then decays, while Region Y generates a response that ramps to its peak in 2 seconds and then decays. Even if Region X and Region Y have the same activation peak, Region Y will exhibit a stronger signal-to-noise ratio for its representations of later materials [near its peak] while Region X will exhibit a stronger signal-to-noise ratio for its representation of earlier content. So even if both Region X and Region Y are representing all the same content, differences in the intrinsic timescales of population activity could act as a kind of filter, which magnifies the effective strength of neural states that are earlier or later. One way to rule this out would be to show that measurements of intrinsic timescales [e.g. autocorrelation width, as in Raut et al., 2020] are uncorrelated with the gradients shown in Figure 6.*

*Ref: Raut, R. V., Snyder, A. Z., & Raichle, M. E. (2020). Hierarchical dynamics as a macroscopic organizing principle of the human brain. *Proceedings of the National Academy of Sciences*, 117(34), 20890-20897.*

Response: Thank you for bringing up this point. We agree that intrinsic timescales could be a property of the findings that we report here, where different brain regions have longer temporal receptive windows than others. However, we do not believe that autocorrelation in a brain region’s activity over time should influence pattern similarity between activity at a given time point and independently identified template patterns of activity acquired during a separate task. Nevertheless, we believe our findings are broadly consistent with Raut et al.

(2020) and others. For example, Brunec and Momennejad (2022) showed that autocorrelation decays less quickly in more anterior aspects of hippocampus and PFC during navigation, and this was taken to indicate that anterior brain regions had longer predictive windows than posterior ones. Overall, we believe that our work and the past research on the decay rates of the autocorrelation signal across different brain regions are converging towards the same conclusion: that there are hierarchies of timescales in the brain, with progressively longer timescales in progressively more anterior regions.

Note that, unlike that prior work, we do not examine brain activity over a continuously unfolding event (or a resting state period). In our task, we examined the similarity in activity patterns between a given cue environment and independently identified patterns of activity for other environments. Thus, we think it is intriguing that both our approach and the work on intrinsic timescales converge to the same conclusion about hierarchically organized timescales in the brain. We have added a section to the Discussion to address this:

“Our results **build upon** influential theories of prediction in the brain. Graded coding of upcoming events is consistent with successor representation models^{2,24,44,45}, which propose that information about future states becomes cached into the representation of the current state in a temporally discounted manner. These models have been extended to account for multiple timescales of prediction by incorporating different scales of temporal discounting²⁴. **In line with these theories, recent work has shown that multiple timescales of prediction are represented simultaneously in the brain, with progressively further-reaching predictions in progressively more anterior brain regions¹² and with relatively less evidence for far away vs nearby predictions^{22,46}. Converging with this past work, in our prediction task we found that patterns for nearby vs far away environments (defined in an independent task) were activated at multiple scales in a hierarchical manner across brain regions.** Strikingly, although our asymmetric Gaussian analysis was designed to allow differential coding of the future vs the past, representations were not uniquely biased toward future states. Instead, the hippocampus and visual system represented temporal structure bidirectionally, with graded representations into the past and future.” (p. 24-25, Discussion)

*** Could the observation of bidirectional sequence activation which was found here, but not in some prior studies, be a consequence of the story-based training that was used during the environment decoding? If participants in prior studies were not encouraged to use stories at encoding then perhaps this is another factor that could be mentioned in the Discussion paragraph beginning “An important distinction between our experiment and past studies of prediction ...”. More generally, the manuscript would be stronger if it commented on how the story generation factor is expected to have impacted any of the results. (To what extent is this study even about a sequences of “environments” if there is a self-generated story involved... is the anticipation then not really anticipation of “situations” which happen to have taken place in environments?*

Response: Thank you for the suggestion. We agree that our bidirectional sequence findings may be a consequence of the stories that individuals generated and used during anticipation

of the sequences. Our results, however, suggest that individuals represented a sequence of environments, potentially alongside a sequence of situations. The sequence of situations are *participant-specific*, because each participant generated their own story. However, our analyses relied on activity patterns for each environment that were obtained from *across-participant* analyses; here, the only shared information across people was the environments themselves and not the stories. Therefore, while we agree that the stories influenced participants' strategies, and that participants may indeed have been anticipating particular situations, they were nevertheless still representing the environments in a reliable and consistent way during the anticipation task.

It is also worth noting that, even if individuals were representing a series of situations and not a sequence of discrete environments, our interpretations would not change: we would still be able to conclude that different brain regions represent more or less of anticipated situations or narratives, consistent with our prior work (Lee, Aly, & Baldassano, 2021).

We have expanded on how stories could have influenced individual's strategies and our results in the Discussion:

“Alternatively, the environment sequences used in our current study may not have engaged the insula as strongly as a continuously unfolding audiovisual movie stimulus that allowed the generation of social or emotional predictions²³.

In our study, individuals were asked to generate narratives to tie together the environments in each sequence to help learning, memory, and prediction. These stories were, however, idiosyncratic, meaning that the content of a given person's story could have influenced the conceptualization and representation of the environments and the extent to which an individual reactivated past or future states. Our analyses, however, relied on activity patterns for each environment that were obtained from across-participant templates; because the only shared information across people was the environments themselves and not the sequences or the generated stories, this allowed us to test for group-level similarities in the graded activation of environment representations. Thus, our findings show that despite the idiosyncrasies of individual stories, participants were nevertheless still representing the environments in a reliable and consistent way during the anticipation task – a way that was systematic enough that we could detect evidence for bidirectional and graded representations of the future and the past across participants.” (p. 28-29, Discussion)

SMALLER POINTS

** The manuscript is missing a summary (either at the end of the Intro or start of the Results) of the basic experimental design, which briefly describes the experimental design and logic, and the previewing the names of the task phases. Currently, the Results section (which is located before the Methods in the Nature Communications format) begins describing the outcomes of the Anticipation Task, even though this task and its purpose have not been previously described. Similarly, it could be helpful to provide a one-sentence textual*

description of the Localizer Task when it is first mentioned in the Results. If the Localizer corresponds to the "training data" (Figure 1), this should be made clear in the text, and not left to a figure caption or to the Methods.

Response: Thank you for the suggestion. We agree that a summary of the experimental approach and task names would be useful for understanding our results. We have made the following changes to clarify this in the text:

"In the present study, we investigated how context-specific temporal structure is represented in the brain during a novel multistep anticipation task. Participants learned, in immersive virtual reality, four temporally extended sequences of eight environments each (**Figure 1**). Critically, pairs of sequences (**Green Path vs. Blue Path**) contained the same environments in a different order, requiring individuals to flexibly anticipate environments based on the current sequence context. Sequences were circular, such that environments were temporally predictable multiple steps into the future and the past regardless of location in the sequence. This allowed us to test whether temporal structure in both the prospective and retrospective direction is automatically represented in the brain even if only future states are task-relevant.

Following sequence learning, participants were scanned with fMRI as they **completed an Anticipation Task, in which they anticipated upcoming environments one to four steps into the future in a given (cued) sequence (Figure 2). Following the Anticipation Task, participants completed a Localizer Task in which we obtained template patterns of brain activity for each environment (Figure 3).** We used these templates to conduct multivoxel pattern similarity analyses in visual cortex, hippocampus, insula, and across the brain. Specifically, while participants viewed a given cue environment and attempted to anticipate upcoming environments in the Anticipation Task, we looked for multivoxel evidence of surrounding environments in the sequence. Using this approach, we determined the extent to which temporal structure was (1) represented in a graded and bidirectional manner, with simultaneous representations of future and past environments; (2) represented in a hierarchical fashion among lower and higher order brain regions, with further-reaching representations in higher-order regions; and (3) modulated by context, with prioritized representations for the cued vs uncued sequence." (p. 5-6, Introduction)

** On page 9 the basic method for pattern-matching is described as a between-subjects method: " we calculated pattern similarity between (1) multivoxel patterns of brain activity evoked during the Anticipation Task for each trial type (cue and path combination) for each participant and (2) the multivoxel patterns of brain activity evoked during Localizer Task, averaged across the remaining participants." It seems worth discussing why the pattern match is not computed within-subject. Is there not enough held-out data for this to be feasible? Does the cross-subject method simply generate more consistent results? There are other settings in fMRI where within-subject pattern match seems to perform better than between-subject pattern match (e.g. in single-subject language encoding models from the Gallant lab or Huth lab) and so it's worth commenting on the logic behind this between-subject method.*

Response: Thank you for this suggestion, which was echoed by Reviewer #1 (Comment #3); our response is shared here as well for convenience. We did try to conduct within-participant analyses by creating participant-specific templates for each environment, using the four runs from the Localizer Task. However, these participant-specific templates were noisier than the across-participant templates. Specifically, we conducted the same analyses as described in the *Localizer Task Analyses* section of the manuscript, but using held-out runs of the Localizer Task within a participant. We obtained the within-participant multivoxel pattern of brain activity for each environment within each of our ROIs for a single run of the Localizer Task (e.g. P1, Localizer Task Run 1). We then compared that pattern to the averaged within-participant multivoxel patterns for the remaining runs of the Localizer Task (e.g. P1, Localizer Task Runs 2-4). Akin to Figure 3d, we then compared within-participant activity patterns for the same environment to the activity patterns for different environments within each ROI. However, the resulting within-participant pattern similarity values were noisier than the across-participant analysis: there was not a clear difference in pattern similarity values for the same environment vs different environments. We believe that there are two potential explanations for noisier within- vs across-participant environment patterns: 1) a given individual's template for a given environment could contain predictive information about upcoming environments if they are activating one path more than the other during the Localizer Task, or activating one path sometimes and the other path at other times during the Localizer Task; and 2) we may have been underpowered to detect participant-specific patterns of activity for all 16 environments from just four runs of the Localizer Task.

The across-participant approach we took may have therefore helped us identify hippocampal activity patterns that reflected the features of each environment uncontaminated by any prediction or other forms of participant-specific cognition that may have occurred during the Localizer Task. We think this is a particular strength because 1) the across-participant templates are cleaner measures of individual environments, uncontaminated by future environments, than within-participant templates and therefore offer a stronger test of our hypotheses; and 2) the finding of reliable hippocampal activity patterns for individual environments *across participants* adds to an emerging body of work demonstrating shared hippocampal representations across individuals which previously went undetected (Koch et al., 2020; Chen et al., 2021). One possible reason for our ability to detect across person representations in the hippocampus could be due to the immersiveness of our virtual reality environments during encoding. While speculative, It's possible that participants were more strongly reactivating environment characteristics that were learned in immersive VR compared to environments or images viewed on a desktop computer, which is the standard approach in other cognitive neuroscience studies. We now discuss the implications of these across-participant findings in the Discussion as follows:

“To investigate bidirectional, context-dependent sequence representations, we carefully manipulated overlapping sequences. Pairs of sequences contained the same environments in a different order. They were structured such that, for each environment, the environments one and two steps into the future and the past were all different in the two sequences. We found that prospective and retrospective representations in hippocampus were context-dependent, emerging only for the cued

but not the uncued sequence. This dovetails with prior work showing context-sensitive representations of future events in hippocampus³⁷, and extends this work to graded and bidirectional sequence representations within a context. Strikingly, we observed such effects in hippocampus even with environment templates that were identified across participants. The finding of reliable hippocampal activity patterns for individual environments *across participants* adds to an emerging body of work demonstrating shared hippocampal representations across individuals, representations that were previously difficult to detect^{57,58}.” (p. 27-28, Discussion)

** It could strengthen the manuscript if there was more discussion earlier-on (e.g. in the Introduction) regarding the functional significance of bidirectional representations. On page 4 the manuscript states: “Although this work suggests that the brain may represent both anticipated and past events, these prior studies did not test whether forward and backward representations of temporally extended structure existed simultaneously in the same brain regions.” I think that one or two of the key ideas from the Discussion could be mentioned in the Introduction could help to motivate the interest in the directionality of the anticipatory sequence-representations. Similarly, at the bottom of page 4 the text states: “However, it remains unknown whether contextual modulation of temporal structure representations is specific to planning trajectories in the forward direction or if contextual relevance also modulates representations of the past.” It seems clear why you would want to have forward trajectories that are context-specific (because you want accurate context-specific predictions that can enable you to plan and prepare for action). But it seems worth laying out some of the functional considerations for why one would/would-not want to have context specific backward trajectories. Is it the case that you don’t need to have an accurate backward representation of where you were previously, at least in some specific brain regions? Again, some motivating considerations on these points could strengthen the Introduction.*

Response: Thank you for this suggestion. We have now included a discussion of the utility of bidirectional representations in the Introduction, mirroring the ideas laid out in our Discussion section. We have included the following new sections:

“Although this work suggests that the brain may represent both anticipated and past events, these prior studies did not test whether forward and backward representations of temporally extended structure existed simultaneously in the same brain regions. Such bidirectional representations would accord with temporal context models that propose that events experienced nearby one another come to be associated with similar representations, such that the retrieval of a given item can be a strong cue for both preceding and subsequent items^{35,36}. We therefore examined whether the brain contains bidirectional representations of the past and future, with the scale of these representations varying systematically across the brain.” (p. 4, Introduction)

“For the brain’s representations of temporal structure to be adaptive for behavior, they should flexibly change depending on context. Recent work in humans has shown context-specific patterns of activity in the hippocampus during goal-directed planning of future trajectories, suggesting that anticipation of temporally structured experience is specific to the upcoming items in a given context³⁷. However, it remains unknown

whether contextual modulation of temporal structure representations is specific to planning trajectories in the forward direction or if associations with preceding items are also activated in a context-specific way. If bidirectional representations help disambiguate overlapping contexts – which may overlap in the future, the past, or both – then context-specific representations of both past and future states would be useful in planning trajectories within a context.” (p. 5, Introduction)

** Related to the point above, the Discussion might be strengthened by mentioning any specific theories of cortical prediction which are consistent with (or, better, inconsistent with) the hierarchical and bidirectional sequence reactivation data reported here.*

Response: Thank you for this excellent suggestion. We agree the Discussion would be strengthened by more closely relating our findings to theories of cortical prediction, such as successor representations (Dayan, 1993; Gershman, 2018). Our findings of hierarchical sequence reactivation, such that more anterior regions have further horizons, are consistent with hierarchical successor representation frameworks (Momennejad & Howard, 2018; Brunec & Momennejad, 2022). Interestingly, however, our work also differs from past studies of predictive processing in that we found bidirectional representations of both future *and* past states during anticipation (see also Reviewer 3, Comment #13). There is limited evidence for such bidirectional representations in prior studies. For example, past work has found evidence for separate past and future representations of sequences when participants were explicitly told to think in the forward or the backward direction during retrieval (Wimmer et al., 2020) or when exposed to items that form a sequence without the demand to anticipate future events (Ekman, 2023). Here, we show that such bidirectional representations are present even when individuals were explicitly told to anticipate future events, with no overt demand to simulate past events. Importantly, we go beyond those prior studies in showing that such bidirectional representations are present in numerous visual areas and are organized hierarchically.

Why were both past and future states represented in the brain in our task? This finding is seemingly at odds with theories of predictive processing, such as Successor Representations (Dayan, 1993; Gershman, 2018), which assume that representations should be future-oriented. Instead, our finding of bidirectional representations might be consistent with event segmentation and temporal context models, which suggest that a whole event is brought online during behavior, including other memories nearby in time from the same event (Manning et al., 2011). Therefore, we believe one possible explanation for our findings is the proximity to an event boundary: if cued with the beginning of an event, individuals may exhibit future oriented predictions, but not retrospective representations. But, if cued with the middle of an event, individuals may need to bring online representations of past states to access their memory for the whole event (Michelmann et al, 2023). Because our sequences were circular, every environment was in the “middle”, i.e., was neither at the beginning nor at the end, potentially explaining our finding of bidirectional rather than future-oriented representations. Regardless, bidirectional representations are not predicted by typical successor representation theories of future prediction, and presents an intriguing avenue for future

research to disentangle when bidirectional representations might be present rather than just forward ones.

“Our results build upon influential theories of prediction in the brain. Graded coding of upcoming events is consistent with successor representation models^{2,24,44,45}, which propose that information about future states becomes cached into the representation of the current state in a temporally discounted manner. These models have been extended to account for multiple timescales of prediction by incorporating different scales of temporal discounting²⁴. In line with these theories, recent work has shown that multiple timescales of prediction are represented simultaneously in the brain, with progressively further-reaching predictions in progressively more anterior brain regions¹² and with relatively less evidence for far away vs nearby predictions^{22,46}. Converging with this past work, in our prediction task we found that patterns for nearby vs far away environments (defined in an independent task) were activated at multiple scales in a hierarchical manner across brain regions. Strikingly, although our asymmetric Gaussian analysis was designed to allow differential coding of the future vs the past, representations were not uniquely biased toward future states. Instead, the hippocampus and visual system represented temporal structure bidirectionally, with graded representations into the past and future. This finding is seemingly at odds with successor representation models^{44,47}, which assume that representations should be future-oriented. Taken together with prior work showing that hippocampal representations of temporal sequences^{46,48,49} can be flexibly biased in either the forward or backward direction based on task demands³⁴, our findings suggest that representations of the past and future can exist simultaneously within the hippocampus, even though the task demands were to anticipate future states.

Why were both past and future states represented in the brain in our task? Our finding of bidirectional representations might be consistent with event segmentation and temporal context models, which suggest that a whole event is brought online during behavior, including other memories nearby in time from the same event³⁵. Therefore, we believe one possible explanation for our findings is the proximity to an event boundary: if cued with the beginning of an event, individuals may exhibit future oriented predictions, but not retrospective representations. But, if cued with the middle of an event, individuals may need to bring online representations of past states to access their memory for the whole event⁵⁰. An important distinction between our experiment and past studies of prediction is that our sequences were circular and temporally extended, whereas sequences in prior studies tended to have a clear end point (i.e. were linear instead of circular)^{12,22,37,46,51} or were shorter¹⁹. Because our sequences were circular, environments were neither at the beginning nor at the end of the sequence, potentially explaining our finding of bidirectional rather than future-oriented representations. Thus, our findings present an intriguing avenue for future research to disentangle when bidirectional representations might be present, in line with temporal context models, rather than just forward ones, in line with successor representations.” (p. 24-26, Discussion)

** I was surprised that the hippocampus did not exhibit a context effect [i.e. different representations of the same regions within different Paths]. If the participants told different stories for Path A and Path B, then should we perhaps expect to see context effects in*

regions of the brain that represent "situational" content, such as regions in the default-mode network? Some more discussion of this point could be fruitful.

Response: Thank you for this suggestion. Our previous analyses of context-dependence were limited in examining only one and two steps in the future and past. In response to suggestions by Reviewer #3 to apply a consistent analytic approach throughout the paper, we applied our Gaussian analysis to the order of the uncued path to test for context-dependence in the hippocampus (see response to Reviewer 1, comment 2). In hippocampus, we found that the uncued path Gaussian fit was not significantly better for the correctly ordered data vs. the shuffled null excluding the cue ($p = 0.915$). Additionally, the Gaussian fit was significantly better for the order of the cued path compared to the uncued path (correctly ordered data vs. shuffled null excluding cue, $p = 0.005$). This suggests that graded similarity to nearby environments in the forward and backward direction in hippocampus was context-dependent. We have made the following changes to the results and, as per your suggestion, to the discussion to reflect these new findings:

“Next, we determined whether the gradedness of sequence representations was specific to the order of the cued path or if it was also present for the uncued path, which contains the same environments but in a different order (see Methods). To investigate this, we repeated our Gaussian analysis with pattern similarity values arranged along the order of the uncued path. If the hippocampus represents the order of surrounding environments in a context-dependent manner, we may not observe statistically reliable Gaussian fits for the uncued path. To test for graded representations of the past and future along the uncued path, we compared the Gaussian fit to the shuffled null excluding the cue environment (because the cue environment was the same for both cued and uncued paths). We also repeated this analysis in visual cortex for completeness, but because visual cortex showed no evidence for graded representations of the past and future for the cued path, we also expected it to show no gradedness for the uncued path.

As expected, the Gaussian fit in visual cortex for the uncued path was not better than the fit for the shuffled null excluding the cue environment ($p = 0.993$, $R^2 = -0.0173$; **Figure 5a**), and the Gaussian fit was not significantly different for the cued path vs the uncued path ($p = 0.132$), indicating that visual cortex did not represent surrounding environments on either the cued or uncued path. In the hippocampus, the Gaussian fit for the uncued path was not significantly better than the fit for the shuffled null excluding the cue environment ($p = 0.915$, $R^2 = -0.012$, **Figure 5b**). Further, the Gaussian fit was significantly better for the order of the cued path compared to the uncued path ($p = 0.005$). Together, these results show that the graded similarity to nearby environments in the past and future in hippocampus was specific to the order of the cued path.” (p. 16-17, Results)

“Finally, to test whether the Gaussian representations were context dependent, we repeated the same Gaussian fitting procedure as above but with the pattern similarity values arranged on the order of the uncued path. Importantly, we only tested whether the Gaussian fit for the uncued path was significantly better than the shuffled null excluding the cue because 1) the cue environment was the same across paths and 2) this comparison specifically tests evidence for graded representations of the past and future. We also assessed whether the cued order Gaussian fit was significantly better

than the uncued order Gaussian fit when compared to the shuffled null excluding the cue. This allowed us to specifically test whether there was more evidence for graded representations of the past and future along the cued vs. uncued path in the absence of similarity to the cue environment. We computed the difference between the sum of squared errors for 1) the correctly ordered data along the cued path vs. the shuffled null, excluding the cue from both; and 2) the correctly ordered data along the uncued path vs. the shuffled null, excluding the cue from both. We then tested how often the difference in errors for the cued path was larger than the difference for the uncued path.” (p. 48-49, Methods)

“To investigate bidirectional, context-dependent sequence representations, we carefully manipulated overlapping sequences. Pairs of sequences contained the same environments in a different order. They were structured such that, for each environment, the environments one and two steps into the future and the past were all different in the two sequences. We found that prospective and retrospective representations in hippocampus were context-dependent, emerging only for the cued but not the uncued sequence. This dovetails with prior work showing context-sensitive representations of future events in hippocampus³⁷, and extends this work to graded and bidirectional sequence representations within a context.” (p. 27-28, Discussion)

Reviewer #3 (Remarks to the Author):

This paper is aimed at investigating representations of future and past environments in different brain regions and how these are modulated by context. This is a topic that is currently receiving a lot of attention in the field and there are several related recent papers that have just been published. The current study used a clever design in which participants learned the order of a set of virtual environments and then anticipated future environments in the scanner. By comparing the activity patterns in this anticipation task, to the patterns elicited in other participants during the localization task, it is possible to get an estimate of the representation of future environments that is not contaminated with the learned order of environments, as the order is different across participants. While the experimental setup seems to be very well considered, I am concerned about the analyses that are used to investigate the evidence for representations of future and past environments and the conclusions that are drawn from those. My main concern is about the amount of evidence that there is an anticipation signal in the brain regions that are studied. In the first section of the paper, anticipation is studied by investigating the width of a gaussian fit to the pattern similarities of environment with varying steps into the future and the past. However, there are several issues with this approach, which I detail below. The other important concern is the number of different analyses approaches that are used in the paper in different sections, even though it would have been possible to study all questions within the same basic analysis framework. In addition, some rather weak findings seem to be taken at face value without considering the borderline significance of the results and the conclusions are phrased too strongly in relation to the quality of the evidence.

1. Looking at the raw data in figure 4, there does not seem to be a lot of evidence for representations of future or past environments in visual cortex or hippocampus. The pattern similarities are close to zero. In the hippocampus, the representations of both the current and

future/past environments are very weak. This might be because the analyses rely on across-participant pattern similarity analyses. Improved matching of fine-grained patterns across participants might be necessary to see these effects more clearly, for example by hyperaligning the data first.

Response: Thank you for your comment. We agree that stronger evidence for our graded pattern similarity results would greatly strengthen the paper. It is the case that the pattern similarity values in hippocampus are relatively low. However, in our Gaussian analysis, we were primarily interested in detecting differences in *relative* correlation values – differences that follow the order of the cued path – rather than comparing the absolute value of the correlations to 0. The magnitude of pattern similarity can be affected by multiple factors, including signal-to-noise ratios, how driven a given brain region is by visual stimulation, and how robust across-participant activity patterns are for a given environment (because of the way our template activity patterns were defined). These factors may influence the overall values of the pattern similarity correlations, but the magnitude of these values is not of primary interest – instead, our focus is on *relative* evidence for environments as a function of their distance from the cued environment. If pattern similarity values follow the order of the cued path, we take these ordered correlation values to indicate graded similarity to nearby environments. To test the strength of the relative ordering of correlations, we conducted two new analyses.

Firstly, we created another test of our Gaussian fit to specifically assess whether there were graded representations for future and past environments. To do so, we removed pattern similarity to the cue environment from the correctly ordered and shuffled data and fit the Gaussian model to these no-cue datasets. For the shuffled dataset, therefore, we shuffled the order of pattern similarity values (excluding pattern similarity to the cue environment) 10000 times to create a second baseline condition that specifically tests for gradedness for past and future environments (“shuffled null excluding cue”). Specifically, if the Gaussian model provides a better fit to correctly ordered data vs. shuffled data *even when pattern similarity to the cue environment is removed from both*, that would indicate that there is significant evidence for graded similarity between the activity pattern on a given trial and nearby environments in the sequence. That is, by removing pattern similarity to the cue environment from both the observed data and the shuffled baseline, this analysis allows us to see how much of the original Gaussian fits were due to representation of the cue environment vs. representations of the upcoming and past environments. If a brain region shows a significant Gaussian fit compared to the shuffled null when the cue environment is included but *not* when the cue environment is excluded, that would indicate that its Gaussian fit is entirely driven by its representation of the cue environment. If a brain region shows a significant Gaussian fit compared to the shuffled baseline both when the cue environment is included and when it is excluded, that would indicate that its representations of both the cue and nearby environments contribute to a significant Gaussian fit.

These analyses revealed that the Gaussian fit in visual cortex was no longer significant compared to the shuffled baseline excluding the cue environment from the analysis ($p = 0.848$). This indicates that the significance of the Gaussian in visual cortex was driven by the

cue environment but not by graded similarity to nearby environments. In contrast, the Gaussian fit in hippocampus was still significant compared to the shuffled baseline excluding the cue environment from the analysis ($p = 0.023$), indicating that the hippocampal Gaussian fit shows significantly graded similarity to nearby environments, in addition to representations of the cued environment.

Secondly, we also tested the Gaussian fit when the pattern similarity values were arranged on the order of the uncued path. Importantly, because the uncued path had the same pattern similarity values in a different order, any differences between the cued vs uncued Gaussian fit would be due to the relative order of the correlations on the cued path. Representations in hippocampus were context-dependent: the uncued path Gaussian fit was not significantly better for the correctly ordered data vs. the shuffled null excluding the cue ($p = 0.915$). We then assessed whether the cued path Gaussian fit was significantly better than the uncued path Gaussian fit. The evidence for graded past and future representations along both the cued and uncued paths was measured by comparing each fit to the shuffled null excluding the cue, and then testing whether the difference in fit for the correctly ordered vs shuffled data was larger for the cued vs. uncued path. Indeed, the Gaussian fit was significantly better for the order of the cued path compared to the uncued path (correctly ordered data vs. shuffled null excluding cue, $p = 0.005$). Together, these results show that we did indeed detect graded representations of nearby environments along the order of the cued path (but not the uncued path) in the hippocampus. We believe that these new analyses offer stronger evidence for the graded representations in hippocampus, even if the absolute pattern similarity values are relatively low. We have made the following changes to the manuscript to reflect these new analysis and results:

“We statistically tested the Gaussian fit in two ways. First, we compared the goodness-of-fit for the Gaussian model when applied to the correctly ordered pattern similarity values vs. a shuffled-order version of the pattern similarities, including pattern similarity to the cue environment (“shuffled null including cue”; **Figure 4b**). This allows us to test the null hypothesis that there was no structure in the similarity values. Second, we removed the pattern similarity to the cue environment and fit the Gaussian model only to the pattern similarities for upcoming and past environments, in both the correct order and the shuffled order (“shuffled null excluding cue”; **Figure 4b**). If a brain region shows a superior Gaussian fit for the observed vs. shuffled data only when the cue environment is included but not when it is excluded, that would indicate that its Gaussian fit is entirely driven by its representation of the cue environment. If a brain region shows a superior Gaussian fit for the observed vs. shuffled data both when the cue environment is included and when it is excluded, that would indicate that this region represents the cue environment and also has systematically graded representations of nearby environments.

In visual cortex, the Gaussian model provided a significantly better fit to the correctly ordered vs. shuffled pattern similarity values when the cue environment was included in both the observed and shuffled data (correctly ordered data vs. shuffled null including cue, $p < 0.001$, $R^2 = 0.699$). The amplitudes of participants’ Gaussian fits were significantly higher than the different-map baseline, indicating that the cue environment was represented while it was on the screen (mean = 0.091, standard deviation = 0.029; $t(31) = 18.01$, $p < 0.000001$; **Figure 4c**). The asymptote was

significantly lower than the different-map baseline, suggesting that other environments surrounding the cue were suppressed (mean = -0.014, standard deviation = 0.009; $t(31) = -5.97$, $p = 0.000001$; **Figure 4c**). The backward and forward widths (σ) were 0.712 steps and 0.634 steps, respectively, and were not significantly different from each other ($V(31) = 195.00$, $p = 0.203$), suggesting that representations were not biased toward one direction over the other. However, when pattern similarity to the cue environment was excluded, the Gaussian model was no longer a better fit for correctly ordered vs. shuffled data (correctly ordered data vs. shuffled null excluding cue, $p = 0.848$, $R^2 = -0.006$; **Figure 4c**). This indicates that the significance of the Gaussian was driven by the cue environment but not by graded similarity to nearby environments in the sequence.

Turning to representations in hippocampus, the asymmetric Gaussian once again provided better fits to the correctly ordered vs. shuffled pattern similarity values when the cue environment was included (correctly ordered data vs. shuffled null including cue, $p = 0.029$, $R^2 = 0.032$). In the hippocampus, similar to visual cortex, the amplitude of the Gaussian fit was significantly higher than the different-map baseline (mean = 0.0185, standard deviation = 0.030; $t(31) = 2.959$, $p = 0.005$; **Figure 4d**) and the asymptote was significantly lower than the different-map baseline (mean = -0.007, standard deviation = 0.014; $t(31) = -3.476$, $p = 0.001$; **Figure 4d**). Thus, hippocampus represented the cue environment while it was on the screen and suppressed environments further away. The backward and forward widths (σ) were 2.313 steps and 1.782 steps, respectively, and were again not significantly different from each other ($V(31) = 231.00$, $p = 0.548$), suggesting that the hippocampus represented forward and backward environments similarly. Critically, even when the cue environment was excluded, the Gaussian was still a better fit to the correctly ordered vs. shuffled data (correctly ordered data vs. shuffled data excluding cue, $p = 0.023$, $R^2 = 0.028$; **Figure 4d**). This indicates that the hippocampal Gaussian fit shows significantly graded similarity to nearby environments in the past and future, in addition to representing the cued environment. In contrast to hippocampus and visual cortex, the Gaussian fit was not significantly better for the correctly ordered data vs. the shuffled null (including the cue) in the insula ($p = 0.139$, **Supplementary Figure 1**).” (p. 11-13, Results)

“Next, we determined whether the gradedness of sequence representations was specific to the order of the cued path or if it was also present for the uncued path, which contains the same environments but in a different order (see Methods). To investigate this, we repeated our Gaussian analysis with pattern similarity values arranged along the order of the uncued path. If the hippocampus represents the order of surrounding environments in a context-dependent manner, we may not observe statistically reliable Gaussian fits for the uncued path. To test for graded representations of the past and future along the uncued path, we compared the Gaussian fit to the shuffled null excluding the cue environment (because the cue environment was the same for both cued and uncued paths). We also repeated this analysis in visual cortex for completeness, but because visual cortex showed no evidence for graded representations of the past and future for the cued path, we also expected it to show no gradedness for the uncued path.

As expected, the Gaussian fit in visual cortex for the uncued path was not better than the fit for the shuffled null excluding the cue environment ($p = 0.993$, $R^2 = -0.0173$; **Figure 5a**), and the Gaussian fit was not significantly different for the cued path vs the

uncued path ($p = 0.132$), indicating that visual cortex did not represent surrounding environments on either the cued or uncued path. In the hippocampus, the Gaussian fit for the uncued path was not significantly better than the fit for the shuffled null excluding the cue environment ($p = 0.915$, $R^2 = -0.012$, **Figure 5b**). Further, the Gaussian fit was significantly better for the order of the cued path compared to the uncued path ($p = 0.005$). Together, these results show that the graded similarity to nearby environments in the past and future in hippocampus was specific to the order of the cued path.” (p. 16-17, Results)

“To test whether the parameters of the Gaussian curve were consistent across participants, we fit the asymmetrical Gaussian curve on all but one participant’s data (e.g. P2-32) and then measured the sum of squared errors (observed vs. predicted pattern similarity values) when using this curve to predict the held-out participant’s data (e.g. P1). We repeated this procedure for each choice of held-out participant to obtain an average error value. We conducted two tests. First, we compared the Gaussian fit to correctly ordered data and the Gaussian fit to shuffled pattern similarity values, including pattern similarity to the cue environment (“shuffled null including cue”). Second, we conducted the same comparison but removed pattern similarity to the cue environment from both the correctly ordered and shuffled data (“shuffled null excluding cue”). In both cases, we fit the Gaussian to shuffled pattern similarity values 10000 times and obtained the goodness of fit each time to create a null distribution. The goodness of fit for the correctly ordered data vs. null distributions were then compared. If, for a given brain region, the Gaussian model provides a better fit to the correctly ordered vs. shuffled data both when the cue is included and when it is excluded, that would indicate that the brain region’s representations of both the cue and graded representations of nearby environments contribute to a significant Gaussian fit. If, however, the Gaussian model provides a better fit to the correctly ordered vs. shuffled data only when the cue is included and not when it is excluded, that would indicate that its Gaussian fit was driven by its representation of the cue environment without any evidence for graded representations of the past or future.

We obtained a p-value by calculating the fraction of 10,000 shuffles that produced Gaussian fits with lower error than the fit for the correctly ordered data. We also calculated R^2 as 1 minus the sum of squared errors of the Gaussian fit for the correctly ordered data divided by the mean sum of squared errors of the Gaussian fits for the shuffled data. R^2 values greater than 0 indicate smaller squared errors for the correctly ordered data than the shuffled data.” (p. 47-48, Methods)

“Finally, to test whether the Gaussian representations were context dependent, we repeated the same Gaussian fitting procedure as above but with the pattern similarity values arranged on the order of the uncued path. Importantly, we only tested whether the Gaussian fit for the uncued path was significantly better than the shuffled null excluding the cue because 1) the cue environment was the same across paths and 2) this comparison specifically tests evidence for graded representations of the past and future. We also assessed whether the cued order Gaussian fit was significantly better than the uncued order Gaussian fit when compared to the shuffled null excluding the cue. This allowed us to specifically test whether there was more evidence for graded representations of the past and future along the cued vs. uncued path in the absence of similarity to the cue environment. We computed the difference between the sum of squared errors for 1) the correctly ordered data along the cued path vs. the shuffled null, excluding the cue from both; and 2) the correctly ordered data along the uncued

path vs. the shuffled null, excluding the cue from both. We then tested how often the difference in errors for the cued path was larger than the difference for the uncued path.” (p. 48-49, Methods)

We also appreciate your suggestion to hyperalign the data to improve our across-participant analyses. Unfortunately, hyperalignment requires a relatively long and naturalistic stimulus, such as a movie (Haxby et al., 2020). The trial-wise design in the current experiment does not make it a good candidate for hyperalignment. Nevertheless, we believe the above analyses now provide stronger evidence for across-participant graded representations of sequences in hippocampus.

Additionally, we did try to conduct within-participant analyses and created participant-specific templates for each environment by using the four runs from the Localizer Task. This suggestion was also made by the other two Reviewers, and we include our response here as well for convenience. We found that individual-participant environment templates were noisier than the across-participant environment templates. Specifically, we conducted the same analyses as described in the *Localizer Task Analyses* section of the manuscript, but using held-out runs of the Localizer Task within a participant. We obtained the within-participant multivoxel pattern of brain activity for each environment within each of our ROIs for a single run of the Localizer Task (e.g. P1, Localizer Task Run 1). We then compared that pattern to the averaged within-participant multivoxel patterns for the remaining runs of the Localizer Task (e.g. P1, Localizer Task Runs 2-4). Akin to Figure 3d, we then compared within-participant activity patterns for the same environment to the activity patterns for different environments within each ROI. However, the resulting within-participant pattern similarity values were noisier than the across-participant analysis: there was not a clear difference in pattern similarity values for the same environment vs different environments. We believe that there are two potential explanations for noisier within- vs across-participant environment patterns: 1) a given individual’s template for a given environment could contain predictive information about upcoming environments if they are activating one path more than the other during the Localizer Task, or activating one path sometimes and the other path at other times during the Localizer Task; and 2) we may have been underpowered to detect participant-specific patterns of activity for all 16 environments from just four runs of the Localizer Task. Therefore, in our data, the across-participant analysis was the stronger approach to find significant representations of environments.

2. The pattern similarity between participants for one and two steps in the past and future provide a very clear and straightforward metric that can be used to directly test the evidence for representation of future and past environments. However, in the first section of the paper, the authors chose a more convoluted approach to look at this question, which involves fitting a gaussian similarity model and then investigating the parameters of this model. It is not clear why this approach is needed and the investigation of these parameters is not straightforward. Currently, the authors test the gaussian fit in 3 ways: 1. comparing the full model against a shuffled environment baseline to test if the parameters are consistent across participants 2. comparing the amplitude and asymptote of the gaussian fit with a

different-environment baseline and 3. comparing the width estimates across different brain regions. Below I detail the limitations of the overarching approach and for each of these tests.

a. Test 1 does not provide any information about which parameters of the model are driving the consistency across participants. Most likely, it is the amplitude the gaussian that is driving the observed consistency. Therefore, it is not very informative about representation of future and past environments.

Response: Thank you for bringing up this point. To test whether the significance of the Gaussian model was driven only by the representation of the cue environment or also included representations of surrounding environments in the future and past, we conducted a new analysis in which we removed the cue environment from both the correctly ordered data and the shuffled null. Comparing the Gaussian fit for the correctly ordered data to the Gaussian fit to the shuffled null, excluding the cue environment from both datasets, allows us to test the significance of graded similarity to nearby environments (see response to Comment #1). In visual cortex, we found that the Gaussian fit for the correctly ordered data was no longer significantly different than the fit to the shuffled null when excluding the cue environment from the analysis ($p = 0.848$). This suggests that the visual cortex Gaussian fit was primarily driven by the representation of the cue environment. In the hippocampus, however, the Gaussian fit remained significantly better for the correctly ordered data vs. the shuffled null when excluding the cue environment from the analysis ($p = 0.023$). This indicates that the hippocampal Gaussian fit shows significantly graded similarity to nearby environments, in addition to representing the cued environment. We believe that this is stronger evidence to suggest that the significant Gaussian fits in hippocampus were driven by representations of future and past environments, in addition to representing the current moment. We have added these new null models to the manuscript as follows:

“To test whether the parameters of the Gaussian curve were consistent across participants, we fit the asymmetrical Gaussian curve on all but one participant’s data (e.g. P2-32) and then measured the sum of squared errors (observed vs. predicted pattern similarity values) when using this curve to predict the held-out participant’s data (e.g. P1). We repeated this procedure for each choice of held-out participant to obtain an average error value. We conducted two tests. First, we compared the Gaussian fit to correctly ordered data and the Gaussian fit to shuffled pattern similarity values, including pattern similarity to the cue environment (“shuffled null including cue”). Second, we conducted the same comparison but removed pattern similarity to the cue environment from both the correctly ordered and shuffled data (“shuffled null excluding cue”). In both cases, we fit the Gaussian to shuffled pattern similarity values 10000 times and obtained the goodness of fit each time to create a null distribution. The goodness of fit for the correctly ordered data vs. null distributions were then compared. If, for a given brain region, the Gaussian model provides a better fit to the correctly ordered vs. shuffled data both when the cue is included and when it is excluded, that would indicate that the brain region’s representations of both the cue and graded representations of nearby environments contribute to a significant Gaussian fit. If, however, the Gaussian model provides a better fit to the correctly ordered vs. shuffled data only when the cue is included and not when it is excluded, that would indicate that its Gaussian fit was driven by its representation of the cue environment without any evidence for graded representations of the past or future.

We obtained a p-value by calculating the fraction of 10,000 shuffles that produced Gaussian fits with lower error than the fit for the correctly ordered data. We also calculated R^2 as 1 minus the sum of squared errors of the Gaussian fit for the correctly ordered data divided by the mean sum of squared errors of the Gaussian fits for the shuffled data. R^2 values greater than 0 indicate smaller squared errors for the correctly ordered data than the shuffled data.” (p. 47-48, Methods)

b. Test 2 compared the amplitude and asymptote of the gaussian fit to the shuffled environment baseline but does not investigate the width of the gaussian and therefore cannot say anything about representation of future and past environments. It is not clear why the width was not compared to the shuffled environment baseline. I would imagine that directly comparing the width to the shuffled environment baseline would allow for a direct test of whether there are more representations of future and past environments than expected by chance.

Response: Thank you for this suggestion. We compare the amplitude and asymptote to the other-map baseline because all three of these parameters are pattern similarity values (y-axis values), making their comparison feasible. Unfortunately it is not feasible to compare the width to the other-map baseline because the width parameter is on a different scale (steps into the future; x-axis value). However, we agree that it is important to test whether there are more future and past representations than would be expected by chance. We believe that the new analyses in which we removed the cue environment from the correctly ordered and shuffled data (see above response) now provides strong evidence that the hippocampus did indeed represent future and past environments in a graded manner.

c. In test 3, the width of the fits of different brain regions are compared. However, this means that for the brain region with the narrowest width (the visual ROI), representation of future and past states cannot be tested.

Response: Thank you for this comment. We agree that our previous analyses did not specifically test whether there were robust representations of future and past states, because the significance of the Gaussian model could have been driven by the representation of the cue environment. To resolve this, we tested the gradedness of visual cortex representations with an analysis in which we removed the cue environment from the correctly ordered and shuffled data (see above, “shuffled null excluding cue”). If there are representations of past and future states that follow the order of the cued path, we would still expect the Gaussian fit in visual cortex to be significant when we remove the cue environment. Instead, the Gaussian fit was not better for the correctly ordered visual cortex data vs. shuffled data excluding the cue ($p = 0.848$), indicating that there was no evidence for representations of future or past states in this region. In light of your suggestion, and the results from these new analyses, we have made sure to carefully revise the text in our manuscript to avoid implying that there are representations of past and future states in visual cortex.

“However, when pattern similarity to the cue environment was excluded, the Gaussian model was no longer a better fit for correctly ordered vs. shuffled data (correctly ordered data vs. shuffled null excluding cue, $p = 0.848$, $R^2 = -0.006$; **Figure 4c**). This indicates that the significance of the Gaussian was driven by the cue environment but not by graded similarity to nearby environments in the sequence.”
(p. 12-13, Results)

“Unlike hippocampus, we were unable to detect bidirectional temporal structure representations in visual cortex. This result is seemingly in contrast to a rich literature showing predictive representations in early visual regions¹¹. One possibility for why we were unable to detect predictions in early visual cortex is due to the nature of our stimuli. Past work decoding predictions from early visual regions has tended to use relatively simple stimuli such as gratings^{13,14}, fractals¹⁶, or specific spatial locations⁵⁹. In contrast, we used rich, naturalistic scenes that were experienced in immersive virtual reality from multiple viewpoints, making it unlikely that participants were generating predictions of low-level visual features tied to specific retinotopic locations. Instead, individuals in our study may have been predicting whole scenes at a relatively coarse (vs detailed) level, leading to the predictive representations observed in higher order scene-specific visual regions such as PPA and RSC³⁹.” (p. 28, Discussion)

d. A bigger issue with test 3 is that is that the estimates of width might be affected by the estimates of the other parameters. The parameters of the Gaussian similarity model are fit simultaneously and are therefore not independent. I also expect that data with lower signal to noise will tend to have a lower amplitude and a wider gaussian fit. This would mean that the differences between hippocampus and the visual ROI, do not reflect increased anticipation, but decreased signal in the hippocampus. Indeed, visual inspection of the results in hippocampus suggests that there is not much evidence for representation of future and past states in that ROI. Simulations could be used to establish whether this is an issue. For example, you might simulate data with a given width and amplitude of representation and vary the signal-to-noise ratio. The question would be whether the estimate of the width co-varies with the signal-to-noise ratio. I am aware that the current draft of the manuscript already gives one example where there was a significant change in width, but not amplitude (in RSC), but I don't think the absence of a significant effect in a single case is sufficient evidence. A more systematic exploration with simulations would be necessary.

Response: Thank you for bringing up this excellent point. We agree that it is critical to test whether the width of the Gaussian fit is affected by variation in signal-to-noise ratio. To investigate this, we systematically simulated datasets by adding varying amounts of noise to visual cortex data and assessing changes in the resulting Gaussian fits for each dataset. These analyses were reported in response to a similar comment by Reviewer #2; we include our response here as well for convenience.

To conduct this simulation, we added random noise to each voxel's activity in the visual cortex patterns of activity elicited during the cue screen of the Anticipation Task and the across-participant environment templates elicited during the Localizer Task (random values drawn from a normal distribution centered at 0). To create datasets with different amounts of

noise, we varied the standard deviation of this normal distribution from 0.1 to 2 in increments of 0.1. We then ran the same Gaussian analysis as reported in the manuscript on each of these simulated datasets.

The amplitude of the Gaussian fits became successively smaller in datasets with successively more noise. Additionally, R^2 became successively smaller in datasets with successively more noise. Critically, however, the forward and backward sigmas did not become wider in datasets with successively more noise. Instead, as amplitude and R^2 decreased, the forward and backward sigmas remained similar. See the figure below for a visualization of these results. Therefore, our simulations reveal that (1) decreased signal to noise ratio does not explain the variation in Gaussian widths and (2) variation in goodness of fit does not account for changes in Gaussian width, as R^2 and width vary independently in our simulations. We now report these simulations in the Results section and the Supplementary Materials as follows:

“Simulation of BOLD Data

We conducted simulations of BOLD data to investigate whether variation in signal-to-noise and R^2 was confounded with the width of the Gaussian fit across regions. To investigate this, we systematically simulated datasets by adding varying amounts of noise to visual cortex data and assessing changes in the resulting Gaussian fits for each dataset.

We added random noise to the visual cortex patterns of activity elicited during the cue screen of the Anticipation Task and to the across-participant environment templates elicited during the Localizer Task (random values drawn from a normal distribution centered at 0). To create datasets with different amounts of noise, we varied the standard deviation of this normal distribution from 0.1 to 2 in increments of 0.1. We then ran the same Gaussian analysis as reported in the manuscript on each of these simulated datasets. We simulated five datasets at each noise level.

The amplitude of the Gaussian fits became successively smaller in datasets with successively more noise (Supplementary Fig 4a and 4d). Additionally, R^2 became successively smaller in datasets with successively more noise (Supplementary Fig 4b). Critically, however, the forward and backward sigmas did not become wider in datasets with successively more noise (Supplementary Fig 4c and 4d). Instead, as amplitude and R^2 decreased, the forward and backward sigmas remained similar. Therefore, our simulations reveal that (1) decreased signal to noise ratio does not explain the variation in Gaussian widths and (2) variation in goodness of fit does not account for changes in Gaussian width, as R^2 and width vary independently in our simulations.” (p. 2-3, Supplementary Materials)

Supplementary Figure 4 | Results of BOLD Data Simulation. The amplitude (a) and R^2 (b) of the Gaussian fit decreased for simulated datasets with successively higher noise levels, but there was no systematic change in the width of the Gaussian fit (c). Black lines indicate the average value across five simulated datasets at each noise level and transparent gray ribbons indicate the 95% confidence interval. Note that the error ribbon is also shown in panel (a) but difficult to detect due to very low variability. (d) Gaussian curve at three sample noise levels.

We have made the following changes to the main text to reflect these findings:

“There was a significant positive correlation between width (σ) and y-coordinate, indicating that Gaussian fits became progressively wider in progressively more anterior aspects of both RSC ($t(31) = 2.638$, $p = 0.013$; **Figure 7b**) and PPA ($t(31) = 2.424$, $p = 0.021$; **Figure 7c**). There was a negative correlation between amplitude and y-coordinate in PPA ($t(31) = -2.636$, $p = 0.013$; **Figure 7c**), but not RSC ($t(31) = -0.550$, $p = 0.586$; **Figure 7b**). To determine whether the change in width along the posterior-anterior axis was related to the change in amplitude, such that wide Gaussian fits are associated with lower amplitudes, we ran a partial correlation between width and y-coordinate, controlling for amplitude. The negative correlation between width and y-coordinate remained significant in RSC ($t(31) = 2.050$, $p = 0.049$), but not PPA ($t(31) = 0.08$, $p = 0.936$). Thus, the amplitude and width of the Gaussian fit can vary independently in some regions, but are coupled in others. We further conducted simulations of BOLD data to confirm the lack of an inherent, systematic

relationship between the width and amplitude of the Gaussian fits (see Supplementary Materials, Supplementary Figure 4 for details). Finally, there was no correlation between asymptote and y-coordinate in either PPA ($t(31) = 1.721$, $p = 0.095$) or RSC ($t(31) = 1.047$, $p = 0.303$). Overall, this suggests a within-region hierarchical organization of representations in the visual system, such that more anterior (vs posterior) aspects of PPA and RSC represent environments that are further away in the past and future. Importantly, the partial correlation between width and y-coordinate while controlling for amplitude in RSC, as well as our simulations, suggests that further reaching representations are not necessarily a consequence of reduced processing of the present.” (p. 21-22, Results)

3. A related point to the previous is that analysis approaches are quite different per subsection of the results. This gives the impression that the analyses were selected based on what provided the optimal results for each sub question. I will shortly list the three main approaches here: 1. To study the evidence for anticipation in different brain regions, the analyses are based on the gaussian fit described above. 2. To study context dependent predictions during the cue representation, one and two steps in the future/past are taken together in the analyses to compare backward and forward representations. 3. To study context dependent predictions during the blank screen, one and two steps in the future are analysed separately (see figure 7), but only the results for one step are reported. It is not clear why the analyses were set up so differently in each case.

Response: Thank you for raising this point, which made us realize the importance of unifying our analysis approach to avoid giving the impression that we selected the approach based on whether optimal results were obtained. Given this comment and comments elsewhere, we decided to focus our analysis on the Gaussian approach, which allowed us to address whether there was evidence for: i) graded representations; ii) context-specific representations; and iii) suppression of far-away environments with a single analytic approach. As noted elsewhere, our addition of the shuffled null excluding the cue environment and the application of the Gaussian analysis to the uncued path and to the blank screen period (see response to Reviewer #1's Comment #2 and your Comment #10) made it unnecessary to include the other analysis approaches that were initially in the paper. The revision now focuses on the Gaussian analysis and we have removed the other methods. Thank you for encouraging us to take a more streamlined approach!

4. Given all the issues listed in point 2 and the discrepancies I described in point 3, I would suggest that the authors use a consistent approach throughout the manuscript. In all analyses, the pattern similarity could be investigated, separately for 1 and 2 steps, without fitting a gaussian. Then the effect of direction, step-distance and context could all be based on these dependent variables. To test whether there is anticipation of future/past environments, it would be most straightforward to see whether the pattern similarity for steps 1 and 2 is significantly higher than zero (or higher than the average over the baseline timepoints (-4,-3,3 & 4)).

Response: Thank you for this suggestion. We agree that using a consistent analysis approach would indeed simplify and streamline the paper, as noted in our response above. We ultimately decided to streamline our approach by keeping the Gaussian analysis as the primary analyses in the paper and removing the analyses testing for context-dependent representations one and two steps into the future and the past. We opted to keep the Gaussian analyses because there are a few limitations with only investigating pattern similarity as a function of step, distance, and context. Firstly, this approach would limit us to looking at only two steps into the past and future, because the two paths were only designed to be completely non-overlapping for one and two steps into the future. For example, an environment 3 steps away in one path would be 1 step away in the other path; thus, tests of context-dependence at 3 steps and 1 steps would not be independent. The Gaussian analysis bypasses this limitation by looking at the *relative ordering* of pattern similarity values, along the order of the cued vs. uncued paths, rather than comparing the magnitude of the correlations. The Gaussian analysis revealed that the widths in many brain areas of interest, specifically in hippocampus, are greater than two steps, thus showing that the approach of looking at the magnitude of pattern similarity values for one and two steps in the future and past would have reduced our ability to find this evidence for past and future states on longer timescales. Similarly, because of this limitation of the pattern similarity approach for only one and two steps in the future, we would not have been able to test for suppression of further environments (more than two steps away). Secondly, the Gaussian fitting approach is a holistic approach to detect graded and bidirectional representations on multiple timescales. This allows us to look for prediction and retrospection with a single model. If we were to test the same questions comparing pattern similarity across steps, direction, and context, we would need individual tests for pattern similarity at each step, which would lead to multiple comparisons problems. Overall, for these reasons and because the other reviewers saw value and innovation in our Gaussian fitting approach, we opted to keep it as the primary analysis in the paper and to instead remove the other analyses.

However, as discussed in our response to your Comment #1, we believe that we have now provided stronger evidence for graded and context-dependent representations in hippocampus using our Gaussian analysis approach by 1) implementing the shuffled null excluding the cue environment and 2) running the Gaussian analysis on the order of the uncued path. By implementing these changes, we are now able to address all of the questions we posed in our initial manuscript but with a single analysis approach – removing the need for the pattern similarity tests at each step in the future. As discussed in response your Comment #1 and shared here again for convenience, we have made the following additions to our manuscript and removed the other analyses as follows:

“We statistically tested the Gaussian fit in two ways. First, we compared the goodness-of-fit for the Gaussian model when applied to the correctly ordered pattern similarity values vs. a shuffled-order version of the pattern similarities, including pattern similarity to the cue environment (“shuffled null including cue”; **Figure 4b**). This allows us to test the null hypothesis that there was no structure in the similarity values. Second, we removed the pattern similarity to the cue environment and fit the Gaussian model only to the pattern similarities for upcoming and past environments, in both the correct order and the shuffled order (“shuffled null excluding cue”; **Figure 4b**). If a

brain region shows a superior Gaussian fit for the observed vs. shuffled data only when the cue environment is included but not when it is excluded, that would indicate that its Gaussian fit is entirely driven by its representation of the cue environment. If a brain region shows a superior Gaussian fit for the observed vs. shuffled data both when the cue environment is included and when it is excluded, that would indicate that this region represents the cue environment and also has systematically graded representations of nearby environments.

In visual cortex, the Gaussian model provided a significantly better fit to the correctly ordered vs. shuffled pattern similarity values when the cue environment was included in both the observed and shuffled data (correctly ordered data vs. shuffled null including cue, $p < 0.001$, $R^2 = 0.699$). The amplitudes of participants' Gaussian fits were significantly higher than the different-map baseline, indicating that the cue environment was represented while it was on the screen (mean = 0.091, standard deviation = 0.029; $t(31) = 18.01$, $p < 0.000001$; **Figure 4c**). The asymptote was significantly lower than the different-map baseline, suggesting that other environments surrounding the cue were suppressed (mean = -0.014, standard deviation = 0.009; $t(31) = -5.97$, $p = 0.000001$; **Figure 4c**). The backward and forward widths (σ) were 0.712 steps and 0.634 steps, respectively, and were not significantly different from each other ($V(31) = 195.00$, $p = 0.203$), suggesting that representations were not biased toward one direction over the other. However, when pattern similarity to the cue environment was excluded, the Gaussian model was no longer a better fit for correctly ordered vs. shuffled data (correctly ordered data vs. shuffled null excluding cue, $p = 0.848$, $R^2 = -0.006$; **Figure 4c**). This indicates that the significance of the Gaussian was driven by the cue environment but not by graded similarity to nearby environments in the sequence.

Turning to representations in hippocampus, the asymmetric Gaussian once again provided better fits to the correctly ordered vs. shuffled pattern similarity values when the cue environment was included (correctly ordered data vs. shuffled null including cue, $p = 0.029$, $R^2 = 0.032$). In the hippocampus, similar to visual cortex, the amplitude of the Gaussian fit was significantly higher than the different-map baseline (mean = 0.0185, standard deviation = 0.030; $t(31) = 2.959$, $p = 0.005$; **Figure 4d**) and the asymptote was significantly lower than the different-map baseline (mean = -0.007, standard deviation = 0.014; $t(31) = -3.476$, $p = 0.001$; **Figure 4d**). Thus, hippocampus represented the cue environment while it was on the screen and suppressed environments further away. The backward and forward widths (σ) were 2.313 steps and 1.782 steps, respectively, and were again not significantly different from each other ($V(31) = 231.00$, $p = 0.548$), suggesting that the hippocampus represented forward and backward environments similarly.

Critically, even when the cue environment was excluded, the Gaussian was still a better fit to the correctly ordered vs. shuffled data (correctly ordered data vs. shuffled data excluding cue, $p = 0.023$, $R^2 = 0.028$; **Figure 4d**). This indicates that the hippocampal Gaussian fit shows significantly graded similarity to nearby environments in the past and future, in addition to representing the cued environment. In contrast to hippocampus and visual cortex, the Gaussian fit was not significantly better for the correctly ordered data vs. the shuffled null (including the cue) in the insula ($p = 0.139$, **Supplementary Figure 1**).” (p. 11-13, Results)

“Next, we determined whether the gradedness of sequence representations was specific to the order of the cued path or if it was also present for the uncued path,

which contains the same environments but in a different order (see Methods). To investigate this, we repeated our Gaussian analysis with pattern similarity values arranged along the order of the uncued path. If the hippocampus represents the order of surrounding environments in a context-dependent manner, we may not observe statistically reliable Gaussian fits for the uncued path. To test for graded representations of the past and future along the uncued path, we compared the Gaussian fit to the shuffled null excluding the cue environment (because the cue environment was the same for both cued and uncued paths). We also repeated this analysis in visual cortex for completeness, but because visual cortex showed no evidence for graded representations of the past and future for the cued path, we also expected it to show no gradedness for the uncued path.

As expected, the Gaussian fit in visual cortex for the uncued path was not better than the fit for the shuffled null excluding the cue environment ($p = 0.993$, $R^2 = -0.0173$; **Figure 5a**), and the Gaussian fit was not significantly different for the cued path vs the uncued path ($p = 0.132$), indicating that visual cortex did not represent surrounding environments on either the cued or uncued path. In the hippocampus, the Gaussian fit for the uncued path was not significantly better than the fit for the shuffled null excluding the cue environment ($p = 0.915$, $R^2 = -0.012$, **Figure 5b**). Further, the Gaussian fit was significantly better for the order of the cued path compared to the uncued path ($p = 0.005$). Together, these results show that the graded similarity to nearby environments in the past and future in hippocampus was specific to the order of the cued path.” (p. 16-17, Results)

“To test whether the parameters of the Gaussian curve were consistent across participants, we fit the asymmetrical Gaussian curve on all but one participant’s data (e.g. P2-32) and then measured the sum of squared errors (observed vs. predicted pattern similarity values) when using this curve to predict the held-out participant’s data (e.g. P1). We repeated this procedure for each choice of held-out participant to obtain an average error value. We conducted two tests. First, we compared the Gaussian fit to correctly ordered data and the Gaussian fit to shuffled pattern similarity values, including pattern similarity to the cue environment (“shuffled null including cue”). Second, we conducted the same comparison but removed pattern similarity to the cue environment from both the correctly ordered and shuffled data (“shuffled null excluding cue”). In both cases, we fit the Gaussian to shuffled pattern similarity values 10000 times and obtained the goodness of fit each time to create a null distribution. The goodness of fit for the correctly ordered data vs. null distributions were then compared. If, for a given brain region, the Gaussian model provides a better fit to the correctly ordered vs. shuffled data both when the cue is included and when it is excluded, that would indicate that the brain region’s representations of both the cue and graded representations of nearby environments contribute to a significant Gaussian fit. If, however, the Gaussian model provides a better fit to the correctly ordered vs. shuffled data only when the cue is included and not when it is excluded, that would indicate that its Gaussian fit was driven by its representation of the cue environment without any evidence for graded representations of the past or future.” (p. 47, Methods)

“Finally, to test whether the Gaussian representations were context dependent, we repeated the same Gaussian fitting procedure as above but with the pattern similarity values arranged on the order of the uncued path. Importantly, we only tested whether the Gaussian fit for the uncued path was significantly better than the shuffled null

excluding the cue because 1) the cue environment was the same across paths and 2) this comparison specifically tests evidence for graded representations of the past and future. We also assessed whether the cued order Gaussian fit was significantly better than the uncued order Gaussian fit when compared to the shuffled null excluding the cue. This allowed us to specifically test whether there was more evidence for graded representations of the past and future along the cued vs. uncued path in the absence of similarity to the cue environment. We computed the difference between the sum of squared errors for 1) the correctly ordered data along the cued path vs. the shuffled null, excluding the cue from both; and 2) the correctly ordered data along the uncued path vs. the shuffled null, excluding the cue from both. We then tested how often the difference in errors for the cued path was larger than the difference for the uncued path.” (p. 48-49, Methods)

5. In the hippocampus there was a significant negative correlation between the asymptote of the Gaussian curve and response time slope. However, the p-value for this association was close to 0.05, suggesting that the evidence for this effect is very weak. This should be considered in the interpretation of this result. It is also not clear why the association with the asymptote was investigated but not with the width. I would expect that especially the width of the gaussian fit would relate to the speed of anticipation of nearby timepoints.

Response: We agree that it is important to address the strength of the correlation between asymptote and response time slope, and also discuss the limitation of interpreting individual differences analyses with relatively small sample sizes. We have added a section to the Discussion below to address the limitations of interpreting this correlation given its strength and our sample size.

As suggested, we also examined the correlation between the width of the Gaussian fit and response time slope. There were no significant relationships between forward and backward Gaussian widths and response time or accuracy on the Anticipation Task in any brain region. However, this null result should be interpreted with the caveat that it is difficult to detect individual differences with relatively small sample sizes (see also response to Reviewer #1, Comment #5). Therefore, it is possible that a relationship does exist between response time and width of the Gaussian fit, but we were unable to detect it because we were underpowered or lacked the variability in our data to examine such a relationship. We now mention this null effect in the paper, alongside the caveat that null effects – as well as the significant effect with the asymptote – should not be over-interpreted given the relatively small sample size for individual differences analyses.

“We further tested whether there was a relationship between response time slopes and width of the Gaussian fit, such that narrower widths are associated with steeper response time slopes. There was no relationship between either forward or backward width and response time slope in visual cortex (Forward Width: $\rho = -0.149$, $p = 0.415$; Backward Width: $\rho = 0.071$, $p = 0.698$) or hippocampus (Forward Width: $\rho = 0.233$, $p = 0.199$; Backward Width: $\rho = -0.008$, $p = 0.966$).” (p. 19-20, Results)

“We determined whether an individual’s asymptote from their Gaussian model, indicating suppression of environments not captured by the Gaussian’s width, was related to response time costs for further environments. We also tested whether an individual’s forward and backward width from their Gaussian model was related to response time costs. Response time costs were quantified with participant-specific regressions that predicted response time as a function of steps into the future. We then performed an individual differences analysis by obtaining the Spearman rank-order correlation between participants’ response time costs and their asymptotes in (1) hippocampus and (2) visual cortex. We repeated this analysis for the forward and backward width instead of the asymptote.” (p. 51, Methods)

“Indeed, hippocampal suppression of distant environments was related to response times costs for anticipating further events. This highlights a trade-off between prioritizing nearby events and being able to quickly respond to upcoming events further in the future. However, it is important to note that this individual-differences correlation was observed with a relatively small sample size, requiring replication in future work. Future work could also test whether the widths of brain regions’ predictive horizons influence behavioral performance in studies specifically powered for individual differences analyses.” (p. 26, Discussion)

6. The reasoning for selecting specifically the PPA and RSC to look at the association between width and y-coordinate is unclear. Based on previous work, I would expect to see a hierarchy of prediction (i.e. more anticipation in more anterior/higher-level areas) across the entire (visual) cortex. Looking at the results in figure 6a, that pattern does not seem to be present more generally.

Response: Thank you for this suggestion. Our decision to focus on within-region hierarchical gradients in PPA and RSC was motivated by our prior work demonstrating that posterior vs anterior parts of PPA are differentially connected to brain regions involved in perception vs memory (Baldassano et al., 2013). Echoing this finding, other recent work has similarly demonstrated that posterior parts of both PPA and RSC are selectively activated during perception, whereas anterior parts of these regions are activated during memory (Silson et al., 2019; Steel et al., 2021). Based on this prior work, we reasoned that posterior PPA and RSC may represent more of the current moment (indicated by higher amplitudes in our Gaussian fitting procedure) whereas anterior PPA and RSC may exhibit more representation of past and future states (indicated by wider Gaussian fits). Therefore, we investigated the relationship between posterior-anterior axis and the Gaussian parameters—specifically amplitude and width—in PPA and RSC. We have clarified this reasoning in the manuscript as follows:

“Next, we conducted an exploratory analysis to test whether temporal structure was represented hierarchically across searchlights that exhibited significant Gaussian fits. We decided to focus our analysis on PPA and RSC, as prior work has shown within-region functional differences in posterior vs anterior parts of these visual regions^{40–42}. Specifically, posterior aspects of these regions may play a larger role in scene perception while anterior aspects may represent scene memories. Based on these differences, we hypothesized that there may be hierarchical representations of temporal structure within PPA and RSC, with further reaching representations (as

indicated by wider vs. narrower widths (σ) in successively more anterior aspects of these regions.” (p. 21, Results)

However, we also agree with you that, based on other past work, there may additionally be an across-region predictive hierarchy in visual regions (Himberger et al., 2018; Lee et al., 2021). We investigated this possibility by testing whether width increased in successively more anterior regions across the entire visual cortex. Specifically, we obtained the average width of the Gaussian fit for each participant separately for V1, V2, V3, V4, and PPA. We then ordered these regions based on their position along the visual hierarchy, with earlier regions (V1) first and the later regions (PPA) last. For each participant we then obtained the Spearman’s correlation between average width and order along the visual hierarchy across regions. Finally, we compared these correlations to 0 using a t test. We found that, indeed, the average width of the Gaussian fit increased across the visual cortex, with successively larger widths in successively more anterior visual cortex regions ($t(31) = 2.136$, $p = 0.041$). This correlation remained significant when only considering V1, V2, V3, and V4 ($t(31) = 4.322$, $p = 0.0001$). This shows that more anterior (vs. posterior) visual regions have a wider Gaussian fit, indicating further-reaching representations of past and future states, consistent with prior work on cortical representational hierarchies (Himberger et al., 2018). Taken together with the within-region hierarchies in PPA and RSC, this suggests that representational hierarchies may exist simultaneously within and across regions. We now report these new findings in the Results and Methods section, and add an interpretation in the Discussion section, as follows:

“We additionally tested whether there were across-region differences in width along the visual hierarchy. We hypothesized that Gaussian fits would be successively wider in later visual cortex regions, compared to earlier ones^{12,43}. We computed the average width of the Gaussian fit across all voxels separately for V1, V2, V3, V4, and PPA for each participant, and then obtained the correlation between 1) average width of the Gaussian fit in a region and 2) that region’s order along the visual hierarchy. We then tested whether these correlations were different from 0 across participants. We found a significant positive correlation, indicating that successively later visual cortex regions had successively wider Gaussian fits ($t(31) = 2.136$, $p = 0.041$). This correlation remained significant when only considering V1, V2, V3, and V4 ($t(31) = 4.322$, $p = 0.0001$). Taken together, this suggests that past and future states were represented hierarchically both within and across regions in visual cortex.” (p. 22, Results)

“Finally, we determined whether there was a timescale hierarchy across regions in visual cortex, with average widths becoming increasingly wider in later visual cortex regions. We first computed the average width across voxels separately for V1, V2, V3, V4, and PPA for each participant. We then ordered each visual region based on their location along the visual hierarchy (V1-V4, then PPA). For each participant, we then obtained the Spearman rank-order correlation between the average forward and backward widths (σ) and the region’s location along the visual hierarchy. We determined whether the correlation was significant at the group level by comparing the participant-specific r values to 0 using a one-sample t-test. A significantly positive r

value would indicate that Gaussian curves become increasingly wider in successively later visual cortex regions.” (p. 50-51, Methods)

“Representations of temporal structure extended beyond hippocampus. In an exploratory whole-brain searchlight analysis, we found representations of temporal structure across the visual system, including PPA and RSC, regions that play an important role in spatial cognition³⁹. Both PPA and RSC represented the cued environment but also represented the temporal structure of surrounding environments in the sequence in both the forward and backward direction. Our findings therefore extend prior work showing that PPA responses can be modulated by temporal context⁵² and prior contextual associations more generally⁵³⁻⁵⁶. Notably, our findings go beyond this prior work by showing a gradual progression of sequence coding within PPA and RSC, with progressively more anterior regions representing more of the future and past and less of the present. This is broadly consistent with prior work suggesting a posterior vs. anterior division within PPA, with posterior aspects playing a larger role in scene perception and anterior aspects playing a larger role in scene memory⁴⁰⁻⁴². This within-region hierarchy was complemented by an across-region hierarchy, with regions higher up the visual hierarchy, such as V4 or PPA, representing further states into the past and future than regions earlier in the visual hierarchy, such as V1. Thus, we show that, within a context, visual regions may balance representations of perception and memory, gradually incorporating less information from perception and more information about learned temporal structure along a posterior to anterior hierarchy.” (p. 27, Discussion)

7. For the RSC and PPA, only the fits are shown and not the data. I think it's important to also show the data.

Response: Thank you for this suggestion. We agree that it is important to see the Gaussian for each participant in RSC and PPA. We have created a supplementary figure that shows the participant-specific Gaussian curves for posterior, mid, and anterior PPA and RSC. We have updated the Supplementary Materials as follows:

Supplementary Figure 3 | Participant-specific Gaussian Curves of Sample Voxels in PPA and RSC. Gaussian fit from an exploratory searchlight in PPA (a) and RSC (b) for three randomly selected voxels in posterior (left, red), mid (middle, orange), and anterior (right, yellow) parts of both regions. Gray lines indicate participant-specific Gaussian curves and solid colored lines indicate the average Gaussian fit across participants.

8. Related to point 2d above, I wonder if there is a correlation between the RSC and PPA y-coordinate and the width of the gaussian fit if the amplitude is used a covariate (i.e. is there an independent association?).

Response: We agree that it would be very interesting to look at the correlation between width and y-coordinate when controlling for amplitude as a covariate. We reported this analysis for Reviewer #2 and also mentioned it in passing in response to your Comment #2D; we share it here again for convenience.

We found that the partial correlation between width and y-coordinate with amplitude as a covariate was significant in RSC ($t(31) = 2.050, p = 0.049$). However, the partial correlation was not significant in PPA ($t(31) = 0.08, p = 0.936$). This suggests that, while the amplitude and width of the Gaussian fit can vary independently and systematically in some brain regions,

they are negatively correlated with one another in others, such that more anterior parts of these regions show both a lower amplitude and a larger width.

The negative correlation between amplitude and width within PPA suggests that there is a trade-off between representing the present versus associated future and past environments, because voxels with broader timescales are also those that are less responsive to the cue. Within RSC, however, the increased timescales in more anterior voxels are not tightly related to decreases in visual responsiveness. This suggests that there is no inherent conflict between representing the current stimulus and the learned sequence in some brain regions (and in our Gaussian modeling approach).

Overall, we show that amplitude and width are sometimes, but not always, correlated with one another – as also shown in our simulations (in response to your Comment #2d). This suggests that regions may vary in how much they trade-off between representing the current moment vs future predictions. We have updated the Methods and Results section with these new analysis and the Discussion section with more detail on our interpretation as follows:

“There was a significant positive correlation between width (σ) and y-coordinate, indicating that Gaussian fits became progressively wider in progressively more anterior aspects of both RSC ($t(31) = 2.638$, $p = 0.013$; **Figure 7b**) and PPA ($t(31) = 2.424$, $p = 0.021$; **Figure 7c**). There was a negative correlation between amplitude and y-coordinate in PPA ($t(31) = -2.636$, $p = 0.013$; **Figure 7c**), but not RSC ($t(31) = -0.550$, $p = 0.586$; **Figure 7b**). To determine whether the change in width along the posterior-anterior axis was related to the change in amplitude, such that wide Gaussian fits are associated with lower amplitudes, we ran a partial correlation between width and y-coordinate, controlling for amplitude. The negative correlation between width and y-coordinate remained significant in RSC ($t(31) = 2.050$, $p = 0.049$), but not PPA ($t(31) = 0.08$, $p = 0.936$). Thus, the amplitude and width of the Gaussian fit can vary independently in some regions, but are coupled in others. We further conducted simulations of BOLD data to confirm the lack of an inherent, systematic relationship between the width and amplitude of the Gaussian fits (see **Supplementary Materials, Supplementary Figure 4** for details). Finally, there was no correlation between asymptote and y-coordinate in either PPA ($t(31) = 1.721$, $p = 0.095$) or RSC ($t(31) = 1.047$, $p = 0.303$). Overall, this suggests a within-region hierarchical organization of representations in the visual system, such that more anterior (vs posterior) aspects of PPA and RSC represent environments that are further away in the past and future. Importantly, the partial correlation between width and y-coordinate while controlling for amplitude in RSC, as well as our simulations, suggests that further reaching representations are not necessarily a consequence of reduced processing of the present.” (p. 21-22, Results)

“Additionally, we repeated the same analysis as above assessing the correlation between width and y-coordinate while controlling for the amplitude of the Gaussian fit.” (p. 50, Methods)

9. It is not clear what the theoretical reason was for selecting the insula as an a priori ROI. The ROI also seems to be covered by relatively few reliable voxels and the voxels that do show a reliable response seem to be only in the anterior insula.

Response: Thank you for this comment. We realize that we may not have been clear about our a priori ROI selection. In our prior work investigating predictive hierarchies across the brain during movie watching (Lee, Aly, & Baldassano, 2021), we found that the insula predicted further into the future than any other brain region. Therefore, in this study, we wanted to determine whether such long-timescale predictions in the insula would also be observed in the context of sequence representations and an overt prediction task, as opposed to a movie. We were not able to find evidence for long-timescale predictions in the insula, however, because the insula data were not well fit by a Gaussian model, precluding any assessment of the Gaussian's parameters. We think the insula may not have been well-fit by the Gaussian for a number of reasons, including: i) the relative paucity of environment-sensitive voxels in the insula (see response to your Comment #18); and ii) the difference in stimulus from this study vs. our last study (Lee et al., 2021) – our previous study used a continuously unfolding audiovisual movie stimulus that allowed the generation of social or emotional predictions, which may drive activity in the insula (Singer et al., 2009) more strongly than the environment sequences used in our current study.

We have updated the text to reflect this as follows:

“For example, during repeated viewing of a movie clip, posterior regions like visual cortex primarily represent the current moment, while progressively more anterior regions represent upcoming events successively further into the future¹². This past work has also highlighted the insula as a brain region that shows particularly far-reaching predictions¹² perhaps due to its role in generating social predictions during naturalistic events²³. These findings of multistep anticipatory signals are generally consistent with computational theories that the brain builds models of the world that cache temporal information about successive events, with different predictive timescales in different brain regions (i.e., multi-scale successor representations^{22,24}).” (p. 3-4, Introduction)

“We were also unable to find evidence for long-timescale predictions in the insula, unlike our prior work investigating predictive hierarchies across the brain during movie watching¹². The relative paucity of environment-sensitive voxels in the insula may have hurt our ability to detect sequence representations in this region. Alternatively, the environment sequences used in our current study may not have engaged the insula as strongly as a continuously unfolding audiovisual movie stimulus that allowed the generation of social or emotional predictions²³.” (p. 28, Discussion)

10. For the visual ROI, context-dependent anticipation is more clearly visible during the blank screen period. Therefore, I would think that it also makes sense to run the initial analyses that test for anticipation in hippocampus and visual cortex, on this blank screen period. That might give more convincing evidence for representation of future and past states.

Response: Thank you for this comment. We agree it would be informative to run our Gaussian analyses in hippocampus and visual cortex during the blank screen period. This was also requested by Reviewer #1 (Comment #2); our response is shared here again for convenience.

The Gaussian fit in hippocampus during the blank screen was not significantly better for the correctly ordered data vs. shuffled data including the cue ($p = 0.604$, $R^2 = -0.003$) or shuffled data excluding the cue ($p = 0.570$, $R^2 = -0.002$). There was a trend for the Gaussian fit in visual cortex during the blank screen to be better for the correctly ordered data vs. shuffled data including the cue ($p = 0.063$, $R^2 = 0.049$), although this was driven by a negative amplitude indicating suppression of the cue environment (amplitude = -0.012). The Gaussian fit in visual cortex during the blank screen was not significantly better for the correctly ordered data vs. shuffled data excluding the cue ($p = 0.7178$, $R^2 = -0.005$). Together, this suggests that the systematic representations of temporal structure were specific to the cue screen period. However, the lack of significant findings during the blank screen period should be interpreted with caution because the multivoxel patterns of activity were obtained from GLMs that included the cue screen and the blank screen periods in a single model, with different regressors. Brain activity triggered by the cue period that persisted into the blank period may have been accounted for by the regressor for the cue period, complicating interpretation of the blank period. Indeed, pattern similarity to the cue environment was negative during the blank screen period, as indicated by negative amplitudes in the Gaussian fit for both visual cortex and hippocampus (visual cortex amplitude = -0.012 , hippocampus amplitude = -0.008). This is consistent with the interpretation that cue-triggered responses were explained by the cue regressor and led to an anticorrelation with the blank period. Therefore, it's unclear whether hippocampus did not contain sequence representations during the blank screen or whether the results were confounded with the cue screen results; for this reason, we did not include the blank screen Gaussian analysis in the main manuscript.

However, we also appreciate your request to provide stronger evidence for representations of past and future states. As described in our response to earlier comments, we conducted a new analysis in which we removed pattern similarity to the cue environment from both the correctly ordered data and the shuffled data. This allowed us to specifically test the significance of graded similarity to past and future environments (see response to Comment #1). In visual cortex, we found that the Gaussian fit was not significantly better for the correctly ordered data vs. shuffled data excluding the cue ($p = 0.848$), indicating that the visual cortex Gaussian fit was primarily driven by the representation of the cue environment rather than representations of nearby environments. In the hippocampus, however, the Gaussian fit was significantly better for the correctly ordered data vs. shuffled data excluding the cue ($p = 0.023$), indicating that the hippocampal Gaussian fit shows significantly graded similarity to past and future states, in addition to representing the cued environment. We believe that this is stronger evidence to suggest that the significant Gaussian fits in hippocampus were driven by representations of future and past environments, in addition to representing the current moment. These results are reported in the manuscript as follows, as discussed in response to your earlier comments and repeated here for your convenience:

“We statistically tested the Gaussian fit in two ways. First, we compared the goodness-of-fit for the Gaussian model when applied to the correctly ordered pattern similarity values vs. a shuffled-order version of the pattern similarities, including pattern similarity to the cue environment (“shuffled null including cue”; **Figure 4b**). This allows us to test the null hypothesis that there was no structure in the similarity values. Second, we removed the pattern similarity to the cue environment and fit the Gaussian model only to the pattern similarities for upcoming and past environments, in both the correct order and the shuffled order (“shuffled null excluding cue”; **Figure 4b**). If a brain region shows a superior Gaussian fit for the observed vs. shuffled data only when the cue environment is included but not when it is excluded, that would indicate that its Gaussian fit is entirely driven by its representation of the cue environment. If a brain region shows a superior Gaussian fit for the observed vs. shuffled data both when the cue environment is included and when it is excluded, that would indicate that this region represents the cue environment and also has systematically graded representations of nearby environments.

In visual cortex, the Gaussian model provided a significantly better fit to the correctly ordered vs. shuffled pattern similarity values when the cue environment was included in both the observed and shuffled data (correctly ordered data vs. shuffled null including cue, $p < 0.001$, $R^2 = 0.699$). The amplitudes of participants’ Gaussian fits were significantly higher than the different-map baseline, indicating that the cue environment was represented while it was on the screen (mean = 0.091, standard deviation = 0.029; $t(31) = 18.01$, $p < 0.000001$; **Figure 4c**). The asymptote was significantly lower than the different-map baseline, suggesting that other environments surrounding the cue were suppressed (mean = -0.014, standard deviation = 0.009; $t(31) = -5.97$, $p = 0.000001$; **Figure 4c**). The backward and forward widths (σ) were 0.712 steps and 0.634 steps, respectively, and were not significantly different from each other ($V(31) = 195.00$, $p = 0.203$), suggesting that representations were not biased toward one direction over the other. However, when pattern similarity to the cue environment was excluded, the Gaussian model was no longer a better fit for correctly ordered vs. shuffled data (correctly ordered data vs. shuffled null excluding cue, $p = 0.848$, $R^2 = -0.006$; **Figure 4c**). This indicates that the significance of the Gaussian was driven by the cue environment but not by graded similarity to nearby environments in the sequence.

Turning to representations in hippocampus, the asymmetric Gaussian once again provided better fits to the correctly ordered vs. shuffled pattern similarity values when the cue environment was included (correctly ordered data vs. shuffled null including cue, $p = 0.029$, $R^2 = 0.032$). In the hippocampus, similar to visual cortex, the amplitude of the Gaussian fit was significantly higher than the different-map baseline (mean = 0.0185, standard deviation = 0.030; $t(31) = 2.959$, $p = 0.005$; **Figure 4d**) and the asymptote was significantly lower than the different-map baseline (mean = -0.007, standard deviation = 0.014; $t(31) = -3.476$, $p = 0.001$; **Figure 4d**). Thus, hippocampus represented the cue environment while it was on the screen and suppressed environments further away. The backward and forward widths (σ) were 2.313 steps and 1.782 steps, respectively, and were again not significantly different from each other ($V(31) = 231.00$, $p = 0.548$), suggesting that the hippocampus represented forward and backward environments similarly. Critically, even when the cue environment was excluded, the Gaussian was still a better fit to the correctly ordered vs. shuffled data (correctly ordered data vs. shuffled data excluding cue, $p = 0.023$, $R^2 = 0.028$; **Figure 4d**). This indicates that the hippocampal Gaussian fit shows significantly graded similarity to nearby environments in the past and future, in addition to representing the

cued environment. In contrast to hippocampus and visual cortex, the Gaussian fit was not significantly better for the correctly ordered data vs. the shuffled null (including the cue) in the insula ($p = 0.139$, **Supplementary Figure 1**.)” (p. 11-13, Results)

“To test whether the parameters of the Gaussian curve were consistent across participants, we fit the asymmetrical Gaussian curve on all but one participant’s data (e.g. P2-32) and then measured the sum of squared errors (observed vs. predicted pattern similarity values) when using this curve to predict the held-out participant’s data (e.g. P1). We repeated this procedure for each choice of held-out participant to obtain an average error value. We conducted two tests. First, we compared the Gaussian fit to correctly ordered data and the Gaussian fit to shuffled pattern similarity values, including pattern similarity to the cue environment (“shuffled null including cue”). Second, we conducted the same comparison but removed pattern similarity to the cue environment from both the correctly ordered and shuffled data (“shuffled null excluding cue”). In both cases, we fit the Gaussian to shuffled pattern similarity values 10000 times and obtained the goodness of fit each time to create a null distribution. The goodness of fit for the correctly ordered data vs. null distributions were then compared. If, for a given brain region, the Gaussian model provides a better fit to the correctly ordered vs. shuffled data both when the cue is included and when it is excluded, that would indicate that the brain region’s representations of both the cue and graded representations of nearby environments contribute to a significant Gaussian fit. If, however, the Gaussian model provides a better fit to the correctly ordered vs. shuffled data only when the cue is included and not when it is excluded, that would indicate that its Gaussian fit was driven by its representation of the cue environment without any evidence for graded representations of the past or future.” (p. 47, Methods)

11. The evidence for context-dependent predictions in visual cortex is very limited, the p-value is only just below 0.05 uncorrected. This should be acknowledged explicitly, and any conclusions based on this result should be phrased with caution.

Response: Thank you for bringing up this important point. We agree that these visual cortex findings were weak. As per your suggestion above to streamline our analyses, we have removed the pattern similarity analyses that focused on context-dependence one and two steps in the future. We have focused our manuscript on the Gaussian analyses presented in Figure 4. These analyses – with the additions and revisions suggested by the reviewers – allowed us to focus on a single approach that holistically assessed evidence for the cue environment, graded representations of past and future environments, and context dependence. We have therefore removed the pattern similarity analyses of context-dependence one and two steps in the future, which showed only a weak effect in visual cortex, as mentioned by the reviewer. However, we have acknowledged that past work has found stronger predictive representations in visual cortex, whereas our study did not find such evidence; we address reasons for this discrepancy in the Discussion as follows:

“Unlike hippocampus, we were unable to detect bidirectional temporal structure representations in visual cortex. This result is seemingly in contrast to a rich literature showing predictive representations in early visual regions¹¹. One possibility for why we were unable to detect predictions in early visual cortex is due to the nature of our

stimuli. Past work decoding predictions from early visual regions has tended to use relatively simple stimuli such as gratings^{13,14}, fractals¹⁶, or specific spatial locations⁵⁹. In contrast, we used rich, naturalistic scenes that were experienced in immersive virtual reality from multiple viewpoints, making it unlikely that participants were generating predictions of low-level visual features tied to specific retinotopic locations. Instead, individuals in our study may have been predicting whole scenes at a relatively coarse (vs detailed) level, leading to the predictive representations observed in higher order scene-specific visual regions such as PPA and RSC³⁹.” (p. 28, Discussion)

12. If the insula was an a priori ROI, why was it not used as an ROI in the analyses of figure 4?

Response: Thank you for this comment. We ran the Gaussian analysis in the insula but did not find a significant Gaussian fit compared to the shuffled null including the cue. Therefore, we did not proceed with the investigation of the model parameters as the overall Gaussian was not reliable. We have now included a figure of the Gaussian fit in hippocampus in Supplementary Materials for interested readers:

“In contrast to hippocampus and visual cortex, the Gaussian fit was not significantly better for the correctly ordered data vs. the shuffled null (including the cue) in the insula ($p = 0.139$, see **Supplementary Figure 1**).” (p. 13, Results)

Supplementary Figure 1 | Gaussian Curve in Insula. The Gaussian curve in the insula was not a significantly better fit for the correctly ordered data vs. the shuffled null including the cue, indicating a lack of temporal structure representations in this region. Points indicate average pattern similarity at each step from the cue and error bars indicate standard error of the mean. Thick, colored line indicates the average Gaussian fit across participants, with the red end of

the rainbow scale indicating higher pattern similarity and the purple end indicating lower pattern similarity. Thin gray lines indicate participant specific Gaussian fits.

13. The results in this manuscript are quite different from what would be expected based on previous work, as far as I have seen. I think the manuscript would benefit from a more thorough discussion about similarities and differences both with respect to the study setup and outcomes. For example, regarding the finding that environments further away from the current environment are suppressed.

Response: Thank you for this excellent suggestion. We agree that the manuscript would be strengthened by a discussion of similarities and differences between our work and past work.

As you mentioned, one way in which our results diverge from past work is the finding that further environments in the sequence are suppressed. To our knowledge, prior work has not looked at suppression of far away environments in a sequence during prediction of upcoming events. For example, past work using movie watching (Lee, Aly & Baldassano, 2021) or navigation (Brunec & Momennejad, 2022) has found successively less evidence for further away events, but has not compared those far-away event patterns to a baseline to show suppression. It's possible that, in this previous work, suppression of further events was present but went undetected. Another possibility for such suppression is the structure of the overlapping paths in the current study: individuals may have suppressed further environments on the cued path if they were coming up sooner on the uncued path. For example, 3 steps into the future on the cued path was the same environment as 1 step into the future on the uncued path. Thus, suppressing environments that were far away in the cued path but nearby in the uncued path may have been useful in avoiding confusion between the overlapping paths, and contributed to the findings observed here. We have added a section to the Discussion to reflect this:

“In addition to representing nearby environments in the past and future, we also found that the hippocampus suppressed more distant environments, showing deactivation of these environments' patterns relative to an unrelated-environment baseline. To our knowledge, prior work has not looked at suppression of far away environments in a sequence during prediction of upcoming events^{12,22}. It is possible that, in this previous work, suppression of further events was present but went undetected. Another possibility for such suppression is a result of the overlapping paths in the current study: individuals may have suppressed further environments on the cued path if they were coming up sooner on the uncued path. Thus, suppressing environments that were far away in the cued path but nearby in the uncued path may have been useful in avoiding confusion between the overlapping paths, and contributed to the findings observed here. Although suppressing distant environments can be beneficial for responding to imminent events, it can also lead to behavioral costs. For example, if a more distant environment appears as a probe, its representation may have to be reactivated, and this reactivation will take more time if it was initially suppressed – leading to response time costs. Indeed, hippocampal suppression of distant environments was related to response times costs for anticipating further events. This highlights a trade-off between prioritizing nearby events and being able to quickly

respond to upcoming events further in the future. However, it is important to note that this individual-differences correlation was observed with a relatively small sample size, requiring replication in future work. Future work could also test whether the widths of brain regions' predictive horizons influence behavioral performance in studies specifically powered for individual differences analyses.” (p. 26, Discussion)

Another difference between our work and prior studies is the insula finding. As mentioned in our response to your Comment #9, in our prior work investigating predictive hierarchies across the brain during movie watching (Lee, Aly, & Baldassano, 2021), we found that the insula predicted further into the future than any other brain region. Therefore, in this study, we wanted to determine whether such long-timescale predictions in the insula would also be observed in the context of sequence representations and an overt prediction task, as opposed to a movie. We were not able to find evidence for long-timescale predictions in the insula, however, because the insula data were not well fit by a Gaussian model, precluding any assessment of the Gaussian's parameters. We think the insula may not have been well-fit by the Gaussian for a number of reasons, including: i) the relative paucity of environment-sensitive voxels in the insula (see response to your Comment #18); and ii) the difference in stimulus from this study vs. our last study (Lee et al., 2021) – our previous study used a continuously unfolding audiovisual movie stimulus that allowed the generation of social or emotional predictions, which may drive activity in the insula (Singer et al., 2009) more strongly than the environment sequences used in our current study.

Additionally, we failed to find strong representations of future states in early visual cortex (V1-4), even during the blank screen period, in contrast to a rich literature showing predictive representations in early visual regions (de Lange et al., 2018). One possibility for why we were unable to detect predictions in early visual cortex is due to the nature of our stimuli. While we used rich, naturalistic scenes, past work decoding predictions from early visual regions has tended to use simpler stimuli such as line gratings (Kok et al., 2012; Kok et al., 2014), fractals (Hindy et al., 2016), or specific spatial locations (Favila & Aly, 2023). Instead, individuals in our study may have been predicting whole scenes at a relatively coarse (vs detailed) level, leading to the predictive representations observed in higher order, scene-specific visual regions such as PPA and RSC (Epstein, 2008). We have added a section to the Discussion about this topic:

“Unlike hippocampus, we were unable to detect bidirectional temporal structure representations in visual cortex. This result is seemingly in contrast to a rich literature showing predictive representations in early visual regions¹¹. One possibility for why we were unable to detect predictions in early visual cortex is due to the nature of our stimuli. Past work decoding predictions from early visual regions has tended to use relatively simple stimuli such as gratings^{13,14}, fractals¹⁶, or specific spatial locations⁵⁹. In contrast, we used rich, naturalistic scenes that were experienced in immersive virtual reality from multiple viewpoints, making it unlikely that participants were generating predictions of low-level visual features tied to specific retinotopic locations. Instead, individuals in our study may have been predicting whole scenes at a relatively coarse (vs detailed) level, leading to the predictive representations observed in higher order scene-specific visual regions such as PPA and RSC³⁹. We were also unable to find

evidence for long-timescale predictions in the insula, unlike our prior work investigating predictive hierarchies across the brain during movie watching¹². The relative paucity of environment-sensitive voxels in the insula may have hurt our ability to detect sequence representations in this region. Alternatively, the environment sequences used in our current study may not have engaged the insula as strongly as a continuously unfolding audiovisual movie stimulus that allowed the generation of social or emotional predictions²³.” (p. 28, Discussion)

Finally, our work differs from past studies of predictive processing in that we found bidirectional representations of both future *and* past states during anticipation. There is limited evidence for such bidirectional representations in prior studies. For example, past work has found evidence for separate past and future representations of sequences when participants were explicitly told to think in the forward or the backward direction during retrieval (Wimmer et al., 2020) or when exposed to items that form a sequence without the demand to anticipate future events (Ekman, 2023). Here, we show that such bidirectional representations are present even when individuals were explicitly told to anticipate future events, with no overt demand to simulate past events. Importantly, we go beyond those prior studies in showing that such bidirectional representations are present in numerous visual areas and are organized hierarchically.

Why were both past and future states represented in the brain in our task? This finding is seemingly at odds with theories of predictive processing, such as Successor Representations (Dayan, 1993; Gershman, 2018), which predict that representations should be future-oriented. Instead, our finding of bidirectional representations might be consistent with event segmentation and temporal context models, which suggest that a whole event is brought online during behavior, including other memories nearby in time from the same event (Manning et al., 2011). Therefore, we believe one possible explanation for our findings is the proximity to an event boundary: if cued with the beginning of an event, individuals may exhibit future oriented predictions, but not retrospective representations. But, if cued with the middle of an event, individuals may need to bring online representations of past states to access their memory for the whole event (Michelmann et al, 2023). Because our sequences were circular, every environment was in the “middle”, i.e., was neither at the beginning nor at the end, potentially explaining our finding of bidirectional rather than future-oriented representations. Regardless, bidirectional representations are not predicted by typical successor representation theories of future prediction, and presents an intriguing avenue for future research to disentangle when bidirectional representations might be present rather than just forward ones. We have made the following changes to the Discussion to reflect this:

“Our results build upon influential theories of prediction in the brain. Graded coding of upcoming events is consistent with successor representation models^{2,24,44,45}, which propose that information about future states becomes cached into the representation of the current state in a temporally discounted manner. These models have been extended to account for multiple timescales of prediction by incorporating different scales of temporal discounting²⁴. In line with these theories, recent work has shown that multiple timescales of prediction are represented simultaneously in the brain, with progressively further-reaching predictions in progressively more anterior brain

regions¹² and with relatively less evidence for far away vs nearby predictions^{22,46}. Converging with this past work, in our prediction task we found that patterns for nearby vs far away environments (defined in an independent task) were activated at multiple scales in a hierarchical manner across brain regions. Strikingly, although our asymmetric Gaussian analysis was designed to allow differential coding of the future vs the past, representations were not uniquely biased toward future states. Instead, the hippocampus and visual system represented temporal structure bidirectionally, with graded representations into the past and future. This finding is seemingly at odds with successor representation models^{44,47}, which assume that representations should be future-oriented. Taken together with prior work showing that hippocampal representations of temporal sequences^{46,48,49} can be flexibly biased in either the forward or backward direction based on task demands³⁴, our findings suggest that representations of the past and future can exist simultaneously within the hippocampus, even though the task demands were to anticipate future states.

Why were both past and future states represented in the brain in our task? Our finding of bidirectional representations might be consistent with event segmentation and temporal context models, which suggest that a whole event is brought online during behavior, including other memories nearby in time from the same event³⁵. Therefore, we believe one possible explanation for our findings is the proximity to an event boundary: if cued with the beginning of an event, individuals may exhibit future oriented predictions, but not retrospective representations. But, if cued with the middle of an event, individuals may need to bring online representations of past states to access their memory for the whole event⁵⁰. An important distinction between our experiment and past studies of prediction is that our sequences were circular and temporally extended, whereas sequences in prior studies tended to have a clear end point (i.e. were linear instead of circular)^{12,22,37,46,51} or were shorter¹⁹. Because our sequences were circular, environments were neither at the beginning nor at the end of the sequence, potentially explaining our finding of bidirectional rather than future-oriented representations. Thus, our findings present an intriguing avenue for future research to disentangle when bidirectional representations might be present, in line with temporal context models, rather than just forward ones, in line with successor representations.” (p. 24-25, Discussion)

14. *This sentence in the discussion is unclear: ‘In line with these theories, recent work has shown that multiple timescales of prediction are represented simultaneously in the brain¹², with less evidence for further predictions^{22,41}.’*

Response: Thank you for requesting this clarification. We have changed this sentence to the following:

“These models have been extended to account for multiple timescales of prediction by incorporating different scales of temporal discounting²⁴. In line with these theories, recent work has shown that multiple timescales of prediction are represented simultaneously in the brain, with progressively further-reaching predictions in progressively more anterior brain regions¹² and with relatively less evidence for far away vs nearby predictions^{22,46}.” (p.24, Discussion)

15. *The conclusions are somewhat contradictory. First it is stated that: ‘Visual cortex (V1-4) primarily represented the current environment’ then ‘visual cortex and insula exhibited context-dependent representations in the forward but not backward direction’. This confusion about whether/not there is representation of future environments in visual cortex, might be due to the mix-up of different analysis approaches. For the initial analyses in figure 4, there was no formal test for representation of future environments in visual cortex.*

Response: Thank you for pointing out this inconsistency. We agree that our initial framing of the results was contradictory due to the differing nature of the representations in visual cortex during the cue and the blank screen period and due to the different analyses we used across these periods of the task. That is, across analyses we investigated different periods of the task (cue vs. blank screen) and used different methods (Gaussian analysis vs. pattern similarity separately for each step in the future). We see now how that was confusing and muddled our results.

Visual cortex primarily represented the current moment during the cue period (based on the Gaussian analysis), but weakly represented one step in the future during the blank screen period (based on pattern similarity analyses that were done separately for one and two steps). As per your suggestion to streamline our analyses above, we have removed the pattern similarity approach separately for each step in the future and past, and instead focused on the Gaussian analysis. Thus, we no longer discuss the very weak effect in visual cortex (which would not have survived multiple comparisons corrections across steps in the future), in which it represented one step in the future during the blank screen only. This allows for a simpler conclusion: during the cue period, as assessed by the Gaussian analysis, visual cortex primarily represented the current moment, but not past or future states. As noted elsewhere and again below, we also conducted the Gaussian analysis for the blank screen period, but did not find evidence for past or future states during the blank screen.

We also appreciate your request to clarify the formal test for future representations in our Gaussian analysis. To do so, we compared the Gaussian fit for correctly ordered data vs. a shuffled null, excluding pattern similarity to the cue environment from both datasets (see response to Comment #1). As mentioned in response to earlier comments, the Gaussian fit in visual cortex was no longer significant when compared to the shuffled null excluding the cue ($p = 0.848$), suggesting that it was primarily representing the cue environment but not surrounding environments in the sequence. This indicates that there was no evidence for graded representations of past or future environments. We have included this clarification to our interpretation in the Results and Discussion sections as follows:

“However, when pattern similarity to the cue environment was excluded, the Gaussian model was no longer a better fit for correctly ordered vs. shuffled data (correctly ordered data vs. shuffled null excluding cue, $p = 0.848$, $R^2 = -0.006$; **Figure 4c**). This indicates that the significance of the Gaussian was driven by the cue environment but not by graded similarity to nearby environments in the sequence.”(p. 12-13, Results)

“As expected, the Gaussian fit in visual cortex for the uncued path was not better than the fit for the shuffled null excluding the cue environment ($p = 0.993$, $R^2 = -0.0173$;

Figure 5a), and the Gaussian fit was not significantly different for the cued path vs the uncued path ($p = 0.132$), indicating that visual cortex did not represent surrounding environments on either the cued or uncued path.” (p. 17, Results)

“Unlike hippocampus, we were unable to detect bidirectional temporal structure representations in visual cortex. This result is seemingly in contrast to a rich literature showing predictive representations in early visual regions¹¹. One possibility for why we were unable to detect predictions in early visual cortex is due to the nature of our stimuli. Past work decoding predictions from early visual regions has tended to use relatively simple stimuli such as gratings^{13,14}, fractals¹⁶, or specific spatial locations⁵⁹. In contrast, we used rich, naturalistic scenes that were experienced in immersive virtual reality from multiple viewpoints, making it unlikely that participants were generating predictions of low-level visual features tied to specific retinotopic locations. Instead, individuals in our study may have been predicting whole scenes at a relatively coarse (vs detailed) level, leading to the predictive representations observed in higher order scene-specific visual regions such as PPA and RSC³⁹.” (p. 28, Discussion)

16. The conclusion that ‘graded retrospection of past states can occur alongside prediction of future ones.’ is based only on the analyses of the Gaussian fits. Given the limitations I mentioned above, I think that there is not sufficient evidence to support this.

Response: Thank you for this suggestion. We agree that the original evidence for graded representations was weak. To address this, we further probed the gradedness of the representations in hippocampus and visual cortex by comparing the Gaussian fit to the correctly ordered data vs. the shuffled null when excluding pattern similarity to the cue from both the correctly ordered and shuffled datasets (see response to previous comments). As discussed in earlier comments, we found that hippocampus showed a significant Gaussian fit compared to the shuffled null excluding the cue ($p = 0.023$), suggesting that it did indeed represent future and past states in a graded manner. We believe that the comparison to the shuffled null excluding the cue now provides stronger evidence for graded representations of past and future states in the hippocampus.

17. In the the null models in which the order of the pattern similarity values was shuffled, was the order of all environments shuffled, or only those in the past and the future?

Response: Thank you for requesting this clarification. In the initial manuscript, we shuffled the order of all environments 10,000 times, including the cue environment (“shuffled null including cue”). In this revision, we added a new analysis in which we removed the cue environment from both the correctly ordered and shuffled data (“shuffled null excluding cue”). This approach allowed us to specifically test the gradedness of representations for past and future environments. We have now clarified this in the manuscript as follows:

“To test whether the parameters of the Gaussian curve were consistent across participants, we fit the asymmetrical Gaussian curve on all but one participant’s data (e.g. P2-32) and then measured the sum of squared errors (observed vs. predicted

pattern similarity values) when using this curve to predict the held-out participant's data (e.g. P1). We repeated this procedure for each choice of held-out participant to obtain an average error value. We conducted two tests. First, we compared the Gaussian fit to correctly ordered data and the Gaussian fit to shuffled pattern similarity values, including pattern similarity to the cue environment ("shuffled null including cue"). Second, we conducted the same comparison but removed pattern similarity to the cue environment from both the correctly ordered and shuffled data ("shuffled null excluding cue"). In both cases, we fit the Gaussian to shuffled pattern similarity values 10000 times and obtained the goodness of fit each time to create a null distribution. The goodness of fit for the correctly ordered data vs. null distributions were then compared. If, for a given brain region, the Gaussian model provides a better fit to the correctly ordered vs. shuffled data both when the cue is included and when it is excluded, that would indicate that the brain region's representations of both the cue and graded representations of nearby environments contribute to a significant Gaussian fit. If, however, the Gaussian model provides a better fit to the correctly ordered vs. shuffled data only when the cue is included and not when it is excluded, that would indicate that its Gaussian fit was driven by its representation of the cue environment without any evidence for graded representations of the past or future." (p. 47, Methods)

18. The ROIs should be visualized more clearly and descriptive statistics about each ROI should be provided. How many voxels did each ROI contain?

Response: Thank you for requesting this additional information. We agree that we could visualize the ROIs more clearly. We have now included larger figures of each of our ROIs in Supplementary Materials as follows:

Supplementary Figure 5 | Regions of Interest. Visualization of the three a priori ROIs analyzed in the experiment. Anatomical ROIs for hippocampus (**a**), visual cortex (**b**), and insula (**c**) are shown in white. Within each anatomical ROI, voxels that reliably discriminated between environments during the Localizer Task are shown in red. The voxels in red make up the conjunction ROIs used in our analyses. MNI coordinates are indicated in square brackets.

We also appreciate your request to report the voxel numbers in each ROI. There were 2931 voxels in visual cortex, 156 voxels in hippocampus, and 84 voxels in insula. For our exploratory searchlight ROIs, there were 631 voxels in PPA and 888 voxels in RSC. The differences in the number of voxels within each ROI are because we selected voxels within each region that showed across-participant discrimination of visual environments. Therefore, it is expected that visual cortex, which is strongly modulated by visual input, would more easily distinguish environment representations across participants than hippocampus, where across participant

representations are harder to detect (Koch et al., 2020; Chen et al., 2021). Nevertheless, we do not believe that ROI size impacted our results because we find *more* evidence for graded representations in hippocampus, which is a smaller ROI, than visual cortex.

The insula ROI was smaller than our other ROIs, and it was the only a priori ROI that did not show evidence for a significant Gaussian fit. This null effect could be partly due to the small ROI size, which reflects the relative paucity of environment-sensitive voxels in the insula (see our response to your comment #9 and comment #13).

We now report the number of voxels in each ROI in the manuscript as follows:

“There were 2931 environment-reliable voxels in visual cortex, 156 environment-reliable voxels in hippocampus, and 84 environment-reliable voxels in insula. We also included PPA and RSC as exploratory ROIs based on our searchlight analyses (see below). There were 631 voxels in PPA and 888 in RSC.” (p. 44, Methods)

We also discuss the size of the insula ROI in the Discussion:

“We were also unable to find evidence for long-timescale predictions in the insula, unlike our prior work investigating predictive hierarchies across the brain during movie watching¹². The relative paucity of environment-sensitive voxels in the insula may have hurt our ability to detect sequence representations in this region. Alternatively, the environment sequences used in our current study may not have engaged the insula as strongly as a continuously unfolding audiovisual movie stimulus that allowed the generation of social or emotional predictions²³.” (p. 28, Discussion)

19. Voxels included in the ROIs showed across participant reliability $r > 0.1$. Which of these voxels showed statistically significant representations of the environments?

Response: Thank you for this question. We agree that it is important to establish the robustness of environment-specific representations. However, the purpose of computing the single-voxel reliability score was not to establish significance of environment representations at the individual voxel level. Rather, we wanted to identify voxels that would contribute to multi-voxel patterns of brain activity that reliably differentiate the same environment from different environments. To do so, we selected $r > 0.1$ as our cut-off, following the threshold used in prior work to detect reliable across-participant representations (Chen et al., 2016; Simony et al., 2016). The same vs different environment multi-voxel patterns shown in Figure 3d therefore illustrate the robustness of environment-specific coding in our ROIs as a whole. We did not assess the statistical significance of these pattern similarity differences (same vs. different environments) because we purposely selected voxels whose activity consistently varied, across participants, as a function of the environment – the analysis would therefore be circular.

“We selected 0.1 as our cutoff following the threshold used in prior work to detect reliable across-participant representations^{82,83}. This threshold resulted in reasonable

spatial coverage while maintaining voxel reliability, including in our a priori regions of interest⁸⁰.” (p. 43, Methods)

Reviewer #1 (Remarks to the Author):

Tarder-Stoll and colleagues revised their manuscript comprehensively, including new analyses and reframing some of their claims. The new manuscript is much improved, and there are only a few minor outstanding points to consider, detailed below.

Response to major comment #2. The new analyses of the uncued path are very welcome and reassuring. It would be helpful if you could also directly compare the Gaussian fits of the cued and uncued paths, rather than solely comparing each to a shuffled baseline. This comparison could be either in terms of goodness-of-fit, or of the amplitude or width of the Gaussians.

Also in regard to major comment #2, I appreciate that analysing the blank screen period separately from the cue period is somewhat problematic given the slow BOLD response. However, this also applies to the visual cortex. I would therefore suggest including the blank screen period analysis for either both visual cortex and hippocampus or neither, to avoid suggestions of cherry picking.

Response to minor comment #2. I understand that there may not be sufficient trials to assess this, and I agree that the non-significant goodness-of-fit results on incorrect trials are not worth including in the paper. One final suggestion I might make is to test whether the *width* of the Gaussian was smaller on incorrect than correct trials, maybe specifically in the forward direction, suggesting less forward-looking leads to less successful behaviour. This is just a suggestion, I'm not insisting you run this analysis, given the small proportion of error trials.

Response to minor comment #3. I still don't understand why there would be across participant consistencies in voxel patterns. Why would a given voxel like a given image more than another image in all participants? This seems especially puzzling given that each individual constructed their own idiosyncratic narrative of the paths. Given individual differences in anatomy and head position etc in the scanner, these patterns must be fairly coarse to generalise across participants, right? Do some environments activate anterior portions of the hippocampus more and others posterior portions more? Or are there lateral vs. medial differences? E.g. because scene images activate anterior hippocampus more than non-scene images? I'm just curious what kind of activity pattern differences would be reliable enough and coarse enough to result in across-participant consistency. I don't expect the authors to have a definitive answer to this, but any insights would be welcome.

Reviewer #2 (Remarks to the Author):

In this revision, the authors have added additional controls and simulations which greatly enhance the rigor of their approach.

I only have two remaining comments, both minor:

1) Given the revised results which demonstrate that early visual cortex does not represent the temporal structure (as shown in Figure 4), the authors may wish to revise this sentence in the abstract: "Temporal structure was represented in the hippocampus and across visual regions....". Perhaps it would make more sense to refer to the visual regions as "higher order visual

regions” in order to indicate regions like PPA / RSC, rather than V1-V3?

2) Another minor issues, perhaps semantic, and yet probably worth mentioning: I remain somewhat unconvinced about the interpretation of the negative patterns correlations and negative asymptotes. In particular, the text states: “a negative asymptote reflects suppression of some environments (i.e. the cue pattern is anticorrelated with some environment templates).” The term “suppression” is a functional ascription indicating that a neural representation is less active, below some kind of baseline (as if it is being functionally inhibited), whereas the finding of “anti correlation” is simply a descriptive feature of the data. If the authors really want to link the functional interpretation (suppression) and the descriptive statistic (anticorrelation), perhaps they could say a little more about why a correlation of zero is expected for “neutrally related” images and environments (when no functional suppression is taking place), when the data is collected and preprocessed in precisely the way it was in this manuscript. To be clear: the connection between these pattern correlations and behavior (with slower reaction times for more suppressed environments) is good evidence that the representations of some environments is meaningfully weaker — but just because the representation is weaker that does not (necessarily) mean that it is "suppressed". What if the negative pattern correlations are expected for neutral representations (neither activated nor suppressed)? It is possible that the data analysis already shows that zero correlation is expected when there is no suppression, but this was not clear to me.

I thank the authors for their open and even-handed approach to the science.

Reviewer #3 (Remarks to the Author):

I am happy with the adjustments that the authors made to the paper. I think the evidence is much more convincing now and discussion is much clearer. However, the conclusion that there is representation of both future and past environments in hippocampus and visual areas does warrant further support. The claim is based on two observations: 1. when the cue environment was excluded, the Gaussian was a better fit to the correctly ordered vs. shuffled data and 2. there is no significant difference between backward and forward widths. However, the lack of a significant difference between the two does yet provide evidence that they are the same. And the comparison between the fit of the ordered and shuffled data, tests the fit of asymptote together with both widths. This does not give evidence that both the forward and backward widths are above zero.

There are several things the authors could do to further investigate whether there truly is a representation of both the future and the past environments.

1. Separately shuffle the data for the environments in the future and the past and test which of these shuffles significantly decreases the model fit. However, with the current design I am not sure if that would be possible.
2. Another way to compare these widths against other regions, as was done in the paper. However, I noticed that this was not done separately for the forward and backward widths: ‘The width of the Gaussian in hippocampus was significantly wider than that in visual cortex (beta = 1.379, 95% CI = [0.559, 2.199], p = 0.0023), indicating that hippocampus represented more distant environments.’ So, I think it needs to be done separately for all model parameters.
3. In a similar way, the forward and backward widths could be compared to the widths on the uncued path.

4. Bayesian statistics could be used to compare the backward and forward fit to see if there is evidence for no difference between the two.
5. It would also be great if the authors could plot the width of the Gaussian fit for V1, V2, V3, V4, and PPA, for the forward and backward widths separately. And test these separately in relation to the observed hierarchy. The same holds for the association between the y-coordinate in PPA and RSC in relation to the width.

REVIEWER COMMENTS

Reviewer #1 (Remarks to the Author):

Tarder-Stoll and colleagues revised their manuscript comprehensively, including new analyses and reframing some of their claims. The new manuscript is much improved, and there are only a few minor outstanding points to consider, detailed below.

Response to major comment #2. The new analyses of the uncued path are very welcome and reassuring. It would be helpful if you could also directly compare the Gaussian fits of the cued and uncued paths, rather than solely comparing each to a shuffled baseline. This comparison could be either in terms of goodness-of-fit, or of the amplitude or width of the Gaussians.

Response: Thank you for this suggestion. We agree that it is important to directly test differences between the Gaussian fit when environments are arranged on the order of the cued and the uncued paths to make claims that hippocampal representations are context dependent. We now realize that the way we explained the comparison of the Gaussian fit along the cued vs uncued path in our previous revision may have been unclear.

We first calculated the sum of squared errors of the Gaussian fit for the correctly ordered data and the mean sum of squared errors of the Gaussian fits for the shuffled data, separately for the cued and uncued path. These comparisons tell us whether the Gaussian model provided a good fit for the cued path and for the uncued path, separately; as noted by the reviewer, however, these comparisons do not involve contrasting the cued and uncued paths to each other. To accomplish that, we computed the difference between the sum of squared errors for 1) the correctly ordered data along the *cued* path vs the correctly ordered data along the *uncued* path, excluding the cue environment from both and 2) the same cued vs uncued comparison for each of the shuffled nulls. We then tested whether the difference in fit for the cued vs uncued path in the real (unshuffled) data was larger than the difference in fit for the cued vs the uncued path in the shuffled nulls. This directly tests whether the pattern similarity to upcoming environments was better modeled by a Gaussian along the cued path vs the uncued path – moreso than what would be expected for null (permuted) data.

Note that in the first revision, we described this approach a bit differently, which we believe may have caused confusion about our approach. What we did in the first revision vs. this second revision are mathematically equivalent, but we hope our revised explanation of the approach is now clearer.

We also realized that our use of shuffled data for the comparison of the cued vs. uncued path may have created confusion. One may ask why we need shuffled data, if we can just directly compare the cued vs. uncued model fits. The reason that we used shuffled data is to compute a p-value for whether the cued vs. uncued model fit difference is significantly different from what would be expected from a null distribution in which there are no true differences. Therefore, to test for context-dependence, we obtained the difference in errors between the observed cued vs uncued paths and statistically tested the significance of this difference compared to the difference in errors between the shuffled null distributions for the cued vs uncued paths. Without inclusion of the shuffled nulls, we would not be able to compute a p-value because the difference between the error on the cued and the uncued path is a single number.

As noted in the first revision, in the hippocampus, the Gaussian fit was significantly better for the order of the cued path compared to the uncued path (sum of squared errors of correctly ordered data vs. shuffled null excluding cue for cued vs uncued path, $p = 0.005$). In contrast, the Gaussian fit in visual cortex was not significantly different for the cued path vs the uncued path (sum of squared errors of correctly ordered data vs. shuffled null excluding cue for cued vs uncued path, $p = 0.132$). This suggests that hippocampus, but not visual cortex, represented bidirectional temporal structure in a context-dependent manner.

We have modified the manuscript to clarify this analysis. Note that the updated text clarifies the approach we used to statistically test for context-dependent representations – but as noted above, it is mathematically the same as the way we described the analysis in the previous revision.

“We also **directly** assessed whether the cued order Gaussian fit was significantly better than the uncued order Gaussian fit. **We computed the difference between the sum of squared errors for 1) the correctly ordered data along the cued path vs the correctly ordered data along the uncued path, excluding the cue environment from both and 2) the same cued vs uncued comparison for each of the shuffled nulls. We then tested whether the difference in fit for the cued vs uncued path in the real (unshuffled) data was larger than the difference in fit for the cued vs uncued path in the shuffled nulls.** This allowed us to **statistically** test whether there was more evidence for graded representations of the past and future along the cued vs. uncued path in the absence of similarity to the cue environment.” (Methods, p., 50)

Also in regard to major comment #2, I appreciate that analyzing the blank screen period separately from the cue period is somewhat problematic given the slow BOLD response. However, this also applies to the visual cortex. I would therefore suggest including the blank screen period analysis for either both visual cortex and hippocampus or neither, to avoid suggestions of cherry picking.

Response: Thank you for raising this important point. We agree with you that it is important to unify our analysis approach to avoid the impression that we selected analyses based on which gave the optimal outcome. Further, we also agree that our blank screen period analysis in visual cortex faces the same interpretational difficulties described in our first revision because of the slow BOLD response. As such, we removed the blank screen analyses (for both visual cortex and hippocampus) and the manuscript focuses solely on our Gaussian modeling approach during the cue screen period.

*Response to minor comment #2. I understand that there may not be sufficient trials to assess this, and I agree that the non-significant goodness-of-fit results on incorrect trials are not worth including in the paper. One final suggestion I might make is to test whether the *width* of the Gaussian was smaller on incorrect than correct trials, maybe specifically in the forward direction, suggesting less forward-looking leads to less successful behaviour. This is just a suggestion, I'm not insisting you run this analysis, given the small proportion of error trials.*

Response: Thank you for this comment. We agree it would be interesting if the width of the Gaussian was smaller on incorrect vs correct trials. However, in our analysis approach in the paper, we take a multistep approach to assessing the Gaussian fit: we first test the significance of the Gaussian fit for the correctly ordered data vs. the shuffled null including and excluding the cue. Then, only if the overall Gaussian fit is significant, we interrogate the model parameters, such as amplitude and width. We do not proceed with further analysis of the model parameters if the Gaussian is not significant because the parameters are difficult to interpret when the model does not fit the underlying data. For example, we report that the Gaussian fit in insula was not significant ($p = 0.132$), but do not report or interrogate the model parameters. For consistency, we opted to take the same approach here: because the Gaussian model fit for incorrect trials was not significantly better than the shuffled null, we did not proceed with the investigation of the model parameters as their interpretation would not be straightforward. We have clarified this in the manuscript as follows:

“We obtained a p-value by calculating the fraction of 10,000 shuffles that produced Gaussian fits with lower error than the fit for the correctly ordered data. We only proceeded with tests of the model parameters (i.e., amplitude, asymptote, width) if the Gaussian fit for the correctly ordered data vs. the shuffled null was statistically significant. We also calculated R^2 as 1 minus the sum of squared errors of the Gaussian fit for the correctly ordered data divided by the mean sum of squared errors of the Gaussian fits for the shuffled data. R^2 values greater than 0 indicate smaller squared errors for the correctly ordered data than the shuffled data.” (Methods, p. 49)

Response to minor comment #3. I still don't understand why there would be across participant consistencies in voxel patterns. Why would a given voxel like a given image more than another

image in all participants? This seems especially puzzling given that each individual constructed their own idiosyncratic narrative of the paths. Given individual differences in anatomy and head position etc in the scanner, these patterns must be fairly coarse to generalise across participants, right? Do some environments activate anterior portions of the hippocampus more and others posterior portions more? Or are there lateral vs. medial differences? E.g. because scene images activate anterior hippocampus more than non-scene images? I'm just curious what kind of activity pattern differences would be reliable enough and coarse enough to result in across-participant consistency. I don't expect the authors to have a definitive answer to this, but any insights would be welcome.

Response: Thank you for requesting this clarification. We agree that the across-participant consistency in environment patterns that we observed was surprisingly robust, especially considering that participants generated idiosyncratic stories. Below, we describe a number of reasons why these shared patterns may be observable in our experiment and in other studies.

We agree that individual differences in anatomy and head position can lead to large variations in voxel location. To mitigate the influence of these differences, we coregistered each participant's data to an MNI template and spatially smoothed the data. These preprocessing steps made it possible to compare voxel-wise patterns across participants, regardless of differences in anatomy and scanner position. It also means, as noted by the Reviewer, that the across-voxel patterns we observed are likely to be relatively coarse patterns, such that the patterns are robust to imperfections in co-registering different brains into a common space.

Across-participant similarity in patterns may be relatively expected in cases of brain regions whose activity is very strongly tied to sensory input, but we agree that it is especially striking in the case of the hippocampus – whose activity is heavily modulated not just by sensory input but also by idiosyncratic internally oriented processes. We found that the majority of the environment-reliable voxels in hippocampus were in its anterior portion, which—as you mentioned in your comment—is in line with prior work showing that anterior hippocampus is more activated by perceiving, remembering, and imagining scenes, relative to objects (Zeidman et al., 2015). Further, attention to the spatial layout of a scene recruits anterior hippocampus more than posterior hippocampus (McCormick et al., 2021). Finally, recent research has shown representations of visual space in the hippocampus, particularly its anterior portions (Silson et al., 2021). Together, these findings support the idea that anterior hippocampus is sensitive to the visual properties of scenes. One possibility is that the spatial layout of the environments were particularly salient to individuals in our task because they were encoded in immersive virtual reality. Perception and imagination of the layout of these environments may lead to the shared environment responses we observed in anterior hippocampus, because the layout was shared across participants and spatial layout is known to drive anterior hippocampal activity. Because we had relatively few environment-reliable voxels in posterior hippocampus, we could not systematically test whether some environments differentially drove anterior vs. posterior

hippocampus; but we agree that future work exploring anterior vs. posterior or medial vs. lateral differences in across-participant hippocampal scene representations would be useful.

More broadly, our work extends prior research showing shared activity patterns across participants in numerous areas of the brain (Hasson et al., 2008). Consistency in these inter-individual activity patterns is present both during perception (often of movie or narrative stimuli that unfold over time) and, crucially, during recall as well. These shared patterns are detectable across a range of brain regions, including visual, medial temporal, and default mode regions (Chen et al., 2017). Therefore, prior work concurs with our findings in suggesting that there are indeed coarse maps that line up across individuals in MNI space, leading to detectable shared activity patterns in response to shared experiences. That these shared patterns are also present during memory recall suggests that idiosyncracies in what is remembered and how it is remembered are not so substantial as to swamp detection of shared patterns. Likewise, for our studies, differences in the generated stories across participants did not prevent us from detecting shared representations of the environments – there was enough commonality in the visual experiences, it seems, for these shared patterns to emerge. We have included a discussion of these shared representations in the manuscript as follows:

“Strikingly, we observed such effects in hippocampus even with environment templates that were identified across participants. The finding of reliable hippocampal activity patterns for individual environments across participants adds to an emerging body of work demonstrating shared hippocampal representations across individuals, representations that were previously difficult to detect^{57,58}. We speculate that these shared representations may include information about the perceived and/or imagined spatial layout of the scenes, given prior work linking the hippocampus to representations of attended spatial configurations^{59–64}. Further, these hippocampal representations of visual scenes are consistent with a burgeoning line of work linking this region to visual representations more broadly^{65–67}. To be detectable across individuals, these shared hippocampal representations are likely to be fairly coarse, similar to other across-individual representations that have been identified across the cortex⁶⁸.”
(Discussion, p. 28)

Reviewer #2 (Remarks to the Author):

In this revision, the authors have added additional controls and simulations which greatly enhance the rigor of their approach.

I only have two remaining comments, both minor:

1) Given the revised results which demonstrate that early visual cortex does not represent the temporal structure (as shown in Figure 4), the authors may wish to revise this sentence in the abstract: “Temporal structure was represented in the hippocampus and across visual regions.... “. Perhaps it would make more sense to refer to the visual regions as “higher order visual regions” in order to indicate regions like PPA / RSC, rather than V1-V3?

Response: Thank you for this astute comment. We agree that we should clarify that specifically higher order visual regions represented bidirectional temporal structure. We have updated the manuscript with your suggested text as follows:

“Temporal structure was represented in the hippocampus and across **higher-order** visual regions (1) bidirectionally, with graded representations into the past and future and (2) hierarchically, with further events into the past and future represented in successively more anterior brain regions.” (Abstract)

2) Another minor issues, perhaps semantic, and yet probably worth mentioning: I remain somewhat unconvinced about the interpretation of the negative patterns correlations and negative asymptotes. In particular, the text states: “a negative asymptote reflects suppression of some environments (i.e. the cue pattern is anticorrelated with some environment templates).” The term “suppression” is a functional ascription indicating that a neural representation is less active, below some kind of baseline (as if it is being functionally inhibited), whereas the finding of “anti correlation” is simply a descriptive feature of the data. If the authors really want to link the functional interpretation (suppression) and the descriptive statistic (anticorrelation), perhaps they could say a little more about why a correlation of zero is expected for “neutrally related” images and environments (when no functional suppression is taking place), when the data is collected and preprocessed in precisely the way it was in this manuscript. To be clear: the connection between these pattern correlations and behavior (with slower reaction times for more suppressed environments) is good evidence that the representations of some environments is meaningfully weaker — but just because the representation is weaker that does not (necessarily) mean that it is “suppressed”. What if the negative pattern correlations are expected for neutral representations (neither activated nor suppressed)? It is possible that the data analysis already shows that zero correlation is expected when there is no suppression, but this was not clear to me.

Response: Thank you for requesting this clarification. We agree that it would not be appropriate to describe a negative pattern similarity value as suppression without comparison to an appropriate baseline. Based on your comment, we now realize we were not clear in the description of how we determine suppression. We have clarified this analysis and interpretational approach below.

You are correct that a negative asymptote in our Gaussian fit indicates an anticorrelation between the cue screen and environment templates but, importantly, does not indicate suppression of environment representations per se. Therefore, to determine whether an asymptote is suppressed, we always statistically test whether the asymptote is lower than a baseline value consisting of pattern similarity to environments from the different map (i.e. environments that are not relevant for the current trial). Because the environments from the different map are subject to the same preprocessing steps but are not relevant for making predictions about temporal structure on a given trial, their pattern similarity with the cue screen can be considered a neutral pattern similarity baseline. In other words, the *relative* value of the asymptote in comparison to the different map baseline allows us to interpret whether the environments represented by the asymptote are *relatively* suppressed, regardless of the asymptote's absolute value. Based on this analysis approach, a brain region with a negative asymptote that is significantly lower than the different-map baseline would be interpreted as having *relatively* suppressed representations of those environments – i.e., suppressed relative to our baseline condition. In contrast, a brain region with a negative asymptote that is *not* significantly lower than the different-map baseline would *not* be interpreted as showing suppression effects: the anticorrelation does not in itself indicate suppression. We have edited our manuscript to ensure that it is clear that the key comparison is that between the asymptote and the different-map baseline:

“The asymptote controls the vertical shift of the Gaussian curve. This asymptote was compared to a baseline value consisting of pattern similarity between the cue and environments from the other map (different-map baseline), because these other-map environments are never accurate predictions from this cue. If the asymptote is lower than this baseline, that would reflect relative suppression of some environment templates in the current map (relative to environments that are currently irrelevant).” (Methods, p. 47)

“We compared the amplitude and asymptote to a baseline that consisted of the pattern similarity values between the cue screen activity pattern and the activity pattern templates for the eight environments that were not in the cued map (different-map baseline). We tested whether the amplitude was significantly above the different-map baseline, and whether the asymptote was significantly below the different-map baseline, using one-sample t-tests across participants. A brain region with an asymptote that is significantly lower than the different-map baseline would be interpreted as having relatively suppressed representations of those environments, compared to our baseline condition.” (Methods, p. 49)

“if the asymptote is lower than baseline (defined as pattern similarity between the cue screen activity patterns and all environment templates from the other map, henceforth referred to as different-map baseline), this suggests that these environments are suppressed relative to our baseline condition.” (Results, p. 11)

“The asymptote was significantly lower than the different-map baseline, suggesting that other environments surrounding the cue were suppressed relative to our baseline (mean = -0.014, standard deviation = 0.009; $t(31) = -5.97$, $p = 0.000001$; **Figure 4c**.” (Results, p. 12)

“In the hippocampus, similar to visual cortex, the amplitude of the Gaussian fit was significantly higher than the different-map baseline (mean = 0.0185, standard deviation = 0.030; $t(31) = 2.959$, $p = 0.005$; **Figure 4d**) and the asymptote was significantly lower than the different-map baseline (mean = -0.007, standard deviation = 0.014; $t(31) = -3.476$, $p = 0.001$; **Figure 4d**). Thus, hippocampus represented the cue environment while it was on the screen and suppressed environments further away, relative to the different-map baseline.” (Results, p. 13)

“We reasoned that relative suppression of environments surrounding the cue, indicated by an asymptote that is lower than the different-map baseline, should interfere with the generation of long timescale predictions: more suppression should be associated with more response time costs for accessing future environments.” (Results, p. 19)

“Hippocampal representations of temporal structure were relevant for behavior: suppression of distant environments (relative to the different-map baseline) was linked to response time costs for anticipating further events.” (Discussion, p. 24)

That said, we agree with you that there is an important difference between *suppression* and *weaker* representations, and weaker representations may not necessarily be suppressed. That is, it is possible that all environments shown in the experiment are relatively activated in mind; and although the environments represented by the asymptote are *less* active than unrelated environments in the other map, they are still ‘activated’ in mind in some sense (e.g., compared to stimuli not in the experiment at all). For this reason, we have gone through the manuscript to ensure that we talk about suppression *relative* to the different-map baseline. We can consider the environments represented by the asymptote to be *relatively* suppressed compared to that baseline, but they may or may not be suppressed relative to other representations. We have clarified this in the Discussion section as follows:

“However, it is important to note that 1) the observed suppression was calculated relative to the different-map baseline, and therefore only indicates relative suppression compared to representations of other environments in the experiment, and 2) the observed individual-differences correlation was observed with a relatively small sample size, requiring replication in future work. With respect to the first point, environments may be suppressed relative to the different-map baseline but not suppressed compared to ongoing task-irrelevant thoughts or experiences; if so, the environments represented by the asymptote may be considered to be more *weakly* represented, rather than suppressed in an absolute sense. Future work could add additional comparisons or use alternative neuroimaging techniques to determine the level of suppression, and test

whether the widths of brain regions' predictive horizons influence behavioral performance in studies specifically powered for individual differences analyses.” (Discussion, p. 26-27)

I thank the authors for their open and even-handed approach to the science.

Response: Thank you so much for these kind words, and thank you for your thorough, incisive, and helpful comments!

Reviewer #3 (Remarks to the Author):

I am happy with the adjustments that the authors made to the paper. I think the evidence is much more convincing now and discussion is much clearer. However, the conclusion that there is representation of both future and past environments in hippocampus and visual areas does warrant further support. The claim is based on two observations: 1. when the cue environment was excluded, the Gaussian was a better fit to the correctly ordered vs. shuffled data and 2. there is no significant difference between backward and forward widths. However, the lack of a significant difference between the two does yet provide evidence that they are the same. And the comparison between the fit of the ordered and shuffled data, tests the fit of asymptote together with both widths. This does not give evidence that both the forward and backward widths are above zero.

There are several things the authors could do to further investigate whether there truly is a representation of both the future and the past environments.

Response: Thank you for these ideas about additional analyses to support our conclusion that both future and past environments are represented in the hippocampus. We discuss each suggestion in turn below; we found that approaches #2 and #5 provided the strongest evidence that both forward and backward widths are meaningful and vary systematically (with the other approaches not providing clear evidence). Thank you for these suggestions, which we believe have strengthened the paper and our conclusion that temporal structure is represented bidirectionally in the brain.

1. Separately shuffle the data for the environments in the future and the past and test which of these shuffles significantly decreases the model fit. However, with the current design I am not sure if that would be possible.

Response: Thank you for this suggestion. It is possible with our current design to separately shuffle the environments in the forward and backward direction to test whether the significance

of our Gaussian fit compared to the shuffled null was more strongly driven by a single direction. We have done as you suggested and conducted two additional tests of the Gaussian fit: in the first test, we shuffled the environments in the forward direction, while keeping the environments in the backward direction intact. In the second test, we shuffled the environments in the backward direction while keeping the forward direction intact. Akin to the shuffled null excluding the cue, we removed the cue environment from both the observed and shuffled data in the forward model and the backward model. However, note that one statistical challenge with this analysis is that it involves shuffling only three datapoints (for each choice of held-out participant), substantially reducing its power compared to shuffling in both directions.

If only one of these shuffles (forward or backward) reduced the fit, that would provide evidence that only one direction is important to the model (i.e. has a meaningfully non-zero width). Instead, when running this analysis in the hippocampus, we found that separately shuffling either the forward or the backward direction numerically decreased the model fit relative to the correctly ordered data. However, this difference was not significant for the forward shuffled null ($p = 0.1187$, $r^2 = 0.0229$) or the backward shuffled null ($p = 0.1456$, $r^2 = 0.0098$). This suggests that the significant fit for our full bidirectional model is not driven purely by one direction, but does not provide conclusive evidence about the relative importance of each direction. Critically, as noted above, this analysis is likely underpowered because it involves shuffling only three datapoints (i.e., steps 1-3 in the forward or backward directions for each choice of held-out participant; step 4 is the same in both directions so could not be included in the shuffles). As such, we opted to not include it in the revised paper, particularly because it is not clear that the results of this analysis would change our interpretations. Additional analyses, reported below, show that both forward and backward widths vary systematically across the brain, and we consider that evidence to more clearly indicate that both widths are meaningful. Those analyses are discussed below.

2. Another way to compare these widths against other regions, as was done in the paper. However, I noticed that this was not done separately for the forward and backward widths: 'The width of the Gaussian in hippocampus was significantly wider than that in visual cortex (beta = 1.379, 95% CI = [0.559, 2.199], p = 0.0023), indicating that hippocampus represented more distant environments.' So, I think it needs to be done separately for all model parameters.

Response: We also agree it would be informative to further investigate the differences in forward and backward width of the Gaussian fit between visual cortex and hippocampus. As you mentioned, the Gaussian fit was wider in hippocampus than visual cortex (beta = 1.379, 95% CI = [0.559, 2.199], $p = 0.0023$). Additionally, there was no interaction between region (hippocampus vs visual cortex) and direction (forward width vs backward width, beta=-0.569, 95% CI = [-1.746, 0.606], $p = 0.346$), suggesting that increased hippocampal widths did not depend on forward vs backward representations. As per your suggestion, we conducted follow-up simple effect analyses to test whether the observed difference in the width of the Gaussian fit between hippocampus and visual cortex was significant for both the forward and backward direction.

Indeed, the widths were significantly larger in hippocampus than visual cortex when examining only the forward width (beta = 1.094, 95% CI = [0.111, 2.077], p = 0.033) and only the backward width (beta = 1.664, 95% CI = [0.621, 2.707], p = 0.003). This suggests further-reaching representations of temporal structure in hippocampus vs visual cortex were bidirectional. We have reported these new analyses as follows:

“We then statistically compared the width parameters in hippocampus and visual cortex, to determine whether hippocampus represented more distant environments surrounding the cue and whether this differed in the forward vs. backward direction. The width of the Gaussian in hippocampus was significantly wider than that in visual cortex (beta = 1.379, 95% CI = [0.559, 2.199], p = 0.0023), indicating that hippocampus represented more distant environments. There was no effect of direction (beta = -0.246, 95% CI = [-0.935, 0.443], p = 0.489), nor a region by direction interaction (beta=-0.569, 95% CI = [-1.746, 0.606], p = 0.346). Further, the widths were significantly larger in hippocampus than visual cortex when separately examining only the forward width (beta = 1.094, 95% CI = [0.111, 2.077], p = 0.033) and only the backward width (beta = 1.664, 95% CI = [0.621, 2.707], p = 0.003).” (Results, p. 14)

3. In a similar way, the forward and backward widths could be compared to the widths on the uncued path.

Response: Thank you for this suggestion. In our previous analysis approach, we only proceeded with the investigation of the model parameters (e.g. width, amplitude) if the overall Gaussian fit was significantly better for the correctly ordered data than for the shuffled data, because model parameters are difficult to interpret when the model does not fit the underlying data. For example, we report that the Gaussian fit in insula was not significant (p = 0.132), but do not report or interrogate the model parameters. For consistency, we opted to take the same approach here: because the Gaussian model fit for the uncued path was not significantly better than the shuffled null, we did not proceed with the investigation of the model parameters as their interpretation would not be straightforward. For this reason, we do not compare the forward and backward widths on the cued and uncued paths. We have clarified this in the manuscript as follows:

“We obtained a p-value by calculating the fraction of 10,000 shuffles that produced Gaussian fits with lower error than the fit for the correctly ordered data. We only proceeded with tests of the model parameters (i.e., amplitude, asymptote, width) if the Gaussian fit for the correctly ordered data vs. the shuffled null was statistically significant. We also calculated R^2 as 1 minus the sum of squared errors of the Gaussian fit for the correctly ordered data divided by the mean sum of squared errors of the Gaussian fits for the shuffled data. R^2 values greater than 0 indicate smaller squared errors for the correctly ordered data than the shuffled data.” (Methods, p. 49)

4. *Bayesian statistics could be used to compare the backward and forward fit to see if there is evidence for no difference between the two.*

Response: Thank you for this valuable suggestion. We agree that it would be informative to test whether the forward and backward widths are equivalent. To do so, we conducted Bayesian models using the brms package in R, which allows us to construct linear mixed-effects models with the same structure as the frequentist models we reported in the manuscript. We ran Bayesian mixed effects regression models 1) predicting the width of the Gaussian fit as a function of region (hippocampus vs visual cortex), direction (forward vs backward), and their interaction and 2) predicting the width of the Gaussian fit as a function of region only. We then obtained a Bayes factor for the comparison of the two models. If the forward and backward directions are equivalent, there should be a preference for the model that does not include direction.

We followed prior conventions for considering relative evidence for the models (Kass & Raftery, 1995; Jeffreys, 1961; Wagenmakers et al., 2011): Bayes Factors (BFs) > 3 would indicate stronger evidence for the model including direction as a factor and BFs $< \frac{1}{3}$ would indicate stronger evidence for the model without direction as a factor. Intermediate BFs would provide only anecdotal or no evidence either way. We found a Bayes Factor of 1.13, which does not provide strong support for either model; instead, it only provides anecdotal evidence in favour of the model that included direction compared to the model that did not include direction. Therefore, we did not find strong evidence that the forward and backward directions are equivalent. However, we do not believe this changes the conclusion of our study: we do not make claims about the *equivalence* of the forward and backward widths. Rather, we merely aim to show that both forward and backward directions are represented in the brain simultaneously in our task. We believe that the new analyses provided in the first revision and those described above and below (response to comment #5) provide evidence for graded, bidirectional representations. However, we have carefully gone through the manuscript to ensure that we do not make claims that these bidirectional representations have *equivalent* widths, given that we do not have Bayesian evidence for this conclusion.

5. *It would also be great if the authors could plot the width of the Gaussian fit for V1, V2, V3, V4, and PPA, for the forward and backward widths separately. And test these separately in relation to the observed hierarchy. The same holds for the association between the y-coordinate in PPA and RSC in relation to the width.*

Response: Thank you for this excellent suggestion. We have created additional supplementary figures plotting the forward and backward width separately in each ROI. As per your suggestion, we have also examined whether the width of the Gaussian fit hierarchically increases from lower to higher level visual regions separately for the forward and backward widths.

To do so, we first obtained the forward and backward widths of the Gaussian fit for each participant separately for V1, V2, V3, V4, and PPA. We then ordered these regions based on their position along the visual hierarchy, with earlier regions (V1) first and the later regions (PPA) last. For each participant we then separately obtained the Spearman's correlation between 1) the forward width and order along the visual hierarchy across regions and 2) the backward width and order along the visual hierarchy across regions. Finally, we compared these correlations to 0 using a t-test. We found that both forward and backward widths successively increased along the visual hierarchy, with successively larger forward and backward widths in successively higher order visual cortex regions (Forward width: $t(31) = 2.337$, $p = 0.026$; Backward Width: $t(31) = 3.025$, $p = 0.005$).

Additionally, the correlation between width and y-coordinate in PPA remained significant when separately examining the forward and backward widths, suggesting further reaching bidirectional representations in more anterior parts of this region (Forward Width: $t(31) = 2.11$, $p = 0.043$; Backward Width: $t(31) = 2.046$, $p = 0.049$). In RSC, the forward width successively increased with y-coordinate ($t(31) = 3.931$, $p = 0.0004$), but the backward width did not change with y-coordinate ($t(31) = -0.256$, $p = 0.799$), suggesting that hierarchically ordered representations were driven by progressive changes in the forward width in this region. Of note, however, the lack of hierarchically organized changes in the backward width as a function of y-coordinate in RSC may have been driven by a wide Gaussian fit in posterior voxels, indicating that even posterior aspects of this region had far-reaching backwards representation of temporal structure (**Supplementary Figure 7**). We have updated the manuscript and Supplementary Materials to include these changes:

“We found a significant positive correlation, indicating that successively later visual cortex regions had successively wider Gaussian fits ($t(31) = 2.136$, $p = 0.041$). This correlation remained significant when only examining the forward direction (\$t(31) = 2.337\$, \$p = 0.026\$ ), and when only examining the backward direction (\$t(31) = 3.025\$, \$p = 0.005\$ ), **Supplementary Figure 6**. It also remained significant when only considering V1, V2, V3, and V4 ($t(31) = 4.322$, $p = 0.0001$).” (Results, p. 23)

“There was a significant positive correlation between width (σ) and y-coordinate, indicating that Gaussian fits became progressively wider in progressively more anterior aspects of both RSC ($t(31) = 2.638$, $p = 0.013$; **Figure 7b**) and PPA ($t(31) = 2.424$, $p = 0.021$; **Figure 7c**). In PPA, the correlation between width and y-coordinate remained significant when separately examining the forward and backward widths, suggesting further reaching bidirectional representations in more anterior parts of this region (Forward Width: \$t(31) = 2.11\$, \$p = 0.043\$; Backward Width: \$t(31) = 2.046\$, \$p = 0.049\$ ), **Supplementary Figure 6**. In RSC, the forward width successively increased with y-coordinate ($t(31) = 3.931$, $p = 0.0004$), but the backward width did not change with y-coordinate ($t(31) = -0.256$, $p = 0.799$, **Supplementary Figure 6**), suggesting that hierarchically ordered

representations were driven by progressive changes in the forward width in this region. There was a negative correlation between amplitude and y-coordinate in PPA ($t(31) = -2.636$, $p = 0.013$; **Figure 7c**), but not RSC ($t(31) = -0.550$, $p = 0.586$; **Figure 7b**).” (Results, p. 21-22)

Supplementary Figure 6 | Forward and Backward Widths Across Visual Regions. The backward and forward width of the Gaussian fit increased hierarchically across early visual cortex from V1 to V4 (Top). In PPA, the forward and backward width of the Gaussian fit increased from posterior to

anterior voxels. In RSC, the forward, but not the backward, width of the Gaussian fit increased from posterior to anterior voxels (Bottom). Data are shown binned by anterior vs. posterior aspects of PPA and RSC, but the statistics reported in the main text treated the anterior-posterior axis of these regions as a continuous variable. Bars indicate the average width across participants and points indicate the width of the Gaussian fit for each participant.

Reviewer #3 (Remarks to the Author):

Thank you for your meticulous work in revising the paper. All my remaining comments have been sufficiently addressed. The improvements you have made have greatly enhanced the support for the conclusions that you draw and the quality of the manuscript as a whole.